# Bridging the Gap Between Average and Discounted TD Learning

Haoxing Tian [1]   Zaiwei Chen [2]   Ioannis Ch. Paschalidis [3]   Alex Olshevsky [3]

## Abstract

The analysis of Temporal Difference (TD) learning in the average-reward setting faces notable theoretical difficulties because the Bellman operator is not contractive with respect to any norm. This complicates standard analyses of stochastic updates that are effective in discounted settings. Although a considerable body of literature addresses these challenges, existing theoretical approaches come with limitations. We introduce a novel algorithm designed explicitly for policy evaluation in the average-reward setting, utilizing sampling from two Markovian trajectories. Our proposed method overcomes previous limitations by guaranteeing convergence to the unique solution of a properly defined projected Bellman equation. Notably, and in contrast to earlier work, our convergence analysis is uniformly applicable to both linear function approximation and tabular settings and does not involve explicit dimension-dependent terms in its convergence bounds. These results align with what is known to hold in the discounted setting. Furthermore, our algorithm achieves improved dependence on the problem's condition number, reducing the sample complexity from quartic, as in prior literature, to quadratic scaling, and thus matching the efficiency seen in the discounted setting.

## 1. Introduction

Reinforcement learning with an average-reward objective is well-suited for applications that focus on the long-term performance over an infinite horizon. This framework has proved valuable in a variety of domains, including con-

trol systems (Prieto-Rumeau & Hernández-Lerma, 2008; Krishnamurthy, 2016), telecommunications (Bertsekas & Gallager, 2021; Altman, 2021), and production environments (Feinberg & Shwartz, 2012). While the discounted setting—emphasizing shorter-term returns—has received extensive theoretical attention, especially for Temporal-Difference (TD) learning (Bhandari et al., 2018; Chen et al., 2022; Tian et al., 2023; Srikant & Ying, 2019), analogous understanding of TD methods under the average-reward criterion is comparatively less developed (Tsitsiklis & Van Roy, 1999; Zhang et al., 2021b).

Technically, compared to the discounted setting the challenge in the average-reward setting is that the Bellman operator for policy evaluation is not a contraction mapping with respect to any norm. Consequently, the solution to the corresponding Bellman equation is not unique. This significantly complicates the theoretical analysis, particularly in the model-free setting where the stochastic dynamics of the Markov Decision Process (MDP) are unknown and the agent can only obtain random samples through interaction with the environment.

We now make the above remarks more concrete by surveying existing results on average-reward TD and comparing them to their counterparts in the discounted setting. The first theoretical analysis of average-reward TD, due to (Tsitsiklis & Van Roy, 1999), addressed the challenges discussed above by introducing an assumption that guarantees uniqueness of the solution to the Bellman equation. In the setting where the true value function $V_\pi(s)$ of a policy $\pi$ is approximated as $V_\pi(s) \approx \phi(s)^\top \theta$, that is, as a linear combination of computable features stacked into the vector $\phi(s)$, (Tsitsiklis & Van Roy, 1999) assumed that for any scalar $c \in \mathbb{R}$ and vector $\theta \in \mathbb{R}^d$, we have $\Phi\theta \neq ce$, where $\Phi$ is the matrix whose rows are the feature vectors $\phi(s)^\top$, and $e$ is the all-ones vector. While this assumption enabled the first convergence guarantee for average-reward TD, it is not satisfied even in the tabular case, where $\{\phi(s)\}$ form the canonical basis and $\Phi$ is the identity matrix. Several subsequent works (Yu & Bertsekas, 2009; Zhang et al., 2021a; Li et al., 2024) adopted the same assumption.

In the tabular setting, (Wan et al., 2021) introduced differential learning and planning algorithms for average-reward MDPs, including off-policy model-free prediction and con-

[1]Department of Electrical Engineering, Boston University, Boston, MA, USA [2]School of Industrial Engineering, Purdue University, West Lafayette, IN, USA [3]Department of Electrical Engineering and Division of System Engineering, Boston University, Boston, MA, USA. Supported by NSF Award 2317079.. Correspondence to: Alex Olshevsky <alexols@bu.edu>.

*Proceedings of the 43rd International Conference on Machine Learning*, Seoul, South Korea. PMLR 306, 2026. Copyright 2026 by the author(s).

trol methods. Their centered Differential TD method converges asymptotically to the actual centered differential value function. This work is complementary to outs: its guarantees are asymptotic and tabular, whereas our focus is finite-sample analysis that applies uniformly to both tabular and linear function approximation.

A more recent study Xie et al. (2026) removed the feature-independence assumption by analyzing linear TD($\lambda$) with arbitrary features in both discounted and average-reward settings. However, since the limit is sample-path-dependent, it may differ across independent runs of the algorithm. A similar result was obtained in the linear function approximation setting in (Zhang et al., 2021b), but without a guarantee that the process converges to a point. A more recent paper (Haque & Maguluri, 2024) establishes convergence to a point, but the convergence rate includes explicit dependence on the dimension $d$ of the parameter vector $\theta$, a dependence that does not explicitly appear in the discounted case.

The recent paper (Li et al., 2024) on actor-critic methods includes some results on policy evaluation in the linear approximation setting, using a fairly intricate nested-loop algorithm based on variance reduction. However, when specialized to the policy evaluation problem, that work also makes the $\Phi\theta \neq ce$ assumption, rendering it inapplicable to the tabular case.

Moreover, we note that their sample complexity is stated in terms of the averaged iterate, whereas the other papers discussed above provide guarantees for the last iterate.

To summarize, while a number of papers have developed a convergence theory for average-reward TD, the existing literature consistently includes some mix of caveats compared to the discounted case, ranging from assumptions that exclude the tabular setting, to a lack of point-wise convergence guarantees for the underlying iterates, or explicit dimension dependence in the convergence bounds. Additionally, we note that the state-of-the-art results from (Zhang et al., 2021a; Haque & Maguluri, 2024; Chen et al., 2025) exhibit *quartic* dependence on the condition number, in contrast to the *quadratic* scaling achieved in the discounted setting (Bhandari et al., 2018). Alternatively, the result from (Li et al., 2024) does achieve quadratic scaling, but it relies on the assumption $\Phi\theta \neq ce$ which prevents it from being applied to the tabular case.

The distinction between convergence to a set and converge to a point is not merely cosmetic. For policy-gradient or actor-critic applications, convergence to the correct equivalence class may be sufficient at an abstract level. In actual implementations, however, the solution set often contains flat directions, such as additive constant directions in average-reward value estimation. An algorithm my therefore drive its distance to the solution set to zero while continuing to drift

along the set itself. This can make the raw iterates very large even when the value error modulo constants is small. Such unbounded drift complicates reproducibility, undermines simple stopping rules based on $\|\theta_{t+1} - \theta_t\|$, and can amplify round-off errors in low-precision arithmetic. This provides an additional practical motivation for designing algorithms whose iterates converge to a unique, sample-independent point.

In this work, we take a complementary approach: observing that the average-reward value function must satisfy a certain steady-state constraint, we formulate the solution to the Bellman equation as a constrained optimization problem. To solve this problem, we propose a new algorithm that leverages sampling from two independent Markov chains in each iteration. We provide a finite-sample analysis for this method using a relatively new technique known as "gradient splitting" (Liu & Olshevsky, 2021). Finally, building upon ideas from the Gradient TD (GTD) (Sutton et al., 2008), we give a version of our algorithm which only uses a single Markov chain.

Our work improves upon the state of the art by simultaneously having all of the following features.

- **Provably unique, sample-independent fixed point.** Our analysis shows that the iterate sequence $\{\theta_t\}$ converges almost surely to a single, deterministic solution $\theta^\star$ that does not depend on the random trajectory or on initialization.

- **Tabular + Linear Function Approximation**: Our analysis is applicable to both the tabular and linear function approximation cases. In particular, we do not assume $e \notin \text{span}(\Phi)$.

- **Dependence on Dimensionality**: Our convergence bound does not have any explicit factors of $d$, the dimension of $\theta$. While all algorithms have terms like $\|\theta^*\|$ that might implicitly scale with dimension, our algorithm has no terms scaling with $d$ in addition to those.

- **Last Iterate:** our results are based on the last iterate rather than an iterate averaging, matching the corresponding results in the discounted case.

- **Scaling with Condition Number**: In Zhang et al. (2021b) and Haque & Maguluri (2024), the dependence of the convergence time on the condition number is quartic as $O\left(\eta_2^{-4}\right)$, where $\eta_2$ is defined as

$$\eta_2 = \min_{\|x\|=1, x^\top e=0} \|\Phi x\|^2_{\text{Dir}}, \qquad (1)$$

where $\|\cdot\|$ is the Euclidean norm and $\|\cdot\|_{\text{Dir}}$ denotes the Dirichlet seminorm, formally defined later in Section

*Table 1.* Comparison with previous work. We highlight several key distinctions between our work and prior research:

(1) In the table below, **Linear Only** refers to the assumption that $e \notin \text{span}(\Phi)$, which rules out the tabular case.

(2) A complicating factor in comparing results is that different papers tend to have slightly different notions of condition numbers; this is why the table contains $\eta_1, \eta_2, \eta_3$ (see definitions in Eq. (3) to Eq. (2)). A detailed discussion and precise definitions can be found in Appendix B. We remark that our condition number $\eta_1$ is at least as good as the widely used $\eta_3$, i.e., $\eta_1 \geq \Omega(\eta_3)$, which means bounds based on $\eta_1^{-1}$ are at least as good as bounds based on $\eta_3^{-1}$; and if we assume the stationary probability $\mu$ is not too far from uniform, we also have $\eta_1 \geq \Omega(\eta_2)$. See Appendix B for details.

(3) We consider a convergence time to be independent of the dimension $d$ if its dependence on $d$ appears only through $\|\theta_0\|$ or $\|\theta^*\|$.

(4) We make the standard assumption that $\|\phi(s)\| \leq 1$ for all states $s$, which can be achieved by rescaling. If we instead make the assumption that every entry of $\phi(s)$ is $O(1)$ (so that $\|\phi(s)\| = O(\sqrt{d})$), then our scaling with dimension would be $O(d)$. This modified assumption matches more closely the assumption made in (Haque & Maguluri, 2024), which, unlike this work, analyzes the infinite-dimensional case. Our results are an improvement compared to the larger $\Omega(d^2)$ scaling in that work.

(5) (Li et al., 2024) includes a scaling with approximation error ($\mathbb{E}\|W^* - \Phi\theta^*\|_D^2$) that no other paper has. In addition, we also note that they use a variance reduction method, not an analogue of plain TD; we do not implement variance reduction and our scaling with condition number is similar to their method.

(6) In (Li et al., 2024), the condition number is actually defined as $\min_{y \neq e, \|y\|_D = 1} y^\top D(I - P)y$. However, our analysis in Appendix B suggests that $\eta_3$ is actually greater than their condition number, offering an optimistic approximation of their sample complexity.

(7) In (Kim et al., 2025), the sample complexity is actually $\tilde{O}(\epsilon^{-1} R_{\text{proj}}^2 \eta_2^{-3})$, where $R_{\text{proj}}$ is the projection radius. However, the authors did not provide the choice of this radius. Our analysis in Appendix B shows that $R_{\text{proj}}^2$ is actually $O(1/\eta_3)$.

(8) In the discounted case, $\eta_{\text{discounted}}$ can be defined as $(1 - \gamma)\sigma_{\min}(\Phi^T D\Phi)$ where $\gamma$ is the discount factor (Bhandari et al., 2018)

| REFERENCE | SETTING | CONVERGES TO A SAMPLE INDEPENDENT POINT | SAMPLE COMPLEXITY | SCALING WITH $d$ | ITERATE | METHOD |
|---|---|---|---|---|---|---|
| (Tsitsiklis & Van Roy, 1999) | LINEAR ONLY | YES | $\times$ | $\times$ | LAST | COUPLED SA |
| (Zhang et al., 2021b) | TABULAR & LINEAR | No | $\tilde{O}(\epsilon^{-1}\eta_2^{-4})$ | No | LAST | COUPLED SA |
| (Haque & Maguluri, 2024) | TABULAR & LINEAR | YES | $\tilde{O}(\epsilon^{-1}\eta_2^{-4})$ | YES | LAST | COUPLED SA |
| (Kim et al., 2025) | TABULAR & LINEAR | No | $\tilde{O}(\epsilon^{-1}\eta_2^{-3}\eta_3^{-1})$ | No | LAST | COUPLED SA |
| (Chen et al., 2025) | TABULAR & LINEAR | No | $\tilde{O}(\epsilon^{-1}\eta_2^{-4})$ | No | LAST | COUPLED SA |
| (Li et al., 2024) | LINEAR ONLY | YES | $\tilde{O}(\epsilon^{-1}\eta_3^{-2})$ | No | AVERAGE | VARIANCE REDUCTION |
| OUR DOUBLE-CHAIN ALGORITHM | TABULAR & LINEAR | YES | $\tilde{O}(\epsilon^{-1}\eta_1^{-2})$ | No | LAST | COUPLED SA |
| DISCOUNTED CASE (E.G., (Bhandari et al., 2018)) | TABULAR & LINEAR | YES | $\tilde{O}(\epsilon^{-1}\eta_{\text{discounted}}^{-2})$ | No | LAST | SA |

2.3. Similarly, in Kim et al. (2025), the dependence of the convergence time on the condition number is also quartic as $O\left(\eta_2^{-3}\eta_3^{-1}\right)$, where $\eta_3$ is defined as

$$\eta_3 = \left( \min_{\|x\|=1} x^\top \Phi^\top D\Phi x \right) \cdot \left( \min_{\langle y,e \rangle_D = 0, \|y\|_D = 1} y^\top D(I - P)y \right), \quad (2)$$

where $D$ is the diagonal matrix with the stationary distribution $\mu$ of the policy on the diagonal. These contrast unfavorably with the corresponding results in the discounted case, which scale with the square rather than fourth power of the condition number (Bhandari et al., 2018). That being said, the definition of condition number is different within the discounted case, and it may be that a slightly different notion of condition number is needed for the average-reward setting.

Indeed, that is just what we show: for our algorithm convergence time scales quadratically as $O\left(1/\eta_1^2\right)$, where $\eta_1$ is defined as

$$\eta_1 = \min_{x: \|x\|=1} \|\Phi x\|_{\text{Dir}}^2 + (\mu^\top \Phi x)^2, \quad (3)$$

and $\mu$ is the stationary distribution of the policy.

- **No variance reduction techniques needed.** While variance-reduction schemes are well-known to improve sample complexity, they typically require multilevel structure (e.g., nested loops, periodic full-batch or long-trajectory reference estimates, and additional bookkeeping) and therefore differ substantially from the "plain TD" template. In contrast, our quadratic condition-number scaling is achieved with a single-timescale, simple stochastic-approximation update without using any variance reduction techniques, matching the algorithmic simplicity of discounted TD.

## 2. Preliminaries

This section introduces the necessary background on average-reward reinforcement learning to support the algorithm design and convergence analysis of TD learning presented in the subsequent sections.

## 2.1. Markov Decision Processes (MDP)

We consider an MDP defined by the tuple $(S, A, P_{\text{env}}, r)$, where (i) $S$ is the finite state space, (ii) $A$ is the finite action space, (iii) $P_{\text{env}} = (P_{\text{env}}(s' \mid s, a))_{s, s' \in S, a \in A}$ is the transition probability kernel, and (iv) $r : S \times A \to \mathbb{R}$ is the reward function. Let $r_{\max} := \max_{s \in S, a \in A} |r(s, a)|$, which is finite since the state-action space is finite.

A policy $\pi : S \times A \to \mathbb{R}$ is a function where $\pi(a \mid s)$ represents the probability of the agent taking action $a$ in state $s$. Throughout this paper, we focus exclusively on the policy evaluation problem and therefore assume the policy $\pi$ to be fixed and known. Under this fixed policy, we define the induced transition matrix $P$ as $P = (P(s' \mid s))_{s, s' \in S}$, where $P(s' \mid s) = \sum_{a \in A} P_{\text{env}}(s' \mid s, a) \pi(a \mid s)$.

We make the following assumption regarding the policy $\pi$, which is standard in TD learning (Liu & Olshevsky, 2021; Bhandari et al., 2018).

**Assumption 2.1.** The Markov chain with transition matrix $P$ is irreducible and aperiodic.

Under the above assumption, the Markov chain with transition matrix $P$ has a unique stationary distribution, denoted by $\mu$, which satisfies $\mu_{\min} := \min_s \mu(s) > 0$ (Levin & Peres, 2017). Moreover, according to Theorem 4.9 in (Levin & Peres, 2017), there exist constants $C > 1$ and $\beta \in [0, 1)$ such that

$$\|p_\tau(\cdot|s) - \mu\|_1 \le C\beta^\tau, \quad \forall \tau \ge 0, s \in S, \qquad (4)$$

where $p_\tau$ is the probability distribution of the state of this Markov chain after $\tau$ steps starting at $s$.

## 2.2. The Long-Term Average Reward

We now discuss value functions within the average-reward framework, highlighting their role in policy evaluation. Let $r_s = \sum_a \pi(a \mid s) r(s, a)$ denote the expected reward in state $s$ under policy $\pi$. We also define $p_{s_0 s}^k$ to be the probability that the agent is at state $s$ after $k$ steps starting from $s_0$. Then the value function $v_{s_0}(t)$ reflects the expected cumulative reward starting from state $s_0$ after $t$ transitions:

$$v_{s_0}(t) = r_{s_0} + \sum_s p_{s_0 s} r_s + \cdots + \sum_s p_{s_0 s}^{t-1} r_s.$$

Defining $V_t = [v_s(t)]_{s \in S}$ as the vector stacking up the value function and $R = [r_s]_{s \in S}$ as the vector stacking up the expected rewards, the above equation can be compactly written as $V_t = \sum_{k=0}^{t-1} P^k R$. Under Assumption 2.1, we have $\lim_{n \to \infty} P^n = e\mu^\top$ (Levin & Peres, 2017). Let $g = \mu^\top R$ be the steady-state reward per unit of time. Then, the relative value function, denoted by $W^*$, is defined as

$$W^* = \lim_{t \to \infty} V_t - tge = \lim_{t \to \infty} \sum_{k=0}^{t-1} (P^k - e\mu^\top) R. \qquad (5)$$

Intuitively, the relative value function $W^*$ quantifies the long-term expected cumulative reward differences across states relative to the steady-state reward. It is a central quantity for policy evaluation in the average-reward setting, as it effectively centers the rewards to focus purely on differences due to transient dynamics.

## 2.3. Useful Norms

In this subsection, we introduce several useful norms which will play an important role in our analysis. The so-called $D$-norm and Dirichlet semi-norm have been previously shown to be very useful in TD-like analysis (Ollivier, 2018; Liu & Olshevsky, 2021). Given a vector $f$ with the same number of entries as the number of states in the MDP, its $D$-norm is defined as

$$\|f\|_D^2 = \langle f, f \rangle_D = \sum_s \mu(s) f(s)^2 \qquad (6)$$

and its Dirichlet semi-norm is defined as

$$\|f\|_{\text{Dir}}^2 = \frac{1}{2} \sum_{s, s'} \mu(s) P(s'|s) (f(s) - f(s'))^2. \qquad (7)$$

Intuitively, the Dirichlet seminorm measures the difference between $f$ and the all-ones vector $e$, but in a way that is adapted to the Markov chain with transition matrix $P$. Finally, throughout the paper, we will use $\|\cdot\|$ to denote the standard Euclidean norm.

## 2.4. Markov Noise

Let $s_t$ denote the state at time step $t$. Following standard practice, we consider two distinct sampling scenarios in this paper: (1) *i.i.d. sampling*, where each state $s_t$ is independently drawn from the stationary distribution $\mu$; and (2) *Markov sampling*, where the Markov chain starts at $s_0$ and evolves according to the policy. Under Markov sampling, the states $s_t$ remain marginally distributed according to $\mu$, but exhibit temporal correlation across time steps.

## 2.5. Linear Function Approximation

In practical reinforcement learning applications, the state space $S$ is often extremely large, making it impractical to maintain a vector whose dimension scales with the number of states. To address this challenge, it is common to incorporate function approximation, in particular, a linear function approximator of the form $W = \Phi\theta$, where $\Phi \in \mathbb{R}^{n \times d}$ is the feature matrix and $\theta \in \mathbb{R}^d$ is the parameter vector. Additionally, denote the $s$-th row of the feature matrix by $\phi(s)^\top$. We assume, without loss of generality, that (i) the features are normalized so that $\max_s \|\phi(s)\| \le 1$, and (ii) the columns of $\Phi$ are linearly independent.

We further define

$$\eta = \min_{x:\|x\|=1} \|\Phi x\|^2_{\text{Dir}} + (\mu^\top \Phi x)^2, \quad \text{where } x \in \mathbb{R}^d. \quad (8)$$

Intuitively, $\eta$ measures how close to zero $\Phi x$ can get: the first term measures the distance between $\Phi x$ and the all-ones vector, whereas the second term measures the (squared) distance between the weighted average $\mu^\top \Phi x$ and zero. In particular, under the assumption that the columns of $\Phi$ are linearly independent, we immediately have $\eta > 0$. The quantity $\eta$ will act as a condition number in our algorithms for average reward TD.

## 3. Algorithms

We now introduce the two algorithms studied in this paper. The first, called the **double-chain algorithm**, uses two independent Markov chains. The second, the **single-chain algorithm**, uses only one.

### 3.1. Double-Chain Algorithm: Motivation and Derivation

Recall that in the discounted setting, TD-learning is designed to solve the projected Bellman equation (Tsitsiklis & Van Roy, 1996). To motivate our analysis, we next introduce a natural analogue of the projected Bellman equation in the average-reward setting.

Our goal is to compute $W^*$ as defined in Eq.(5). It is well known (Gallager, 1997) that the relative value function $W^*$ satisfies two properties:

$$W^* + ge = PW^* + R \quad \text{and} \quad \mu^\top W^* = 0.$$

Define $\Pi = I - e\mu^\top$ as a projection onto the subspace $\{x \mid \langle x, e \rangle_D = 0\}$ in the inner product $\langle \cdot, \cdot \rangle_D$ and the Bellman operator $T_\pi$ such that $T_\pi W = R + PW$.

We can write the two properties into an equivalent way:

$$W^* = \Pi T_\pi W^*. \quad (9)$$

It is known that $W^*$ is the unique solution to this equation (Gallager, 1997). Since the matrix $D = \text{diag}(\mu)$ is invertible, the previous equation is equivalent to

$$D\left(\Pi(R + PW^*) - W^*\right) = 0,$$

which can be further written as

$$D\left(R + PW^* - W^*\right) - \mu\mu^\top(R + W^*) = 0 \quad (10)$$

using the explicit definition of $\Pi$.

A natural approach to solve $W^*$ from Eq.(10) is to recursively perform the update

$$W_{t+1} = W_t + \alpha_t \left(D\left(R + PW_t - W_t\right) - \mu\mu^\top(R + W_t)\right). \quad (11)$$

Although the above iterative algorithm seems promising, it cannot be implemented directly, since the transition matrix $P$ and the reward function $R$ are unknown. In the remainder of this section, we develop a data-driven stochastic version of the algorithm presented in Eq. (11). Before delving into the details, we first introduce some notation.

We use $1(s = s_0) \in \mathbb{R}^{|S|}$ to denote the vector whose entries are 0 except a 1 at position $s = s_0$. We also use $\mathbb{E}_\mu$ as expectation assuming that the state $s_t$ is drawn from stationary distribution $\mu$ while $s_{t+1}$ is still drawn according to the MDP with action taken according to policy $\pi$.

With this notation in place, we now describe the intuition behind the equations that we will write down. Keeping in mind that our goal is to provide a stochastic version of Eq. (11), the straightforward approach is to replace the unknown transition matrix $P$ with something depending on samples that has expectation $P$. This works, but it is the second term in Eq. (11) that causes some trouble: it is surprisingly not straightforward to find a quantity such that its expectation is $\mu\mu^\top(R + W_t)$.

Indeed, to form a stochastic estimator for $\mu\mu^\top(R+W_t)$, observe that while $\mathbb{E}_\mu[1(s = s_t)] = \mu$ and $\mathbb{E}_\mu[r_{s_t}+W_t(s_t)] = \mu^\top(R + W_t)$, we cannot multiply these two estimators to obtain the result we want because $\mathbb{E}[XY] \neq \mathbb{E}[X]\mathbb{E}[Y]$ for random variables $X$ and $Y$ if they are not independent. This is known as the double sampling issue (Sutton et al., 2008). A natural and simple way to solve this issue is to sample two independent Markov Chains and base the two estimates on independent samples.

Denoting the state of these two chains by $\{s_t\}$ and $\{\hat{s}_t\}$, respectively, we therefore consider the following update:

$$W_{t+1} = W_t + \alpha_t\left(f(s_t, \hat{s}_t, W_t) + g(s_t, s'_t, W_t)\right), \quad (12)$$

where

$$f(s_t, \hat{s}_t, W_t)[s] = -1(s = \hat{s}_t)(r_{s_t} + W_t(s_t)),$$
$$g(s_t, s'_t, W_t)[s] = 1(s = s_t)\left(r_{s_t} + W_t(s'_t) - W_t(s_t)\right),$$

It is then indeed immediate that

$$\mathbb{E}_\mu f(s_t, \hat{s}_t, W_t) = -\mu\mu^\top(R + W_t)$$
$$\mathbb{E}_\mu g(s_t, s'_t, W_t) = D(R + PW_t - W_t),$$

and therefore Eq. (12) is a stochastic version of Eq. (11).

With linear function approximation, the natural generalization of Eq. (9) becomes

$$\Phi\theta^* = \Pi_D \Pi T_\pi \Phi\theta^* \quad (13)$$

where $\Pi_D = \Phi(\Phi^\top D\Phi)^{-1}\Phi^\top D$ is the projection onto the subspace spanned by the columns of $\Phi$ in $D$-inner product. Compared to Eq. (9), this equation uses $\Phi\theta$ as an approximation and adds a projection to the column space of $\Phi$. We

rewrite the above equation in the following form, which will be easier to implement:

$$\Phi^\top D(I - \Pi P)\Phi\theta^* = \Phi^\top D\Pi R. \qquad (14)$$

The following lemma establishes the existence and uniqueness of $\theta^*$ whose proof can be found in Appendix A.

**Lemma 3.1.** *The solution to the linear system $\Phi^\top D(I - \Pi P)\Phi\theta = \Phi^\top D\Pi R$ exists and is unique.*

Based on this equation, a natural generalization from the tabular to the linear approximation case is therefore

$$\theta_{t+1} = \theta_t + \alpha_t \left( f(s_t, \hat{s}_t, \theta_t) + g(s_t, s'_t, \theta_t) \right), \qquad (15)$$

where

$$\begin{aligned}
f(s_t, \hat{s}_t, \theta_t) &= - (r_{s_t} + \phi(s_t)^\top \theta_t)\phi(\hat{s}_t) \\
g(s_t, s'_t, \theta_t) &= \left( r_{s_t} + \phi(s'_t)^\top \theta_t - \phi(s_t)^\top \theta_t \right) \phi(s_t).
\end{aligned} \qquad (16)$$

### 3.2. Single-Chain Algorithm

It is natural to wonder whether we can perform the update using only a single Markov chain. Inspired by the GTD method (Sutton et al., 2008), we propose a solution that does so.

Our single-chain algorithm is based on the following observation. Our two-chain algorithm uses the term $\phi(\hat{s}_t)$ – but what if, instead, we replace that by an estimate of the expectation $E[\phi(\hat{s}_t)]$? Because $s_t$ is sampled from $\mu$, this expectation equals $\Phi^\top \mu$. We will therefore introduce a new variable $w_t$ which will converge to $\Phi^\top \mu$ and use it in place of $\phi(\hat{s}_t)$.

Our algorithm is thus as follows:

$$\begin{aligned}
w_{t+1} &= \mathbf{Proj}_{R_w}\{w_t + \beta_t f(s_t, w_t)\} \\
\theta_{t+1} &= \mathbf{Proj}_{R_\theta}\{\theta_t + \alpha_t g(s_t, s'_t, w_t, \theta_t)\},
\end{aligned} \qquad (17)$$

where

$$\begin{aligned}
f(s_t, w_t) =& \phi(s_t) - w_t, \\
g(s_t, s'_t, w_t, \theta_t) =& (r_{s_t} + \phi(s'_t)^\top \theta_t - \phi(s_t)^\top \theta_t)\phi(s_t) - (r_{s_t} \\
&+ \phi(s_t)^\top \theta_t)w_t.
\end{aligned} \qquad (18)$$

Note that the first line clearly drives $w_t$ to $\Phi^\top \mu$ while the second line is identical to the double-chain method except $\phi(\hat{s}_t)$ has been replaced by $w_t$.

To see where $\{w_t\}$ and $\{\theta_t\}$ will converge to in this case, we notice that, under i.i.d. sampling,

$$\begin{aligned}
\mathbb{E}_{s_t \sim \mu} f(s_t, w_t) =& \Phi^\top \mu - w_t \\
\mathbb{E}_{s_t \sim \mu} g(s_t, s'_t, w_t, \theta_t) =& \Phi^\top D(R + P\Phi\theta_t - \Phi\theta_t) \\
&- \mu^\top(R + \Phi\theta_t)w_t.
\end{aligned}$$

Therefore, we can define $w^*$ and $\theta^*$ such that

$$\begin{aligned}
&\Phi^\top \mu - w^* = 0 \\
&\Phi^\top D(R + P\Phi\theta^* - \Phi\theta^*) - \mu^\top(R + \Phi\theta_t)w^* = 0.
\end{aligned}$$

The intuition behind this method is that while $w_t$ is derived from the same trajectory as $s_t$, it is an average over past features. We can therefore expect that it will be essentially de-correlated from the instantaneous value $s_t$, so we will be able to estimate $E[XY] \approx E[X]E[Y]$ up to some error for the product of $X = w_t$ and $Y = r_{s_t} + \phi(s_t)^\top \theta_t$. Naturally, this will come at the cost of an increased convergence time because of the additional error incurred.

## 4. Main Results

We now present our main results. We will consider both i.i.d. and Markov sampling, as well as both our double-chain and single-chain algorithms.

### 4.1. Convergence Results for the Double-Chain Method

Our first result assumes that the two Markov chains $\{s_t\}$ and $\{\hat{s}_t\}$ are sampled i.i.d. from the stationary distribution $\mu$. We first consider the case where the stepsizes are constant $\alpha_t = \alpha$. We define $\tau_{\mathrm{mix}}$ as the mixing time $\tau_{\mathrm{mix}} = \tau_{\mathrm{mix}}(\alpha)$. We also denote $\delta_t = \theta_t - \theta^*$. Our first theorem considers the double-chain method.

**Theorem 4.1** (Double-chain, i.i.d. sampling). *Suppose Assumption 2.1 holds. Consider the double-chain algorithm (15) with i.i.d. sampling and constant stepsize $\alpha \leq \eta/18$. Then*

$$\mathbb{E}\|\theta_T - \theta^*\|^2 \leq e^{-\alpha\eta T}\|\theta_0 - \theta^*\|^2 + O\left(\frac{\alpha(r_{\max} + \|\theta^*\|)^2}{\eta}\right).$$

*In particular, choosing $\alpha = \tilde{\Theta}(1/(\eta T))$ yields a convergence rate of $\tilde{O}(1/T)$ with sample complexity $\tilde{O}(\epsilon^{-1}\eta^{-2})$.*

Our next theorem generalizes this result to Markov sampling, i.e., when $s_t, \hat{s}_t$ are sampled from two independent Markov chains. This requires further information on the mixing time of the Markov Chain as mentioned in Section 2.4. We still choose a constant stepsize. We will require the constants

$$\begin{aligned}
B &= 2\|\theta_0 - \theta^*\| + r_{\max} + \|\theta^*\|, \\
G &= 42B^2 + 30(r_{\max} + \|\theta^*\|)^2.
\end{aligned}$$

**Theorem 4.2** (Double-chain, Markov sampling, constant stepsize). *Suppose Assumption 2.1 holds. Consider the double-chain algorithm (15) with Markov sampling and constant stepsize*

$$\alpha \leq \frac{\eta B^2}{(3\tau_{\mathrm{mix}} + 1)G}.$$

*Then for all $T \geq \tau_{\mathrm{mix}}$,*

$$\mathbb{E}\|\theta_T - \theta^*\|^2 \leq e^{-2\alpha\eta(T - \tau_{\mathrm{mix}})}B^2 + \frac{\alpha G(3\tau_{\mathrm{mix}} + 1)}{2\eta}.$$

*Choosing $\alpha = \tilde{\Theta}(1/(\eta T))$ yields a convergence rate of $\tilde{O}(1/T)$ with sample complexity $\tilde{O}(\epsilon^{-1}\eta^{-2})$.*

The rates of both of these theorems match the state-of-the-art for TD learning in the discounted case (Bhandari et al., 2018) in terms of the scaling with the various parameters.

A similar result can be obtained with a decaying stepsize $\alpha_t = a/(t + c_0)^\xi$. In this case, the mixing time is defined as $\tau_{\text{mix}} = \tau_{\text{mix}}(\alpha_T)$. The next theorem formally handles this case.

**Theorem 4.3** (Double-chain, Markov sampling, decaying stepsize). *Suppose Assumption 2.1 holds. Consider the double-chain algorithm (15) with Markov sampling and stepsize $\alpha_t = a/(t + c_0)^\xi$ for $\xi \in (0, 1]$, $a > 0$, and $c_0$ sufficiently large. Then for all $T \geq \tau_{\text{mix}}$:*

1. *If $\xi = 1$, then*

$$
\begin{aligned}
\mathbb{E}\|\theta_T - \theta^*\|^2 &\leq O(1) \cdot \left(\frac{\tau_{\text{mix}} + c_0}{T + c_0}\right)^{a\eta} \\
&+ \tilde{O}\left(\frac{a^2}{(T + c_0)^{\min\{1, a\eta\}}}\right).
\end{aligned}
$$

2. *If $\xi \in (0, 1)$, let $\Delta_T = (T + c_0)^{1-\xi} - (\tau_{\text{mix}} + c_0)^{1-\xi}$ then*

$$
\begin{aligned}
\mathbb{E}\|\theta_T - \theta^*\|^2 &\leq O\left(\exp\left(-\frac{\eta a}{1 - \xi}\Delta_T\right)\right) \\
&+ O\left(\frac{a}{\eta(T + c_0)^\xi}\right).
\end{aligned}
$$

*Choosing $a = \Theta(1/\eta)$ yields sample complexity $\tilde{O}(\epsilon^{-1/\xi}\eta^{-2/\xi})$ which reduces to $\tilde{O}(\epsilon^{-1}\eta^{-2})$ when $\xi = 1$. The $O(1)$ terms depend polynomially on $\|\theta_0 - \theta^*\|$, $r_{\text{max}}$, and $\|\theta^*\|$.*

This is once again consistent with the state-of-the-art results in the discounted setting (Bhandari et al., 2018), where an additional $O(1/\eta)$ factor is introduced when the distance is measured in the parameter space.

### 4.2. Convergence Results for the Single-Chain Method

For the single-chain algorithm given by Eq. (17), we can also establish similar results, albeit with worse scaling with respect to the condition number. We denote $\delta_t^\theta = \theta_t - \theta^*$ and $\delta_t^w = w_t - w^*$. We define $\tau_{\text{mix}}$ as the mixing time $\tau_{\text{mix}} = \tau_{\text{mix}}(\min\{\alpha, \beta\})$.

**Theorem 4.4** (Single-chain, constant stepsize, Markov sampling). *Consider the single-chain algorithm in Eq. (17) with Markov sampling. Assume constant stepsizes $\alpha_t \equiv \alpha > 0$, $\beta_t \equiv \beta > 0$, and let $\rho_0 := \beta/\alpha \leq 1$. Let*

$\tau_{\text{mix}} := \tau_{\text{mix}}(\min\{\alpha, \beta\})$ *denote the mixing time at accuracy level* $\min\{\alpha, \beta\}$. *Let $0 < \alpha < \frac{1}{2\zeta}$ and let $\lambda > 0$ satisfy*

$$
0 < \lambda^2 < \frac{2\eta}{r_{\text{max}} + 2R_\theta},
$$

$$
\zeta := \eta - \frac{\lambda^2(r_{\text{max}} + 2R_\theta)}{2} > 0,
$$

*and define*

$$
\kappa := 1 - 2\alpha\zeta \in (0, 1), \qquad G_1 := e^{-2\rho_0\alpha} \in (0, 1),
$$

*where $\rho := \max\{\kappa, G_1\} \in (0, 1)$. Then for all $t \geq \tau_{\text{mix}}$,*

$$
\begin{aligned}
\mathbb{E}\|\theta_t - \theta^*\|^2 &\leq \frac{4\alpha(r_{\text{max}} + 2R_\theta)(t - \tau_{\text{mix}})\rho^{t-\tau_{\text{mix}}}}{\lambda^2} \\
&+ 4R_\theta^2\kappa^{t-\tau_{\text{mix}}} + \frac{\alpha G_{\text{const}}}{2\zeta},
\end{aligned}
$$

*and where*

$$
\begin{aligned}
G_{\text{const}} := &\frac{(16\tau_{\text{mix}} + 6)(r_{\text{max}} + 2R_\theta)}{\lambda^2} \\
&+ (88\tau_{\text{mix}} + 4)(r_{\text{max}} + 3R_\theta)^2.
\end{aligned}
$$

**Sample Complexity.** As discussed in Appendix A.3, the projection radii can be chosen so that $R_\theta = O(\eta'^{-1/2})$ and $R_w = O(1)$, where $\eta' = \lambda_{\min}(\Phi^\top D\Phi)$. Moreover, Appendix E shows one may select $\lambda^2 = \Theta(\eta'^{1/2}\eta)$, which implies $\zeta = \Theta(\eta)$ (e.g., by taking $\lambda^2$ a fixed fraction of $2\eta/(r_{\text{max}} + 2R_\theta)$). With these choices, the dominant part of $G_{\text{const}}$ comes from the $(r_{\text{max}} + 2R_\theta)/\lambda^2$ term and scales as $G_{\text{const}} = \tilde{O}(\tau_{\text{mix}}/(\eta'\eta))$ (up to polynomial factors in $r_{\text{max}}$ and $R_\theta$), so the steady-state error term satisfies

$$
\frac{\alpha G_{\text{const}}}{\zeta} = \tilde{O}\left(\frac{\alpha \tau_{\text{mix}}}{\eta'\eta^2}\right).
$$

Thus, choosing $\alpha = \tilde{\Theta}(1/(\eta T))$ yields a $\tilde{O}(1/T)$ rate (as in the two-chain case), but the overall condition-number dependence is worse than quadratic: plugging $\alpha = \tilde{\Theta}(1/(\eta T))$ into the steady-state term gives $\tilde{O}(\tau_{\text{mix}}/(\eta'\eta^3 T))$, which becomes quartic $\tilde{O}(1/(\eta^4 T))$ in the common regime where $\eta' = \Theta(\eta)$ (the precise relation between $\eta'$ and $\eta$ is discussed in Appendix B).

Finally, we can also give a similar theorem for the single-chain method with decaying step-size

$$
\alpha_t = \frac{a}{(t + c_0)^\xi},
$$

which parallels Theorem 4.3 for the double-chain algorithm but with the $\eta^{-2}$ scaling replaced with $\eta^{-4}$. For reasons of space, **we state this in the appendix as Theorem F.1**.

**Remark:** Some previous works add an additional $(r_t - g)^2$ to the error measure, e.g., (Zhang et al., 2021b; Haque &

Maguluri, 2024). We observe that using $r_t = (\sum_{i=0}^{t-1} r_{S_i})/t$ as a stochastic estimate of $g$ allows us to obtain

$$\mathbb{E}\left[(r_t - g)^2\right] = O(1/T). \tag{19}$$

Thus, this quantity could easily be estimated separately without affecting the results of this section. A detailed discussion is in Appendix B.

### 4.3. Proof Idea

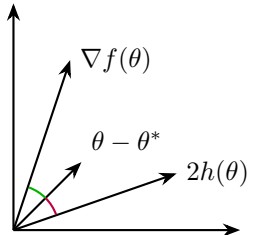

Our main observation is that the methods we propose can be analyzed in much the same way as standard *Stochastic Gradient Descent (SGD)*, which is usually more tractable than stochastic approximation-based methods. The critical tool enabling this perspective is the notion of a *gradient splitting*, introduced by (Liu & Olshevsky, 2021).

*Figure 1.* Illustration of key property of the gradient splitting. The splitting vector $h(\theta)$ has the same inner product (up to a factor of 2) as the true gradient with the vector $\theta - \theta^*$.

To illustrate the idea, let us consider a convex quadratic function $f(\theta) = (\theta - \theta^*)^T A (\theta - \theta^*)$, where $A$ is symmetric and positive definite; and let us also consider a linear function $h(\theta) = B(\theta - \theta^*)$. We will say $h(\theta)$ is a gradient splitting of $f(\theta)$ if $B + B^T = 2A$. In other words, each $(i,j)$-entry of $A$ can be split between the $(i,j)$ and $(j,i)$-entries of the (generally non-symmetric) matrix $B$. This decomposition is not unique, since there are many ways to split the entries of $A$.

An immediate implication of this definition is that for any $\theta$, $(\theta^* - \theta)^T \nabla f(\theta) = 2(\theta^* - \theta)^T h(\theta)$. Hence, updating in the direction $-h(\theta)$ behaves just like an update in the direction of the true gradient $-\nabla f(\theta)$. An illustration is given in Figure 1. One can further conclude

$$\begin{aligned} 2(\theta - \theta^*)^T h(\theta) &= (\theta^* - \theta)^T \nabla f(\theta) \\ &= (\theta^* - \theta)^T (\nabla f(\theta) - \nabla f(\theta^*)) \\ &= 2f(\theta), \end{aligned}$$

where the last equality holds because $f(\theta)$ is a quadratic function.

Concretely, the update rule

$$\theta_{t+1} = \theta_t - \alpha_t \left[h(\theta_t) + w_t\right], \tag{20}$$

where $w_t$ is zero-mean i.i.d. noise, exhibits convergence properties analogous to classical SGD. In fact, using a Tay-

lor expansion,

$$\begin{aligned} E\left[\|\theta_{t+1} - \theta^*\|_2^2 \big| \theta_t\right] = \\ \|\theta_t - \theta^*\|_2^2 - \alpha_t \nabla f(\theta_t)^T (\theta_t - \theta^*) + O(\alpha_t^2). \end{aligned}$$

Notice that the middle term is precisely what one would get with a standard gradient step, even though (20) does not explicitly use $\nabla f(\theta_t)$. Thus, standard SGD analysis applies, with adjustments for the different higher-order $O(\alpha_t^2)$ terms.

Building on these ideas, our contribution in this paper is to show that an algorithm for the average reward case can also be written as a gradient splitting. Specifically, consider Eq. (15), where the expected update direction is $E_\mu\left[f(s_t, \hat{s}_t) + g(s_t, s_t')\right]$, with notation as in Eq. (15). We can rewrite this expected update as $\Phi^T D\left(R + P\Phi\theta - \Phi\theta\right) - \Phi^T \mu\mu^T\left(R + \Phi\theta\right)$; our key technical observation is to show that it serves as a gradient splitting for the composite function

$$\|\Phi(\theta - \theta^*)\|_{\text{Dir}}^2 + \left(\mu^T \Phi(\theta - \theta^*)\right)^2.$$

Once this gradient-splitting viewpoint is in place, we can leverage the standard SGD descent recursion, but closing the argument requires new bounds that control the discrepancy between our stochastic update and the true gradient—specifically, we must show the resulting $O\left(\alpha_t^2\right)$ and bias terms remain uniformly small/summable under the chosen stepsizes.

## 5. Numerical Results

We compare our algorithms with prior work on fifteen tabular MDPs from OpenAI Gym and MO-Gymnasium. We focus on tabular environments because the exact solution can be computed, allowing us to directly quantify how accurately each method approximates the true solution. This enables a fair comparison between algorithms that converge to a single point and those that converge only to a set, since we evaluate all methods using their approximation error relative to the true solution rather than properties of their iterates.

Across all fifteen environments, we evaluate both our Double-Chain and Single-Chain algorithms, together with the baselines listed in Table 1. Overall, we observe that our methods outperform the prior literature in most settings. Additional experimental details are provided in Appendix G.

## 6. Conclusion

The theoretical analysis of average reward TD has traditionally lagged behind the discounted setting due to mathematical difficulties, notably the non-contractive nature of the Bellman operator. This disparity manifested in prior works through various limitations: assumptions incompatible with

the tabular case, convergence guarantees only to sets or sample-dependent points, explicit dimension scaling, and slower (quartic) convergence rates with respect to condition numbers – which, unlike in the discounted case, could only be removed with more intricate algorithms like variance reduction.

In this work, we employed the gradient splitting technique to provide a finite-sample analysis that closes the gap with discounted TD theory. Our methods guarantee convergence to a unique, well-defined solution for both tabular and linear approximation cases. Furthermore, the convergence bounds are dimension-free (in the standard sense) and exhibit quadratic scaling with the relevant condition number, mirroring the performance characteristics known for discounted TD. This contribution removes the persistent caveats associated with average reward TD analysis.

## Impact Statement

This paper presents work whose goal is to advance the field of machine learning. There are many potential societal consequences of our work, none of which we feel must be specifically highlighted here.

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

# A. Bellman Operator and Projection

In this section, we establish the contractivity of $\Pi T_\pi$ and prove the existence and uniqueness of the solution to Eq.(9). Based on the contraction factor, we subsequently determine the choice of the projection radius used in the single-chain algorithm.

## A.1. Contraction of $\Pi T_\pi$

In this section, we prove that $\Pi T_\pi$ is a contractive operator (recall that $\Pi = I - e\mu^T$).

**Lemma A.1.** *The operator $\Pi T_\pi$ is a contractive operator satisfying*

$$\|\Pi T_\pi W_1 - \Pi T_\pi W_2\|_D \le \omega\|W_1 - W_2\|_D,$$

*where $\omega = \sqrt{\max_{\langle z,e\rangle_D=0,\|z\|_D=1} z^\top P^\top DPz} < 1$.*

*Proof of Lemma A.1.* First, we notice that, since $P$ is irreducible according to Assumption 2.1, the eigenvector whose eigenvalue is $1$ is unique (up to a constant factor) and must be the all-one vector.

For any vector $\Delta = W_1 - W_2$, it can be decomposed into $\Delta = \Delta_\| + \Delta_\perp$ where $\langle \Delta_\perp, e\rangle_D = 0$ and $\langle \Delta_\|, e\rangle_D = \|\Delta_\|\|_D \cdot \|e\|_D$. With such decomposition,

$$\|\Pi T_\pi W_1 - \Pi T_\pi W_2\|_D = \|\Pi P \Delta_\perp\|_D.$$

Therefore,

$$\|\Pi T_\pi W_1 - \Pi T_\pi W_2\|_D^2 \le \max_{\langle \Delta_\perp,e\rangle_D=0} \Delta_\perp^\top P^\top DP\Delta_\perp \le \max_{\langle z,e\rangle_D=0,\|z\|_D=1} z^\top P^\top DPz \cdot \|\Delta\|_D^2,$$

which indicates the contraction factor is

$$\omega := \sqrt{\max_{\langle z,e\rangle_D=0,\|z\|_D=1} z^\top P^\top DPz}.$$

It is easy to see that this factor cannot be larger than $1$. If it is exactly $1$, then there exists $z$ such that

$$\|z\|_D = 1 = \max_{\langle z,e\rangle_D=0,\|z\|_D=1} z^\top P^\top DPz = \|Pz\|_D.$$

This indicates that $z$ must be a multiple of all-one vector, which contradicts with $\langle z, e\rangle_D = 0$. $\qquad\square$

## A.2. Existence and Uniqueness of $\theta^*$

Recall that in Lemma 3.1, we define $\theta^*$ as the solution to the linear system $\Phi^\top D(I - \Pi P)\Phi\theta = \Phi^\top DR$ and we claim that $\theta^*$ exists and is unique. In this section, we give the proof of this lemma.

*Proof of Lemma 3.1.* Recall that in Eq.(8) we already established that

$$\eta = \min_{x:\|x\|=1} \|\Phi x\|_{\text{Dir}}^2 + (\mu^\top \Phi x)^2 > 0.$$

Observe that

$$\|\Phi x\|_{\text{Dir}}^2 + (\mu^\top \Phi x)^2 = x^\top \Phi^\top D(I - P)\Phi x + x^\top \Phi^\top \mu\mu^\top \Phi x = x^\top \Phi^\top D(I - \Pi P)\Phi x.$$

Suppose $\Phi^\top D(I - \Pi P)\Phi$ is not invertible. Then there exists a nonzero vector $x_0$ such that $x_0^\top \Phi^\top D(I - \Pi P)\Phi x_0 = 0$, which contradicts the fact that $\eta > 0$. Therefore, the matrix $\Phi^\top D(I - \Pi P)\Phi$ must be invertible. It follows that the linear system admits a unique solution $\theta^*$. $\qquad\square$

### A.3. Choice of Projection Radius

Recall that in our single chain algorithm, we need to project both $w$ and $\theta$ onto a ball of radius $R_w$ and $R_\theta$, respectively. In this section, we will discuss on the choice of radius such that $w^*$ and $\theta^*$ are in the feasible set.

**Bound on $\|w^*\|$:** This is easier because we can explicitly write $w^* = \Phi^\top \mu$. Using the fact that $\|\phi(s)\| \leq 1$ we can conclude $\|w^*\| \leq 1$. Therefore, we need $R_w \geq 1$.

**Bound on $\|\theta^*\|$:** Recall that $W^*$ is defined as

$$W^* = \lim_{t \to \infty} \sum_{k=0}^{t-1} (P^k - e\mu^\top) R.$$

According to Eq.(4),

$$([P^k]_i - \mu^\top) R \leq \|[P^k]_i - \mu^\top\|_1 \cdot \|R\|_\infty \leq r_{\max} C \beta^k,$$

which indicates $\|W^*\|_D \leq r_{\max} C / (1 - \beta)$.

By the Pythagorean theorem,

$$\begin{aligned}
\|\Phi\theta^* - W^*\|_D^2 &= \|W^* - \Pi_D \Pi T_\pi W^*\|_D^2 + \|\Pi_D \Pi T_\pi W^* - \Phi\theta^*\|_D^2 \\
&\leq \|W^* - \Pi_D \Pi T_\pi W^*\|_D^2 + \omega^2 \|\Phi\theta^* - W^*\|_D^2,
\end{aligned}$$

where we use the fact $\Pi T_\pi \Phi\theta^* = \Phi\theta^*$ and Lemma A.1. Therefore,

$$\|\Phi\theta^* - W^*\|_D \leq \frac{1}{\sqrt{1 - \omega^2}} \|W^* - \Pi_D \Pi T_\pi W^*\|_D.$$

Further, according to the Pythagorean theorem, $\|W^* - \Pi_D \Pi T_\pi W^*\|_D \leq \|W^*\|_D$. Therefore,

$$\|\Phi\theta^*\|_D \leq \|W^*\|_D + \|\Phi\theta^* - W^*\|_D \leq \frac{2}{\sqrt{1 - \omega^2}} \|W^*\|_D \leq \frac{2 r_{\max} C}{(1 - \beta)\sqrt{1 - \omega^2}}.$$

This suggests that

$$\|\theta^*\| \leq \frac{1}{\sqrt{\lambda_{\min}(\Phi^T D\Phi)}} \|\Phi\theta^*\|_D \leq \frac{2 r_{\max} C}{(1 - \beta)\sqrt{(1 - \omega^2)\lambda_{\min}(\Phi^\top D\Phi)}}.$$

Therefore, to ensure $\theta^*$ is in the feasible set, we can set $R_\theta$ such that

$$R_\theta \geq \frac{2 r_{\max} C}{(1 - \beta)\sqrt{(1 - \omega^2)\lambda_{\min}(\Phi^\top D\Phi)}}.$$

## B. Comparison with Previous Works

In this section, we will expand our result on reward estimation and compare the difference between condition numbers with previous works.

### B.1. Reward Estimation

Many previous works (Zhang et al., 2021b; Haque & Maguluri, 2024) also include convergence of the averaged reward function. Although not stated in our theorems, we can also achieve such convergence as mentioned in Eq.(19). In this section, we give a proof of Eq.(19) based on a Central Limit Theorem for Markov chains.

*Proof.* For all $t = \tau_{\mathrm{mix}}, \ldots, T$, we can decompose $\mathbb{E}\left[(r_t - g)^2\right]$ as

$$\mathbb{E}\left[(r_t - g)^2\right] \leq 2\mathbb{E}\left[(r_t - \mathbb{E}[r_t])^2\right] + 2\mathbb{E}\left[(\mathbb{E}[r_t] - g)^2\right] = 2\mathrm{Var}(r_t) + 2(\mathbb{E}[r_t] - g)^2.$$

First, we bound the variance of $r_t$, $\mathrm{Var}(r_t) = \mathbb{E}\left[(r_t - g)^2\right]$. Let $\gamma_k = \mathrm{Cov}(r_{S_0}, r_{S_k})$ be the covariance function. It is well known that

$$\mathrm{Var}(r_t) = \frac{1}{t^2} \sum_{i=0}^{t-1} \sum_{j=0}^{t-1} \gamma_{|i-j|}.$$

By changing the order of summation,

$$\sum_{i=0}^{t-1} \sum_{j=0}^{t-1} \gamma_{|i-j|} = \sum_{i=-(t-1)}^{t-1} (t - |i|)\gamma_{|i|} \leq n \sum_{-\infty}^{\infty} \gamma_i = t \left( \gamma_0 + 2 \sum_{i=1}^{\infty} \gamma_i \right).$$

Since all Markov chains considered in this paper is $V$-uniformly ergodic with $V = 1$ and the reward function is also bounded since the Markov chain is finite, the conditions of Theorem 17.0.1 in (Meyn & Tweedie, 2012) hold (up to a constant factor which will not change the final result) and therefore, the term $\gamma^2 := (\gamma_0 + 2\sum_{i=1}^{\infty} \gamma_i)$ must be finite. We further conclude

$$\mathrm{Var}(r_t) \leq \frac{1}{t}\gamma^2 = O\left(\frac{1}{t}\right).$$

Next, we bound the other term on the right hand side, $2(\mathbb{E}[r_t] - g)^2$, using Eq.(4). Since each reward is at most $r_{\max}$, the averaged reward $g$ can also be at most $r_{\max}$. Therefore,

$$|\mathbb{E}[r_{\tau_{\mathrm{mix}}}] - g| = \frac{1}{\tau_{\mathrm{mix}}} \sum_{i=0}^{\tau_{\mathrm{mix}}-1} (r_{S_i} - g) \leq 2r_{\max}.$$

Since we also have $r_t = \tau_{\mathrm{mix}} r_{\tau_{\mathrm{mix}}}/t + \sum_{i=\tau_{\mathrm{mix}}}^{t-1} r_t/t$,

$$\begin{aligned}|\mathbb{E}[r_t] - g| &\leq \frac{\tau_{\mathrm{mix}}}{t}|\mathbb{E}[r_{\tau_{\mathrm{mix}}}] - g| + \frac{1}{t} \sum_{i=\tau_{\mathrm{mix}}}^{t-1} |\mathbb{E}[r_i] - g| \\ &\leq \frac{2\tau_{\mathrm{mix}}}{t}r_{\max} + \frac{r_{\max}}{t} \sum_{i=\tau_{\mathrm{mix}}}^{t-1} \|P_i(\cdot|s_0) - \mu\|_1 \\ &\leq \tilde{O}\left(\frac{1}{t}\right).\end{aligned}$$

Therefore, combining both bounds and we conclude that

$$\mathbb{E}\left[(r_T - g)^2\right] = O(1/T).$$

$\square$

## B.2. Condition Numbers

We notice that, in (Zhang et al., 2021b) and (Li et al., 2024), the definitions on condition number are both different from ours. Our condition number is defined by

$$\eta_1 = \min_{||x||=1} \left\{ ||\Phi x||_{\text{Dir}}^2 + (\mu^\top \Phi x)^2 \right\}.$$

Meanwhile, in (Zhang et al., 2021b),

$$\eta_2 = \min_{||x||=1, x^T e = 0} ||\Phi x||_{\text{Dir}}^2$$

and in (Li et al., 2024; Kim et al., 2025),

$$\eta_3 = \left( \min_{||x||=1} x^\top \Phi^\top D \Phi x \right) \cdot \left( \min_{\langle y, e \rangle_D = 0, ||y||_D = 1} y^\top D (I - P) y \right).$$

Therefore, in this section we will describe this difference.

**Remark:** In (Li et al., 2024), the condition number is actually defined as $\min_{y \neq e, ||y||_D = 1} y^\top D(I - P)y$. However, we can always decompose it as $y = y_\perp + y_\parallel$ where $\langle y_\perp, e \rangle_D = 0$ and $\langle \Delta_\parallel, e \rangle_D = ||y_\parallel||_D \cdot ||e||_D$. We can easily check $y^\top D(I - P)y = y_\perp^\top D(I - P)y_\perp$. Therefore, $\eta_3$ is actually greater than the condition number defined in (Li et al., 2024), offering an optimistic approximation of their sample complexity.

### B.2.1. DIFFERENCE BETWEEN $\eta_1$ AND $\eta_2$

**Lemma B.1.** *We have*

$$\eta_1 \geq \frac{\mu_{\min}}{n \mu_{\max} ||\mu||^2} \eta_2.$$

Notice that $\eta_1$ defined in the above lemma corresponds to the condition number in this paper and $\eta_2$ corresponds to the condition number in (Zhang et al., 2021b). We can conclude that when $\mu$ is a multiple of the all-one vector, them $\eta_1$ is no less than $\eta_2$, which suggests our result is better. In other cases it is difficult to compare them.

Now we provide the proof to Lemma B.1.

*Proof of Lemma B.1.* It is useful to define the $\mu$–weighted inner product and norm by

$$\langle x, y \rangle_\mu = \sum_{s \in \mathcal{S}} \mu(s) x(s) y(s), \quad ||x||_\mu = \sqrt{\langle x, x \rangle_\mu}.$$

We first define $\eta_1', \eta_2'$ to be

$$\eta_1' = \min_{||x||_\mu = 1} \left\{ ||x||_{\text{Dir}}^2 + (\mu^\top x)^2 \right\},$$

and

$$\eta_2' = \min_{||x||_\mu = 1, \mu^\top x = 0} ||x||_{\text{Dir}}^2.$$

We can show a simple fact that $\eta_1' \geq \eta_2'$ when $\eta_2' \leq 1$. We notice that the Dirichlet semi-norm is invariant under addition of a constant. In other words, for any $x \in \mathbb{R}^n$ and $c \in \mathbb{R}$,

$$||x + ce||_{\text{Dir}} = ||x||_{\text{Dir}}.$$

Therefore, we can decompose

$$x = ce + z, \quad \text{where } c \in \mathbb{R} \text{ and } \mu^\top z = 0.$$

On one hand, we obtain $||x||_{\text{Dir}} = ||z||_{\text{Dir}}$. On the other hand,

$$||x||_\mu^2 = ||ce||_\mu^2 + ||z||_\mu^2 = c^2 ||e||_\mu^2 + ||z||_\mu^2 = c^2 + ||z||_\mu^2 = 1.$$

Therefore,

$$||x||_{\text{Dir}}^2 + (\mu^\top x)^2 = ||z||_{\text{Dir}}^2 + (\mu^\top x)^2 \geq \eta_2'(1 - c^2) + c^2 = \eta_2 + c^2(1 - \eta_2).$$

where the second equation uses the fact that $\|z\|_{\text{Dir}}^2 \geq \eta_2' \|z\|_\mu^2$. We conclude that $\eta_1' \geq \eta_2'$ as long as $\eta_2' \leq 1$.

Now we return to the original problem. To distinguish, we denote $\eta_1$ to be what is defined in our paper. Namely,

$$\eta_1 = \min_{\|x\|=1} \left\{ \|x\|_{\text{Dir}}^2 + (\mu^\top x)^2 \right\}.$$

According to (Zhang et al., 2021b), we define $\eta_2$ to be

$$\eta_2 = \min_{\|x\|=1, x^\top e = 0} \|x\|_{\text{Dir}}^2.$$

We first notice the simple fact

$$\mu_{\max}^{-1} \min_{\|x\|=1} x^\top A x = \frac{x^\top A x}{\mu_{\max}\|x\|_2^2} \leq \min_{x:\|x\|_\mu=1} x^\top A x \leq \frac{x^\top A x}{\mu_{\min}\|x\|_2^2} = \mu_{\min}^{-1} \min_{\|x\|=1} x^\top A x.$$

We introduce the following lemma:

**Lemma B.2.** *Let $A \in \mathbb{R}^{n \times n}$ be a symmetric positive semi-definite matrix satisfying $Ae = 0$. Let $\mu \in \mathbb{R}^n$ be a vector with strictly positive entries satisfying*

$$0 < \mu_{\min} \leq \mu_i \leq \mu_{\max} \quad \text{for } i = 1, \ldots, n, \qquad \text{and} \qquad \sum_{i=1}^n \mu_i = 1.$$

*Define*

$$\lambda := \min_{\|x\|_2=1, x^\top e=0} x^\top A x, \quad \lambda^{(\mu)} := \min_{\|x\|_2=1, x^\top \mu=0} x^\top A x$$

*and*

$$\lambda' := \min_{\|x\|_2=1} x^\top A x, \quad \lambda^{(\mu)'} := \min_{\|x\|_\mu=1} x^\top A x.$$

*Then we conclude*

$$\frac{\lambda}{n\|\mu\|_2^2} \leq \lambda^{(\mu)}, \quad \mu_{\max}^{-1}\lambda' \leq \lambda^{(\mu)'} \leq \mu_{\min}^{-1}\lambda'.$$

With the above lemma, one can show

$$\eta_2' \geq \frac{1}{n\mu_{\max}\|\mu\|^2}\eta_2$$

and

$$\eta_1' \leq \mu_{\min}^{-1}\eta_1.$$

Therefore,

$$\eta_1 \geq \frac{\mu_{\min}}{n\mu_{\max}\|\mu\|^2}\eta_2.$$

Now, we only need to prove the above lemma. We split the proof into two parts.

First, we show that $\mu_{\max}^{-1}\lambda' \leq \lambda^{(\mu)'} \leq \mu_{\min}^{-1}\lambda'$. We can always rewrite $\lambda'$ and $\lambda^{(\mu)'}$ as

$$\lambda' = \min_{x^\top \mu=0} \frac{x^\top A x}{\|x\|^2}, \quad \lambda^{(\mu)'} = \min_{x^\top \mu=0} \frac{x^\top A x}{\|x\|_\mu^2}.$$

The result followed by applying the fact $\mu_{\min}\|x\|^2 \leq \|x\|_\mu^2 \leq \mu_{\max}\|x\|^2$.

Next, we show $\frac{\lambda}{n\|\mu\|_2^2} \leq \lambda^{(\mu)}$. Let $x \in \mathbb{R}^n$ be a vector satisfying

$$x^\top \mu = 0, \quad \|x\| = 1.$$

Define

$$y = x - de, \quad \text{where } d = \frac{x^\top e}{n},$$

we can easily check $y^\top e = 0$ and $y^\top A y = x^\top A x$. Consider the norm of $y$,

$$\|y\|_2^2 = \|x - de\|_2^2 = \|x\|_2^2 - 2d\, x^\top e + d^2 \|e\|_2^2 = 1 - \frac{(x^\top e)^2}{n}\,.$$

To bound $|x^\top e|$, decompose $e$ by

$$e = \frac{1}{\|\mu\|^2}\,\mu + w, \qquad \text{with } w^\top \mu = 0.$$

Then $x^\top e = x^\top w$. By the Cauchy–Schwarz inequality, $|x^\top e| \le \|w\|_2$. Notice that

$$\|w\|_2^2 = \|e\|_2^2 - \left\|\frac{\mu}{\|\mu\|_2^2}\right\|^2 = n - \frac{1}{\|\mu\|_2^2}\,,$$

Since $e^\top \mu = 1$,

$$\|y\|_2^2 \ge 1 - \frac{n - \frac{1}{\|\mu\|_2^2}}{n} = \frac{1}{n\|\mu\|_2^2}\,.$$

Now, letting $\hat{y} = y/\|y\|_2$, we have $\hat{y}^\top e = 0$ and thus

$$\hat{y}^\top A \hat{y} \ge \lambda.$$

It follows that

$$x^\top A x = y^\top A y = \|y\|_2^2\, \hat{y}^\top A \hat{y} \ge \frac{\lambda_2}{n\|\mu\|_2^2}\,.$$

Take the minimum over all unit vectors $x$ with $x^\top \mu = 0$ yields

$$\lambda^{(\mu)} \ge \frac{\lambda_2}{n\|\mu\|_2^2}\,.$$

$\square$

### B.2.2. DIFFERENCE BETWEEN $\eta_1$ AND $\eta_3$

**Lemma B.3.** *We have*

$$\eta_1 \ \ge \ \frac{1}{2}\, \eta_3.$$

*Proof.* Recall

$$\eta_1 = \min_{\|x\|=1}\left\{\|\Phi x\|_{\mathrm{Dir}}^2 + (\mu^\top \Phi x)^2\right\}, \qquad \eta_3 = \left(\min_{\|x\|=1} x^\top \Phi^\top D \Phi x\right) \cdot \left(\min_{\langle y,e\rangle_D = 0,\ \|y\|_D = 1} y^\top D(I - P)y\right),$$

where $D = \mathrm{diag}(\mu)$, $\langle a, b\rangle_D = a^\top D b$, and $e$ is the all-ones vector.

Let

$$\lambda := \min_{\langle y,e\rangle_D = 0,\ \|y\|_D = 1} y^\top D(I - P)y, \qquad \sigma := \min_{\|x\|=1} x^\top \Phi^\top D \Phi x.$$

Then $\eta_3 = \sigma\lambda$.

First, for any $v \in \mathbb{R}^{|\mathcal{S}|}$,

$$\begin{aligned}
\|v\|_{\mathrm{Dir}}^2 &= \frac{1}{2}\sum_{s,s'}\mu(s)P(s'|s)\big(v(s) - v(s')\big)^2 \\
&= \sum_s \mu(s)v(s)^2 - \sum_{s,s'}\mu(s)P(s'|s)v(s)v(s') \\
&= v^\top D(I - P)v,
\end{aligned}$$

where we used $\sum_{s'} P(s'|s) = 1$ and stationarity $\sum_s \mu(s)P(s'|s) = \mu(s')$.

We also remark on the standard observation that multiplication by $P$ is a contraction in the $D$-norm:

$$\|Pv\|_D^2 = \sum_s \mu(s)\Big(\sum_{s'} P(s'|s)v(s')\Big)^2 \leq \sum_s \mu(s)\sum_{s'} P(s'|s)v(s')^2 = \sum_{s'} \mu(s')v(s')^2 = \|v\|_D^2,$$

where the inequality is Jensen and the equality again uses stationarity. Hence $\|Pv\|_D \leq \|v\|_D$. Therefore for any $y$ with $\|y\|_D = 1$,

$$y^\top D(I-P)y = \|y\|_D^2 - \langle y, Py \rangle_D \leq 1 + |\langle y, Py \rangle_D| \leq 1 + \|y\|_D \|Py\|_D \leq 2.$$

Taking the minimum over the constraint set gives $\lambda \leq 2$.

Having established that, we next fix arbitrary $v \in \mathbb{R}^{|\mathcal{S}|}$. Let

$$c := \langle v, e \rangle_D = v^\top D e = \mu^\top v, \qquad u := v - ce,$$

so that $\langle u, e \rangle_D = 0$. Since $\|e\|_D^2 = e^\top D e = \sum_s \mu(s) = 1$ and $u \perp e$ in $\langle \cdot, \cdot \rangle_D$,

$$\|v\|_D^2 = \|u\|_D^2 + c^2.$$

Also, adding a constant does not change the Dirichlet seminorm, so $\|v\|_{\mathrm{Dir}} = \|u\|_{\mathrm{Dir}}$. Using Step 1 and the definition of $\lambda$,

$$\|v\|_{\mathrm{Dir}}^2 = \|u\|_{\mathrm{Dir}}^2 = u^\top D(I-P)u \ \geq \ \lambda \|u\|_D^2.$$

Therefore,

$$\|v\|_{\mathrm{Dir}}^2 + (\mu^\top v)^2 = \|u\|_{\mathrm{Dir}}^2 + c^2 \geq \lambda\|u\|_D^2 + c^2 \geq \frac{\lambda}{2}\big(\|u\|_D^2 + c^2\big) = \frac{\lambda}{2}\|v\|_D^2,$$

where we used $\lambda \leq 2$ so that $(\lambda/2)c^2 \leq c^2$.

**Step 3:** Applying Step 2 to $v = \Phi x$ gives, for every $x \in \mathbb{R}^d$,

$$\|\Phi x\|_{\mathrm{Dir}}^2 + (\mu^\top \Phi x)^2 \ \geq \ \frac{\lambda}{2}\|\Phi x\|_D^2 \ = \ \frac{\lambda}{2} x^\top \Phi^\top D \Phi x.$$

Taking $\min_{\|x\|=1}$ on both sides yields

$$\eta_1 = \min_{\|x\|=1}\Big\{\|\Phi x\|_{\mathrm{Dir}}^2 + (\mu^\top \Phi x)^2\Big\} \ \geq \ \frac{\lambda}{2}\min_{\|x\|=1} x^\top \Phi^\top D \Phi x = \frac{\lambda}{2}\sigma = \frac{1}{2}\eta_3.$$

$\square$

### B.2.3. RELATION BETWEEN PROJECTION AND CONDITION NUMBER

In (Kim et al., 2025), their sample complexity depends on the projection radius $R_{\mathrm{proj}}$. Although they did not have a discussion on the projection radius, we will assume $R_{\mathrm{proj}} = R_\theta$ and compare their sample complexity for completeness.

**Lemma B.4.** *Suppose the projection radius is chosen to be*

$$R_{\mathrm{proj}} = R_\theta = \frac{2r_{\max}C}{(1-\beta)\sqrt{(1-\omega^2)\lambda_{\min}(\Phi^\top D \Phi)}}.$$

*We have*

$$R_\theta^2 \geq \frac{2r_{\max}^2 C^2}{(1-\beta)^2 \eta_3}.$$

*Proof of Lemma B.4.* For any vector $y$ such that $\|y\|_D = 1$ and $\langle y, e \rangle_D = 0$,

$$y^\top D(I-P)y = 1 - \langle y, Py \rangle_D.$$

By Cauchy–Schwarz,

$$\langle y, Py \rangle_D \leq \|y\|_D \|Py\|_D = \|Py\|_D \leq \omega.$$

Therefore, we have

$$\min_{\langle y,e \rangle_D = 0, \|y\|_D = 1} y^\top D(I - P)y \geq 1 - \omega.$$

Since $\omega < 1$ which is already established in Lemma A.1, $1 - \omega \geq (1 - \omega^2)/2$. Therefore,

$$\frac{1}{1 - \omega^2} \geq \frac{1}{2 \min_{\langle y,e \rangle_D = 0, \|y\|_D = 1} y^\top D(I - P)y}$$

Now we obtain

$$R_\theta^2 \geq \frac{(2r_{\max}C)^2}{2(1 - \beta)^2 \min_{\langle y,e \rangle_D = 0, \|y\|_D = 1} y^\top D(I - P)y \cdot \lambda_{\min}(\Phi^\top D\Phi)} = \frac{2r_{\max}^2 C^2}{(1 - \beta)^2 \eta_3}.$$

$\square$

## C. Analysis of Double Chain Algorithm with Constant Stepsize

In this section, we will give a detailed proof of Theorem 4.1 and Theorem 4.2. Notice that these proofs can also be applied to the tabular case if one sets $\Phi = I$.

Throughout the analysis, we denote $\mathcal{F}_t$ as the history up to iteration $t - 1$, i.e.,

$$\mathcal{F}_t := \sigma(\{s_0, \ldots, s_{t-1}\} \cup \{\hat{s}_0, \ldots, \hat{s}_{t-1}\}). \tag{filtration}$$

In particular, $\theta_t$ and $\delta_t$ is $\mathcal{F}_t$-measurable. We also denote $B = 2\|\delta_0\| + (r_{\max} + \|\theta^*\|)$.

Before going to the proof, we first record some useful properties about $f(s, \hat{s}, \theta)$ and $g(s, s', \theta)$.

Define the mean field

$$\bar{h}(\theta) := \mathbb{E}_\mu[f(s, \hat{s}, \theta) + g(s, s', \theta)],$$

where under $\mathbb{E}_\mu$ we sample $s \sim \mu$, $\hat{s} \sim \mu$ independently, and then sample $s' \sim P(\cdot \mid s)$. By Eq. (14), it is straightforward to check that

$$\begin{aligned}
\bar{h}(\theta) &= -\Phi^\top \mu \mu^\top (R + \Phi\theta) + \Phi^\top D(R + P\Phi\theta - \Phi\theta) \\
&= -\Phi^\top \mu \mu^\top \Phi\delta + \Phi^\top D(P - I)\Phi\delta,
\end{aligned} \tag{21}$$

where $\delta := \theta - \theta^*$.

Beyond the above fact, we have two additional lemmas.

**Lemma C.1.** *For any $t \geq 0$, we have*

$$\begin{aligned}
\|f(s_t, \hat{s}_t, \theta_t)\| &\leq \|\delta_t\| + r_{\max} + \|\theta^*\|, \\
\|g(s_t, s'_t, \theta_t)\| &\leq 2(\|\delta_t\| + r_{\max} + \|\theta^*\|).
\end{aligned}$$

*Proof of Lemma C.1.* According to Eq.(16),

$$\begin{aligned}
\|f(s_t, \hat{s}_t, \theta_t)\| &\leq \left(|r_{s_t}| + |\phi(s_t)^\top \theta_t|\right) \|\phi(\hat{s}_t)\| \\
&\leq (r_{\max} + \|\theta_t\| \|\phi(s_t)\|) \|\phi(\hat{s}_t)\| &\text{(Cauchy–Schwarz)} \\
&\leq r_{\max} + \|\theta_t\| &(\|\phi(\cdot)\| \leq 1) \\
&\leq r_{\max} + \|\theta_t - \theta^*\| + \|\theta^*\| \\
&= \|\delta_t\| + r_{\max} + \|\theta^*\|.
\end{aligned}$$

Similarly,

$$\begin{aligned}
\|g(s_t, s'_t, \theta_t)\| &\leq \left(|r_{s_t}| + |\phi(s'_t)^\top \theta_t| + |\phi(s_t)^\top \theta_t|\right) \|\phi(s_t)\| \\
&\leq r_{\max} + 2\|\theta_t\| &(\|\phi(\cdot)\| \leq 1) \\
&\leq r_{\max} + 2\|\theta_t - \theta^*\| + 2\|\theta^*\| \\
&\leq 2(\|\delta_t\| + r_{\max} + \|\theta^*\|).
\end{aligned}$$

$\square$

**Lemma C.2.** *The function $f$ is 1-Lipschitz and $g$ is 2-Lipschitz with respect to $\theta$, i.e.,*

$$\|g(s, s', \theta_1) - g(s, s', \theta_2)\| \leq 2\|\theta_1 - \theta_2\|, \quad \|f(s, \hat{s}, \theta_1) - f(s, \hat{s}, \theta_2)\| \leq \|\theta_1 - \theta_2\|.$$

*Proof of Lemma C.2.* Using Eq.(16) and $\|\phi(\cdot)\| \leq 1$,

$$g(s, s', \theta_1) - g(s, s', \theta_2) = \left((\phi(s') - \phi(s))^\top (\theta_1 - \theta_2)\right) \phi(s),$$

hence

$$\|g(s, s', \theta_1) - g(s, s', \theta_2)\| \leq \|\phi(s') - \phi(s)\| \|\phi(s)\| \|\theta_1 - \theta_2\| \leq 2\|\theta_1 - \theta_2\|.$$

Likewise,

$$f(s, \hat{s}, \theta_1) - f(s, \hat{s}, \theta_2) = -(\phi(s)^\top (\theta_1 - \theta_2)) \phi(\hat{s}),$$

so $\|f(s, \hat{s}, \theta_1) - f(s, \hat{s}, \theta_2)\| \leq \|\theta_1 - \theta_2\|.$ $\square$

Now we are ready to provide our proof of Theorem 4.1.

## C.1. Proof of Theorem 4.1

*Proof of Theorem 4.1.* Let $u_t := f(s_t, \hat{s}_t, \theta_t) + g(s_t, s'_t, \theta_t)$. From Eq.(15) with $\alpha_t = \alpha$,

$$\|\delta_{t+1}\|^2 = \|\delta_t\|^2 + 2\alpha\, \delta_t^\top u_t + \alpha^2\|u_t\|^2.$$

Taking conditional expectation given $\mathcal{F}_t$, and using that $\delta_t$ is $\mathcal{F}_t$-measurable,

$$\mathbb{E}\big[\|\delta_{t+1}\|^2 \mid \mathcal{F}_t\big] = \|\delta_t\|^2 + 2\alpha\, \delta_t^\top \mathbb{E}[u_t \mid \mathcal{F}_t] + \alpha^2\, \mathbb{E}[\|u_t\|^2 \mid \mathcal{F}_t].$$

**Step 1: Drift term.** We first introduce the following lemma:

**Lemma C.3.** *The linear function $h(\theta) := \Phi^\top D(I - P)\Phi\theta$ is a gradient splitting of the quadratic function $\|\Phi\theta\|_{\mathrm{Dir}}^2$, i.e.,*

$$\langle \theta, h(\theta)\rangle = \frac{1}{2}\langle \theta, \nabla_\theta\|\Phi\theta\|_{\mathrm{Dir}}^2\rangle, \quad \forall\, \theta \in \mathbb{R}^d.$$

*As a result, we have $\|\Phi\theta\|_{\mathrm{Dir}}^2 = \theta^\top h(\theta)$ for all $\theta \in \mathbb{R}^d$.*

The proof of this lemma can be found in Section C.2.

Under i.i.d. sampling, $(s_t, \hat{s}_t)$ are independent of $\mathcal{F}_t$ with marginals $\mu$, and $s'_t \sim P(\cdot \mid s_t)$. Therefore,

$$\mathbb{E}[u_t \mid \mathcal{F}_t] = \bar{h}(\theta_t),$$

where $\bar{h}(\cdot)$ is defined in Eq.(21).

Using Eq.(21) with $\delta = \delta_t$,

$$\begin{aligned}
\delta_t^\top \bar{h}(\theta_t) &= \delta_t^\top \Phi^\top D(P - I)\Phi\, \delta_t - \delta_t^\top \Phi^\top \mu\mu^\top \Phi\, \delta_t \\
&= -\delta_t^\top \Phi^\top D(I - P)\Phi\, \delta_t - \|\mu^\top \Phi\delta_t\|^2 \\
&= -\|\Phi\delta_t\|_{\mathrm{Dir}}^2 - \|\mu^\top \Phi\delta_t\|^2 && \text{(Lemma C.3)} \\
&\leq -\eta\,\|\delta_t\|^2. && \text{(Eq. (8))}
\end{aligned}$$

**Step 2: Second-moment term.** By Lemma C.1,

$$\|u_t\| \leq \|f(s_t, \hat{s}_t, \theta_t)\| + \|g(s_t, s'_t, \theta_t)\| \leq 3(\|\delta_t\| + r_{\max} + \|\theta^*\|). \qquad \text{(Lemma C.1)}$$

Hence,

$$\begin{aligned}
\mathbb{E}[\|u_t\|^2 \mid \mathcal{F}_t] &\leq 9(\|\delta_t\| + r_{\max} + \|\theta^*\|)^2 \\
&\leq 18\|\delta_t\|^2 + 18(r_{\max} + \|\theta^*\|)^2. && ((a+b)^2 \leq 2a^2 + 2b^2)
\end{aligned}$$

**Step 3: Combine.** Plugging the two bounds back,

$$\mathbb{E}\big[\|\delta_{t+1}\|^2 \mid \mathcal{F}_t\big] \leq (1 - 2\alpha\eta + 18\alpha^2)\|\delta_t\|^2 + 18\alpha^2(r_{\max} + \|\theta^*\|)^2.$$

Since $\alpha \leq \eta/18$, we have $18\alpha^2 \leq \alpha\eta$, hence

$$1 - 2\alpha\eta + 18\alpha^2 \leq 1 - \alpha\eta,$$

and therefore

$$\mathbb{E}\big[\|\delta_{t+1}\|^2 \mid \mathcal{F}_t\big] \leq (1 - \alpha\eta)\|\delta_t\|^2 + 18\alpha^2(r_{\max} + \|\theta^*\|)^2.$$

Taking expectation and iterating the recursion yields

$$\mathbb{E}\|\delta_T\|^2 \leq (1 - \alpha\eta)^T \mathbb{E}\|\delta_0\|^2 + 18\alpha^2(r_{\max} + \|\theta^*\|)^2 \sum_{k=0}^{T-1}(1 - \alpha\eta)^k.$$

Using $\sum_{k=0}^{T-1}(1 - \alpha\eta)^k \leq \frac{1}{\alpha\eta}$ and $1 - x \leq e^{-x}$,

$$\mathbb{E}\|\delta_T\|^2 \leq e^{-\alpha\eta T}\mathbb{E}\|\delta_0\|^2 + \frac{18\alpha(r_{\max} + \|\theta^*\|)^2}{\eta}.$$

$\square$

## C.2. Proof of Lemma C.3

*Proof of Lemma C.3.* According to Eq. (7) and the stationarity $\mu^\top = \mu^\top P$,

$$\|f\|_{\mathrm{Dir}}^2 = \frac{1}{2} \sum_{s,s'} \mu(s) P(s'|s) \big( f(s)^2 + f(s')^2 - 2f(s)f(s') \big)$$

$$= \sum_{s} \mu(s) f(s)^2 - \sum_{s,s'} \mu(s) P(s'|s) f(s) f(s')$$

$$= f^\top D(I - P) f.$$

Taking $f = \Phi\theta$ gives

$$\|\Phi\theta\|_{\mathrm{Dir}}^2 = \theta^\top \Phi^\top D(I - P)\Phi\,\theta = \theta^\top h(\theta).$$

Since $\|\Phi\theta\|_{\mathrm{Dir}}^2$ is a quadratic form, its gradient is

$$\nabla_\theta \|\Phi\theta\|_{\mathrm{Dir}}^2 = \big( \Phi^\top D(I - P)\Phi + \Phi^\top (I - P)^\top D\Phi \big)\theta,$$

hence

$$\frac{1}{2}\langle \theta, \nabla_\theta \|\Phi\theta\|_{\mathrm{Dir}}^2 \rangle = \theta^\top \Phi^\top D(I - P)\Phi\,\theta = \langle \theta, h(\theta)\rangle.$$

$\square$

## C.3. Proof of Theorem 4.2

*Proof of Theorem 4.2.* For convenience, define

$$\bar{g}(\theta_t) := \sum_{s,s'} \mu(s) P(s'|s)\, g(s, s', \theta_t) = \Phi^\top D(R + P\Phi\theta_t - \Phi\theta_t),$$

$$\bar{f}(\theta_t) := \sum_{s,\hat{s}} \mu(s)\mu(\hat{s})\, f(s, \hat{s}, \theta_t) = -\Phi^\top \mu\mu^\top (R + \Phi\theta_t). \tag{22}$$

Let $u_t := f(s_t, \hat{s}_t, \theta_t) + g(s_t, s'_t, \theta_t)$. Using that $\delta_t$ and $\bar{f}(\theta_t) + \bar{g}(\theta_t)$ are $\mathcal{F}_t$-measurable, we have

$$\mathbb{E}\big[\delta_t^\top \big( \Phi^\top D(P - I)\Phi\delta_t - \Phi^\top \mu\mu^\top \Phi\delta_t - \mathbb{E}[u_t \mid \mathcal{F}_t] \big)\big]$$

$$= \mathbb{E}\big[\delta_t^\top \big( \bar{g}(\theta_t) + \bar{f}(\theta_t) - \mathbb{E}[u_t \mid \mathcal{F}_t] \big)\big] \qquad\qquad\qquad (\text{Eq. } (21))$$

$$= \mathbb{E}\big[\delta_t^\top \big( \bar{g}(\theta_t) + \bar{f}(\theta_t) - u_t \big)\big]$$

$$= - \mathbb{E}\big[\delta_t^\top \big( (g(s_t, s'_t, \theta_t) - \bar{g}(\theta_t)) + (f(s_t, \hat{s}_t, \theta_t) - \bar{f}(\theta_t)) \big)\big].$$

Define $B$ and $G$ as in the theorem statement. We introduce the following lemma:

**Lemma C.4.** *Suppose*

$$\alpha \le \frac{\eta B^2}{(3\tau_{\mathrm{mix}} + 1)G},$$

*then* $\mathbb{E}[\|\delta_t\|^2] \le B^2$ *for all* $t \ge 0$.

The proof of this lemma can be found in Section C.4.

Since $\mathbb{E}[\|\delta_t\|^2] \le B^2$ for all $t \ge 0$, we know that the following lemma holds.

**Lemma C.5.** *Suppose* $t \ge \tau_{\mathrm{mix}}$. *With* $\bar{f}, \bar{g}$ *defined in Eq.(22), we have*

$$\mathbb{E}\big[\delta_t^\top \big(g(s_t, s'_t, \theta_t) - \bar{g}(\theta_t)\big)\big] \le \alpha \big(3B^2 + (r_{\max} + \|\theta^*\|)^2\big) + \big(42B^2 + 30(r_{\max} + \|\theta^*\|)^2\big)\tau_{\mathrm{mix}}\,\alpha,$$

$$\mathbb{E}\big[\delta_t^\top \big(f(s_t, \hat{s}_t, \theta_t) - \bar{f}(\theta_t)\big)\big] \le \alpha \big(3B^2 + (r_{\max} + \|\theta^*\|)^2\big) + \big(21B^2 + 15(r_{\max} + \|\theta^*\|)^2\big)\tau_{\mathrm{mix}}\,\alpha.$$

The proof of this lemma can be found in Section C.5.

Summing the two inequalities and multiplying by the factor $2\alpha$, we obtain for all $t \geq \tau_{\text{mix}}$,

$$-2\alpha\,\mathbb{E}\big[\delta_t^\top\big(\Phi^\top D(P-I)\Phi\delta_t - \Phi^\top\mu\mu^\top\Phi\delta_t - \mathbb{E}[u_t \mid \mathcal{F}_t]\big)\big]$$
$$\leq 4\alpha^2\Big(3B^2 + (r_{\max} + \|\theta^*\|)^2\Big) + \alpha^2\tau_{\text{mix}}\Big(126B^2 + 90(r_{\max} + \|\theta^*\|)^2\Big).$$

Using the same $I_1$ and $I_2$ bounds as in the i.i.d. case (so that the drift contributes $-(2\alpha\eta)\|\delta_t\|^2$ and the squared-norm term contributes $18\alpha^2(\|\delta_t\|^2 + (r_{\max} + \|\theta^*\|)^2)$), we obtain for any $t \geq \tau_{\text{mix}}$,

$$\mathbb{E}\|\delta_{t+1}\|^2 \leq (1-2\alpha\eta)\mathbb{E}\|\delta_t\|^2 + 18\alpha^2\,\mathbb{E}\|\delta_t\|^2 + 18\alpha^2(r_{\max} + \|\theta^*\|)^2$$
$$+ 4\alpha^2\Big(3B^2 + (r_{\max} + \|\theta^*\|)^2\Big) + \alpha^2\tau_{\text{mix}}\Big(126B^2 + 90(r_{\max} + \|\theta^*\|)^2\Big)$$
$$\leq (1-2\alpha\eta)\mathbb{E}\|\delta_t\|^2 + 18\alpha^2B^2 + 22\alpha^2(r_{\max} + \|\theta^*\|)^2 + 12\alpha^2B^2$$
$$+ \alpha^2\tau_{\text{mix}}\Big(126B^2 + 90(r_{\max} + \|\theta^*\|)^2\Big) \qquad\text{(Lemma C.4)}$$
$$\leq (1-2\alpha\eta)\mathbb{E}\|\delta_t\|^2 + \alpha^2\Big(42B^2 + 30(r_{\max} + \|\theta^*\|)^2\Big)(3\tau_{\text{mix}} + 1)$$
$$= (1-2\alpha\eta)\mathbb{E}\|\delta_t\|^2 + \alpha^2 G(3\tau_{\text{mix}} + 1).$$

Iterating the above inequality from $t = \tau_{\text{mix}}$ to $T-1$ gives

$$\mathbb{E}\|\delta_T\|^2 \leq (1-2\alpha\eta)^{T-\tau_{\text{mix}}}\mathbb{E}\|\delta_{\tau_{\text{mix}}}\|^2 + \alpha^2 G(3\tau_{\text{mix}} + 1)\sum_{k=0}^{T-\tau_{\text{mix}}-1}(1-2\alpha\eta)^k.$$

Using Lemma C.4, $\sum_{k\geq 0}(1-2\alpha\eta)^k \leq \frac{1}{2\alpha\eta}$, and $1-x \leq e^{-x}$, we obtain

$$\mathbb{E}\|\delta_T\|^2 \leq e^{-2\alpha\eta(T-\tau_{\text{mix}})}B^2 + \frac{\alpha G(3\tau_{\text{mix}} + 1)}{2\eta}.$$

$\square$

### C.4. Proof of Lemma C.4

*Proof of Lemma C.4.* We first handle $t \leq \tau_{\text{mix}}$ via a pathwise bound. From Eq.(15) and Lemma C.1,

$$\|\delta_{t+1}\| \leq \|\delta_t\| + \alpha\big(\|f(s_t, \hat{s}_t, \theta_t)\| + \|g(s_t, s_t', \theta_t)\|\big)$$
$$\leq \|\delta_t\| + 3\alpha(\|\delta_t\| + r_{\max} + \|\theta^*\|) = (1+3\alpha)\|\delta_t\| + 3\alpha(r_{\max} + \|\theta^*\|).$$

Iterating the inequality yields, for $t \leq \tau_{\text{mix}}$,

$$\|\delta_t\| \leq (1+3\alpha)^t\|\delta_0\| + 3\alpha(r_{\max} + \|\theta^*\|)\sum_{k=0}^{t-1}(1+3\alpha)^k.$$

Using $(1+3\alpha)^t \leq e^{3\alpha t}$ and $\sum_{k=0}^{t-1}(1+3\alpha)^k \leq t(1+3\alpha)^t$, we have

$$\|\delta_t\| \leq e^{3\alpha\tau_{\text{mix}}}\|\delta_0\| + 3\alpha\tau_{\text{mix}}e^{3\alpha\tau_{\text{mix}}}(r_{\max} + \|\theta^*\|).$$

The stepsize condition implies $\alpha\tau_{\text{mix}} \leq 1/6$ (indeed $G \geq 42B^2$ and $\eta \leq 3$ imply $\alpha \leq \eta/(42(3\tau_{\text{mix}} + 1)) \leq 1/(6\tau_{\text{mix}})$), so $e^{3\alpha\tau_{\text{mix}}} \leq e^{1/2} < 2$ and $6\alpha\tau_{\text{mix}} \leq 1$. Therefore,

$$\|\delta_t\| \leq 2\|\delta_0\| + 6\alpha\tau_{\text{mix}}(r_{\max} + \|\theta^*\|) \leq 2\|\delta_0\| + (r_{\max} + \|\theta^*\|) = B, \quad \forall t \leq \tau_{\text{mix}}.$$

Hence $\mathbb{E}\|\delta_t\|^2 \leq B^2$ for all $t \leq \tau_{\text{mix}}$.

Now consider $t > \tau_{\text{mix}}$ and use induction on $t$. Assume $\mathbb{E}\|\delta_k\|^2 \leq B^2$ for all $k \leq t$. Then we have the following lemma:

**Lemma C.6.** *Suppose $t \geq \tau_{\mathrm{mix}}$ and $\mathbb{E}[\|\delta_k\|^2] \leq B^2$ for all $k \leq t$. Then Lemma C.5 holds at time $t$.*

The proof of this lemma can be found in Section C.6.

Plugging those bounds into the recursion in the proof of Theorem 4.2 gives

$$\mathbb{E}\|\delta_{t+1}\|^2 \leq (1 - 2\alpha\eta)B^2 + \alpha^2 G(3\tau_{\mathrm{mix}} + 1).$$

Using the stepsize condition $\alpha \leq \frac{\eta B^2}{(3\tau_{\mathrm{mix}}+1)G}$, we have $\alpha^2 G(3\tau_{\mathrm{mix}} + 1) \leq \alpha\eta B^2$, hence

$$\mathbb{E}\|\delta_{t+1}\|^2 \leq (1 - 2\alpha\eta)B^2 + \alpha\eta B^2 \leq B^2.$$

This completes the induction and proves the claim for all $t \geq 0$. $\qquad\square$

### C.5. Proof of Lemma C.5

*Proof of Lemma C.5.* Fix any $t \geq \tau_{\mathrm{mix}}$. Lemma C.4 gives $\mathbb{E}\|\delta_k\|^2 \leq B^2$ for all $k \leq t$, so the assumptions of Lemma C.6 are satisfied. Lemma C.6 therefore implies the two inequalities stated in Lemma C.5. $\qquad\square$

### C.6. Proof of Lemma C.6

*Proof of Lemma C.6.* For simplicity, denote

$$g_t(\theta) := g(s_t, s_t', \theta), \qquad f_t(\theta) := f(s_t, \hat{s}_t, \theta).$$

**Step 1: the $g$-term.** Decompose

$$\mathbb{E}\left[(\theta_t - \theta^*)^\top (g_t(\theta_t) - \bar{g}(\theta_t))\right]$$
$$= \underbrace{\mathbb{E}\left[(\theta_t - \theta_{t-\tau_{\mathrm{mix}}})^\top (g_t(\theta_t) - \bar{g}(\theta_t))\right]}_{I_1} + \underbrace{\mathbb{E}\left[(\theta_{t-\tau_{\mathrm{mix}}} - \theta^*)^\top (g_t(\theta_{t-\tau_{\mathrm{mix}}}) - \bar{g}(\theta_{t-\tau_{\mathrm{mix}}}))\right]}_{I_2}$$
$$+ \underbrace{\mathbb{E}\left[(\theta_{t-\tau_{\mathrm{mix}}} - \theta^*)^\top (g_t(\theta_t) - g_t(\theta_{t-\tau_{\mathrm{mix}}}))\right]}_{I_3} + \underbrace{\mathbb{E}\left[(\theta_{t-\tau_{\mathrm{mix}}} - \theta^*)^\top (\bar{g}(\theta_{t-\tau_{\mathrm{mix}}}) - \bar{g}(\theta_t))\right]}_{I_4}.$$

**Term $I_1$.** We have

$$\theta_t - \theta_{t-\tau_{\mathrm{mix}}} = \sum_{i=t-\tau_{\mathrm{mix}}+1}^{t} \alpha\big(g_i(\theta_i) + f_i(\theta_i)\big),$$

so by Lemma C.1,

$$\|\theta_t - \theta_{t-\tau_{\mathrm{mix}}}\| \leq 3 \sum_{i=t-\tau_{\mathrm{mix}}+1}^{t} \alpha\big(\|\delta_i\| + r_{\mathrm{max}} + \|\theta^*\|\big).$$

Also $\|g_t(\theta_t) - \bar{g}(\theta_t)\| \leq \|g_t(\theta_t)\| + \|\bar{g}(\theta_t)\| \leq 4(\|\delta_t\| + r_{\mathrm{max}} + \|\theta^*\|)$. Therefore, since $2ab \leq a^2 + b^2$, we have

$$I_1 \leq \mathbb{E}[\|\theta_t - \theta_{t-\tau_{\mathrm{mix}}}\| \, \|g_t(\theta_t) - \bar{g}(\theta_t)\|]$$
$$\leq \mathbb{E}\left[12(\|\delta_t\| + c) \sum_{i=t-\tau_{\mathrm{mix}}+1}^{t} \alpha(\|\delta_i\| + c)\right]$$
$$\leq 12 \sum_{i=t-\tau_{\mathrm{mix}}+1}^{t} \alpha \, \mathbb{E}\left[\|\delta_t\|^2 + \|\delta_i\|^2 + 2c^2\right]$$
$$\leq 24 \left(B^2 + c^2\right) \sum_{i=t-\tau_{\mathrm{mix}}+1}^{t} \alpha.$$

**Term $I_2$.** Condition on $\mathcal{F}_{t-\tau_{\mathrm{mix}}}$. Given $\mathcal{F}_{t-\tau_{\mathrm{mix}}}$, the law of $(s_t, s_t')$ differs from $\mu(s)P(s'|s)$ by at most $\|P_{\tau_{\mathrm{mix}}}(\cdot|s_{t-\tau_{\mathrm{mix}}}) - \mu\|_1$, and by definition of $\tau_{\mathrm{mix}} = \tau_{\mathrm{mix}}(\alpha)$ we have $\|P_{\tau_{\mathrm{mix}}}(\cdot|s) - \mu\|_1 \leq \alpha$ for all $s$. Using Lemma C.1, $\sup_{s,s'} \|g(s, s', \theta_{t-\tau_{\mathrm{mix}}})\| \leq 2(\|\delta_{t-\tau_{\mathrm{mix}}}\| + c)$, hence

$$\|\mathbb{E}\left[g_t(\theta_{t-\tau_{\mathrm{mix}}}) - \bar{g}(\theta_{t-\tau_{\mathrm{mix}}}) \mid \mathcal{F}_{t-\tau_{\mathrm{mix}}}\right]\| \leq 2\alpha(\|\delta_{t-\tau_{\mathrm{mix}}}\| + c).$$

Therefore,

$$
\begin{aligned}
I_2 &= \mathbb{E}\left[\mathbb{E}\left[(\theta_{t-\tau_{\mathrm{mix}}} - \theta^*)^\top \left(g_t(\theta_{t-\tau_{\mathrm{mix}}}) - \bar{g}(\theta_{t-\tau_{\mathrm{mix}}})\right) \mid \mathcal{F}_{t-\tau_{\mathrm{mix}}}\right]\right] \\
&\leq \mathbb{E}[\|\delta_{t-\tau_{\mathrm{mix}}}\| \cdot 2\alpha(\|\delta_{t-\tau_{\mathrm{mix}}}\| + c)] \\
&\leq \alpha\, \mathbb{E}\left[3\|\delta_{t-\tau_{\mathrm{mix}}}\|^2 + c^2\right] \leq \alpha\left(3B^2 + c^2\right).
\end{aligned}
$$

**Terms $I_3$ and $I_4$.** By Lemma C.2,

$$\|g_t(\theta_t) - g_t(\theta_{t-\tau_{\mathrm{mix}}})\| \leq 2\|\theta_t - \theta_{t-\tau_{\mathrm{mix}}}\| \leq 6\sum_{i=t-\tau_{\mathrm{mix}}+1}^{t} \alpha(\|\delta_i\| + c).$$

Hence

$$
\begin{aligned}
I_3 &\leq 6\sum_{i=t-\tau_{\mathrm{mix}}+1}^{t} \alpha\, \mathbb{E}[\|\delta_{t-\tau_{\mathrm{mix}}}\|(\|\delta_i\| + c)] \\
&\leq 6\sum_{i=t-\tau_{\mathrm{mix}}+1}^{t} \alpha\, \mathbb{E}[\|\delta_{t-\tau_{\mathrm{mix}}}\|\|\delta_i\| + \|\delta_{t-\tau_{\mathrm{mix}}}\|c] \\
&\leq \sum_{i=t-\tau_{\mathrm{mix}}+1}^{t} \alpha\, \mathbb{E}\left[6 \cdot \frac{\|\delta_{t-\tau_{\mathrm{mix}}}\|^2 + \|\delta_i\|^2}{2} + 6 \cdot \frac{\|\delta_{t-\tau_{\mathrm{mix}}}\|^2 + c^2}{2}\right] \\
&\leq \left(9B^2 + 3c^2\right)\sum_{i=t-\tau_{\mathrm{mix}}+1}^{t} \alpha.
\end{aligned}
$$

The same bound applies to $I_4$ (since $\bar{g}(\cdot)$ is also 2-Lipschitz in $\theta$ by Lemma C.2 and linearity of expectation).

Combining $I_1$–$I_4$,

$$\mathbb{E}\left[(\theta_t - \theta^*)^\top \left(g_t(\theta_t) - \bar{g}(\theta_t)\right)\right] \leq \alpha(3B^2 + c^2) + (42B^2 + 30c^2)\sum_{i=t-\tau_{\mathrm{mix}}+1}^{t} \alpha.$$

With constant stepsize, $\sum_{i=t-\tau_{\mathrm{mix}}+1}^{t} \alpha = \tau_{\mathrm{mix}}\alpha$, proving the first inequality in Lemma C.5.

**Step 2: the $f$-term.** Same as before, we can do the same decomposition so that

$$\mathbb{E}\left[(\theta_t - \theta^*)^\top \left(f_t(\theta_t) - \bar{f}(\theta_t)\right)\right] = I_1' + I_2' + I_3' + I_4'.$$

The only change is in $I_2'$: given $\mathcal{F}_{t-\tau_{\mathrm{mix}}}$, the joint law of $(s_t, \hat{s}_t)$ equals $P_{\tau_{\mathrm{mix}}}(\cdot|s_{t-\tau_{\mathrm{mix}}}) \otimes P_{\tau_{\mathrm{mix}}}(\cdot|\hat{s}_{t-\tau_{\mathrm{mix}}})$, whose $\ell_1$ distance to $\mu \otimes \mu$ is at most $\|P_{\tau_{\mathrm{mix}}}(\cdot|s_{t-\tau_{\mathrm{mix}}}) - \mu\|_1 + \|P_{\tau_{\mathrm{mix}}}(\cdot|\hat{s}_{t-\tau_{\mathrm{mix}}}) - \mu\|_1 \leq 2\alpha.$

Using the same analysis as function $g$, we conclude

$$\mathbb{E}\left[(\theta_t - \theta^*)^\top \left(f_t(\theta_t) - \bar{f}(\theta_t)\right)\right] \leq \alpha(3B^2 + c^2) + (21B^2 + 15c^2)\sum_{i=t-\tau_{\mathrm{mix}}+1}^{t} \alpha,$$

which proves the second inequality in Lemma C.5. $\qquad\square$

## D. Analysis of Double Chain Algorithm with Decaying Stepsize

In this section, we give a detailed proof of Theorem 4.3. We first state the full version of the theorem with all the constants:

**Theorem D.1** (Restatement of Theorem 4.3). *Let $a > 0$, $c_0 > 0$, and $\alpha_t = \frac{a}{(t+c_0)^\xi}$ with $\xi \in (0, 1]$. Denote*

$$\beta_1 := 2\mathbb{E}\left[\left(\|\theta_0\| + r_{\max} + 2\|\theta^*\|\right)^2\right] + 2\mathbb{E}\left[\|\theta_0 - \theta^*\|^2\right],$$
$$\beta_2 := 946\left(r_{\max} + 3\|\theta^*\|\right)^2,$$

*and*

$$L_1 := \max\left\{1, \frac{\log C + 1}{\log(1/\beta)}\right\}$$
$$\beta(T) := 4aL_1\beta_2\left(\log(T + c_0) - \log a + 1\right)$$

*Assume $c_0 \geq \max\{c_{0,1}(a, \xi), c_{0,2}(a, \xi)\}$, where $c_{0,1}, c_{0,2}$ are chosen so that Lemma D.2 holds.*

1. *If $\xi \in (0, 1)$ and $c_0 \geq \left(\frac{2\xi}{a\eta}\right)^{\frac{1}{1-\xi}}$, then*

$$\mathbb{E}\left[\|\delta_T\|^2\right]$$
$$\leq \beta_1 \exp\left(-\frac{\eta a}{1 - \xi}\left((T + c_0)^{1-\xi} - (\tau_{\mathrm{mix}} + c_0)^{1-\xi}\right)\right)$$
$$+ \frac{\beta(T)}{\eta(T + c_0)^\xi}.$$

2. *If $\xi = 1$ and $c_0 \geq \eta a$, then*

$$\mathbb{E}\left[\|\delta_T\|^2\right] \leq \beta_1 \left(\frac{\tau_{\mathrm{mix}} + c_0}{T + c_0}\right)^{\eta a} + \frac{\Gamma(T)}{(T + c_0)^q},$$

*where $q = \min\{1, \eta a\}$, $\tilde{\beta}(T) := 2L_1\beta_2\left(\log(T + c_0) - \log a + 1\right)$ and*

$$\Gamma(T) := \begin{cases} \dfrac{4a^2}{1 - \eta a}\tilde{\beta}(T), & 0 < a < 1/\eta, \\ 4a^2 \log(T + c_0)\,\tilde{\beta}(T), & a = 1/\eta, \\ \dfrac{4ea^2}{\eta a - 1}\tilde{\beta}(T), & a > 1/\eta. \end{cases}$$

Throughout, we use the same notations as in Appendix C.

### D.1. Proof of Theorem 4.3

*Proof of Theorem 4.3.* To control Markov noise, we need an upper bound on $\sum_{i=t-\tau_{\mathrm{mix}}}^{t-1} \alpha_i$. This is captured by the following lemma.

**Lemma D.2.** *There exist constants $c_{0,1} = c_{0,1}(a, \xi) > 0$ and $c_{0,2} = c_{0,2}(a, \xi) > 0$ such that if $c_0 \geq \max\{c_{0,1}, c_{0,2}\}$, then for all $t \in [\tau_{\mathrm{mix}}, T]$,*

1. $\displaystyle\sum_{i=t-\tau_{\mathrm{mix}}}^{t-1} \alpha_i \leq 2L_1\alpha_t\left(\log(1/\alpha_T) + 1\right);$

2. $\displaystyle\sum_{i=t-\tau_{\mathrm{mix}}}^{t-1} \alpha_i \leq \min\{1/12,\ \eta/2694\}.$

The proof of Lemma D.2 can be found in Section D.2.

Denote

$$\bar{g}(\theta_t) := \sum_{s_t, s_t'} \mu(s_t) P(s_t'|s_t) g(s_t, s_t', \theta_t) = \Phi^\top D(R + P\Phi\theta_t - \Phi\theta_t),$$

$$\bar{f}(\theta_t) := \sum_{s_t, \hat{s}_t} \mu(s_t)\mu(\hat{s}_t) f(s_t, \hat{s}_t, \theta_t) = -\Phi^\top \mu\mu^\top (R + \Phi\theta_t).$$

We will also need the following two lemmas.

**Lemma D.3.** Suppose $t_1 \le t_2$ and $\sum_{i=t_1}^{t_2-1} \alpha_i \le 1/12$. Then

1. $\|\theta_{t_2} - \theta_{t_1}\| \le 6\big(\|\theta_{t_1}\| + r_{\max} + 2\|\theta^*\|\big) \sum_{i=t_1}^{t_2-1} \alpha_i;$

2. $\|\theta_{t_2} - \theta_{t_1}\| \le 12\big(\|\theta_{t_2}\| + r_{\max} + 2\|\theta^*\|\big) \sum_{i=t_1}^{t_2-1} \alpha_i.$

**Lemma D.4.** Suppose $t \in [\tau_{\mathrm{mix}}, T]$ and $\sum_{i=t-\tau_{\mathrm{mix}}}^{t-1} \alpha_i \le 1/12$. Then

$$\mathbb{E}\Big[(\theta_t - \theta^*)^\top \big(g(s_t, s_t', \theta_t) - \bar{g}(\theta_t)\big)\Big] \le \Big(880\mathbb{E}\|\delta_t\|^2 + 304(r_{\max} + 3\|\theta^*\|)^2\Big) \sum_{i=t-\tau_{\mathrm{mix}}}^{t-1} \alpha_i,$$

$$\mathbb{E}\Big[(\theta_t - \theta^*)^\top \big(f(s_t, \hat{s}_t, \theta_t) - \bar{f}(\theta_t)\big)\Big] \le \Big(448\mathbb{E}\|\delta_t\|^2 + 160(r_{\max} + 3\|\theta^*\|)^2\Big) \sum_{i=t-\tau_{\mathrm{mix}}}^{t-1} \alpha_i.$$

The proof of Lemma D.3 can be found in Section D.3, and the proof of Lemma D.4 can be found in Section D.4.

Using Lemma D.4 and summing the two bounds, we have

$$2\alpha_t \mathbb{E}\left[\delta_t^\top \big(\bar{g}(\theta_t) + \bar{f}(\theta_t) - \big(g(s_t, s_t', \theta_t) + f(s_t, \hat{s}_t, \theta_t)\big)\big)\right]$$

$$\le 2\alpha_t \Big(1328\mathbb{E}\|\delta_t\|^2 + 464(r_{\max} + 3\|\theta^*\|)^2\Big) \sum_{i=t-\tau_{\mathrm{mix}}}^{t-1} \alpha_i.$$

Combining the basic expansion of $\|\delta_{t+1}\|^2$ in Appendix C with the bounds derived above, for any $t \in [\tau_{\mathrm{mix}}, T]$,

$$\mathbb{E}\|\delta_{t+1}\|^2 \le (1 - 2\eta\alpha_t)\mathbb{E}\|\delta_t\|^2 + 18\alpha_t^2\Big(\mathbb{E}\|\delta_t\|^2 + (r_{\max} + \|\theta^*\|)^2\Big)$$

$$+ 2\alpha_t\Big(1328\mathbb{E}\|\delta_t\|^2 + 464(r_{\max} + 3\|\theta^*\|)^2\Big) \sum_{i=t-\tau_{\mathrm{mix}}}^{t-1} \alpha_i.$$

Since $\alpha_t$ is nonincreasing and $\tau_{\mathrm{mix}} \ge 1$,

$$\sum_{i=t-\tau_{\mathrm{mix}}}^{t-1} \alpha_i \ge \alpha_{t-1} \ge \alpha_t \quad \Rightarrow \quad \alpha_t^2 \le \alpha_t \sum_{i=t-\tau_{\mathrm{mix}}}^{t-1} \alpha_i.$$

Moreover, $(r_{\max} + \|\theta^*\|)^2 \le (r_{\max} + 3\|\theta^*\|)^2$. Therefore,

$$\mathbb{E}\|\delta_{t+1}\|^2 \le (1 - 2\eta\alpha_t)\mathbb{E}\|\delta_t\|^2 + \alpha_t\Big(2674\,\mathbb{E}\|\delta_t\|^2 + 946(r_{\max} + 3\|\theta^*\|)^2\Big) \sum_{i=t-\tau_{\mathrm{mix}}}^{t-1} \alpha_i.$$

By Lemma D.2(2), we have $\sum_{i=t-\tau_{\mathrm{mix}}}^{t-1} \alpha_i \leq \eta/2694$, which implies $2674 \sum_{i=t-\tau_{\mathrm{mix}}}^{t-1} \alpha_i \leq \eta$. Hence,

$$\mathbb{E}\|\delta_{t+1}\|^2 \leq (1-\eta\alpha_t)\mathbb{E}\|\delta_t\|^2 + \beta_2 \, \alpha_t \sum_{i=t-\tau_{\mathrm{mix}}}^{t-1} \alpha_i.$$

Define $\hat{\alpha}_t = \alpha_t \sum_{i=t-\tau_{\mathrm{mix}}}^{t-1} \alpha_i$. Then

$$\mathbb{E}\|\delta_{t+1}\|^2 \leq (1-\eta\alpha_t)\mathbb{E}\|\delta_t\|^2 + \beta_2 \hat{\alpha}_t, \qquad \forall\, t \in [\tau_{\mathrm{mix}}, T].$$

Recursively applying the above inequality from $\tau_{\mathrm{mix}}$ to $T$, we obtain

$$\mathbb{E}\|\delta_T\|^2 \leq \mathbb{E}\|\delta_{\tau_{\mathrm{mix}}}\|^2 \prod_{j=\tau_{\mathrm{mix}}}^{T-1} (1-\eta\alpha_j) + \beta_2 \sum_{k=\tau_{\mathrm{mix}}}^{T-1} \hat{\alpha}_k \prod_{j=k+1}^{T-1} (1-\eta\alpha_j).$$

**Step 1: bound $\mathbb{E}\|\delta_{\tau_{\mathrm{mix}}}\|^2$.** By Lemma D.2(2) applied at $t = \tau_{\mathrm{mix}}$, $\sum_{i=0}^{\tau_{\mathrm{mix}}-1} \alpha_i \leq 1/12$. Then Lemma D.3 (with $t_1 = 0, t_2 = \tau_{\mathrm{mix}}$) yields

$$\|\theta_{\tau_{\mathrm{mix}}} - \theta_0\| \leq 12(\|\theta_0\| + r_{\mathrm{max}} + 2\|\theta^*\|) \sum_{i=0}^{\tau_{\mathrm{mix}}-1} \alpha_i \leq \|\theta_0\| + r_{\mathrm{max}} + 2\|\theta^*\|.$$

Therefore,

$$\mathbb{E}\|\delta_{\tau_{\mathrm{mix}}}\|^2 \leq 2\mathbb{E}\|\theta_{\tau_{\mathrm{mix}}} - \theta_0\|^2 + 2\mathbb{E}\|\theta_0 - \theta^*\|^2$$
$$\leq 2\mathbb{E}\Big[\big(\|\theta_0\| + r_{\mathrm{max}} + 2\|\theta^*\|\big)^2\Big] + 2\mathbb{E}\big[\|\theta_0 - \theta^*\|^2\big] = \beta_1.$$

**Step 2: bound the product term.** Let

$$I_1 := \prod_{j=\tau_{\mathrm{mix}}}^{T-1} (1-\eta\alpha_j).$$

Using $1 - x \leq e^{-x}$, we have

$$I_1 \leq \exp\left(-\eta \sum_{j=\tau_{\mathrm{mix}}}^{T-1} \alpha_j\right) = \exp\left(-\eta a \sum_{j=\tau_{\mathrm{mix}}}^{T-1} \frac{1}{(j+c_0)^\xi}\right) \leq \exp\left(-\eta a \int_{\tau_{\mathrm{mix}}}^{T} \frac{dx}{(x+c_0)^\xi}\right).$$

Thus,

$$I_1 \leq \begin{cases} \left(\dfrac{\tau_{\mathrm{mix}} + c_0}{T + c_0}\right)^{\eta a}, & \xi = 1, \\[2ex] \exp\!\left(-\dfrac{\eta a}{1-\xi}\Big((T+c_0)^{1-\xi} - (\tau_{\mathrm{mix}}+c_0)^{1-\xi}\Big)\right), & \xi \in (0,1). \end{cases}$$

**Step 3: bound the sum term.** Let

$$I_2 := \sum_{k=\tau_{\mathrm{mix}}}^{T-1} \hat{\alpha}_k \prod_{j=k+1}^{T-1} (1-\eta\alpha_j).$$

By Lemma D.2(1),

$$\hat{\alpha}_k = \alpha_k \sum_{i=k-\tau_{\mathrm{mix}}}^{k-1} \alpha_i \leq 2L_1 \alpha_k^2 \big(\log(1/\alpha_T) + 1\big), \qquad \forall\, k \in [\tau_{\mathrm{mix}}, T].$$

Hence,

$$\beta_2 I_2 \leq 2L_1 \beta_2 \big(\log(1/\alpha_T) + 1\big) \sum_{k=\tau_{\mathrm{mix}}}^{T-1} \alpha_k^2 \prod_{j=k+1}^{T-1} (1-\eta\alpha_j).$$

Since $\log(1/\alpha_T) = \xi \log(T + c_0) - \log a \le \log(T + c_0) - \log a$, we can define

$$\tilde{\beta}(T) := 2L_1\beta_2\big(\log(T + c_0) - \log a + 1\big)$$

so that

$$\beta_2 I_2 \le \tilde{\beta}(T) \cdot \underbrace{\sum_{k=\tau_{\mathrm{mix}}}^{T-1} \alpha_k^2 \prod_{j=k+1}^{T-1} (1 - \eta\alpha_j)}_{=:I_3}.$$

We next bound $I_3$ in two cases.

**Case 1: $\xi = 1$.** In this case $\alpha_k = \frac{a}{k+c_0}$, and we assume $c_0 \ge a\eta$. Using $1 - x \le e^{-x}$,

$$\prod_{j=k+1}^{T-1} \left(1 - \frac{\eta a}{j + c_0}\right) \le \exp\left(-\eta a \sum_{j=k+1}^{T-1} \frac{1}{j + c_0}\right) \le \left(\frac{k + 1 + c_0}{T + c_0}\right)^{\eta a}.$$

Therefore,

$$
\begin{aligned}
I_3 &= \sum_{k=\tau_{\mathrm{mix}}}^{T-1} \frac{a^2}{(k + c_0)^2} \prod_{j=k+1}^{T-1} \left(1 - \frac{\eta a}{j + c_0}\right) \\
&\le \sum_{k=\tau_{\mathrm{mix}}}^{T-1} \frac{a^2}{(k + c_0)^2} \left(\frac{k + 1 + c_0}{T + c_0}\right)^{\eta a} = \frac{a^2}{(T + c_0)^{\eta a}} \sum_{k=\tau_{\mathrm{mix}}}^{T-1} \left(\frac{k + 1 + c_0}{k + c_0}\right)^2 (k + 1 + c_0)^{\eta a - 2} \\
&\le \frac{4a^2}{(T + c_0)^{\eta a}} \sum_{k=\tau_{\mathrm{mix}}}^{T-1} (k + 1 + c_0)^{\eta a - 2}.
\end{aligned}
$$

Standard summation bounds give

$$
I_3 \le
\begin{cases}
\dfrac{4a^2}{1 - \eta a} \cdot \dfrac{1}{(T + c_0)^{\eta a}}, & \eta a < 1, \\[2ex]
4a^2 \dfrac{\log(T + c_0)}{T + c_0}, & \eta a = 1, \\[2ex]
\dfrac{4ea^2}{\eta a - 1} \cdot \dfrac{1}{T + c_0}, & \eta a > 1.
\end{cases}
$$

Define $q = \min\{1, \eta a\}$ and

$$
\Gamma(T) :=
\begin{cases}
\dfrac{4a^2}{1 - \eta a} \tilde{\beta}(T), & \eta a < 1, \\[2ex]
4a^2 \log(T + c_0)\, \tilde{\beta}(T), & \eta a = 1, \\[2ex]
\dfrac{4ea^2}{\eta a - 1} \tilde{\beta}(T), & \eta a > 1.
\end{cases}
$$

Then $\tilde{\beta}(T)I_3 \le \Gamma(T)/(T + c_0)^q$. Combining with the bound on $I_1$, we obtain the $\xi = 1$ claim.

**Case 2: $\xi \in (0, 1)$.** Consider the sequence $\{u_t\}_{t \ge \tau_{\mathrm{mix}}}$ defined by

$$u_{\tau_{\mathrm{mix}}} = 0, \qquad u_{t+1} = \left(1 - \frac{\eta a}{(t + c_0)^\xi}\right) u_t + \frac{a^2}{(t + c_0)^{2\xi}}.$$

One can check that $I_3 = u_T$. To bound $u_T$, we use the following lemma.

**Lemma D.5.** *Given a sequence $\{x_t\}_{t \geq \tau}$ and positive constants $c_0, c_1, c_2, \xi$, consider the recursion*

$$x_{t+1} = \left(1 - \frac{c_1 c_2}{(t+c_0)^\xi}\right) x_t + \frac{c_2^2}{(t+c_0)^{2\xi}},$$

*with initial condition $x_\tau \leq \frac{2c_2}{c_1} \frac{1}{(\tau+c_0)^\xi}$. Then $x_t \leq \frac{2c_2}{c_1} \frac{1}{(t+c_0)^\xi}$ for all $t \geq \tau$ if either:*

1. *$\xi = 1$ and $c_1 c_2 \geq 2$;*

2. *$\xi \in (0, 1)$ and $\tau \geq (2\xi/(c_1 c_2))^{1/(1-\xi)} - c_0$.*

The proof of Lemma D.5 can be found in Section D.5.

Applying Lemma D.5 with $c_1 = \eta$, $c_2 = a$, $\tau = \tau_{\mathrm{mix}}$, and noting that $u_{\tau_{\mathrm{mix}}} = 0 \leq \frac{2a}{\eta} \frac{1}{(\tau_{\mathrm{mix}}+c_0)^\xi}$, we get (under the condition $c_0 \geq (\frac{2\xi}{\eta a})^{1/(1-\xi)}$)

$$I_3 = u_T \leq \frac{2a}{\eta(T+c_0)^\xi}.$$

Therefore,

$$\tilde{\beta}(T) I_3 \leq \frac{2a\,\tilde{\beta}(T)}{\eta(T+c_0)^\xi} = \frac{\beta(T)}{\eta(T+c_0)^\xi},$$

where $\beta(T) = 4aL_1\beta_2(\log(T+c_0) - \log a + 1)$. Combining with the bound on $I_1$ proves the $\xi \in (0,1)$ claim. □

## D.2. Proof of Lemma D.2

*Proof of Lemma D.2.* By definition of $\tau_{\mathrm{mix}} = \tau_{\mathrm{mix}}(\alpha_T)$, we have $C\beta^{\tau_{\mathrm{mix}}} \leq \alpha_T$, which implies

$$\tau_{\mathrm{mix}} \leq \frac{\log C + \log(1/\alpha_T)}{\log(1/\beta)} \leq L_1\big(\log(1/\alpha_T) + 1\big), \qquad L_1 = \max\left\{1, \frac{\log C + 1}{\log(1/\beta)}\right\}.$$

Since $\alpha_t$ is nonincreasing,

$$\sum_{i=t-\tau_{\mathrm{mix}}}^{t-1} \alpha_i \leq \tau_{\mathrm{mix}} \alpha_{t-\tau_{\mathrm{mix}}} = \tau_{\mathrm{mix}} \alpha_t \left(\frac{t+c_0}{t-\tau_{\mathrm{mix}}+c_0}\right)^\xi.$$

Note that $\left(\frac{t+c_0}{t-\tau_{\mathrm{mix}}+c_0}\right)^\xi \to 1$ as $c_0 \to \infty$ uniformly over $t \in [\tau_{\mathrm{mix}}, T]$. Hence, there exists $c_{0,1}(a, \xi)$ large enough such that for all $t \in [\tau_{\mathrm{mix}}, T]$,

$$\left(\frac{t+c_0}{t-\tau_{\mathrm{mix}}+c_0}\right)^\xi \leq 2, \qquad \forall c_0 \geq c_{0,1}(a, \xi).$$

Using $\tau_{\mathrm{mix}} \leq L_1(\log(1/\alpha_T) + 1)$, we obtain

$$\sum_{i=t-\tau_{\mathrm{mix}}}^{t-1} \alpha_i \leq 2L_1\alpha_t\big(\log(1/\alpha_T) + 1\big), \qquad \forall t \in [\tau_{\mathrm{mix}}, T],$$

which proves part (1).

For part (2), since $\alpha_t \to 0$ as $c_0 \to \infty$ (for fixed $T$) and the right-hand side in part (1) is $O(\alpha_t \log(1/\alpha_T))$, there exists $c_{0,2}(a, \xi)$ large enough such that for all $c_0 \geq c_{0,2}(a, \xi)$,

$$2L_1\alpha_t\big(\log(1/\alpha_T) + 1\big) \leq \min\{1/12, \eta/2694\}, \qquad \forall t \in [\tau_{\mathrm{mix}}, T].$$

Combining with part (1) proves part (2). □

## D.3. Proof of Lemma D.3

*Proof of Lemma D.3.* By Lemma C.1,

$$\|\theta_{t+1}\| - \|\theta_t\| \le \|\theta_{t+1} - \theta_t\| \le \alpha_t \|f(s_t, \hat{s}_t, \theta_t) + g(s_t, s'_t, \theta_t)\| \le 3\alpha_t(\|\theta_t\| + r_{\max} + 2\|\theta^*\|).$$

Thus,

$$\|\theta_{t+1}\| + r_{\max} + 2\|\theta^*\| \le (1 + 3\alpha_t)(\|\theta_t\| + r_{\max} + 2\|\theta^*\|).$$

For any $t \in [t_1, t_2]$,

$$\|\theta_t\| + r_{\max} + 2\|\theta^*\| \le \prod_{i=t_1}^{t-1}(1 + 3\alpha_i)(\|\theta_{t_1}\| + r_{\max} + 2\|\theta^*\|)$$

$$\le \exp\left(3\sum_{i=t_1}^{t-1}\alpha_i\right)(\|\theta_{t_1}\| + r_{\max} + 2\|\theta^*\|)$$

$$\le \left(1 + 6\sum_{i=t_1}^{t-1}\alpha_i\right)(\|\theta_{t_1}\| + r_{\max} + 2\|\theta^*\|),$$

where the last step uses $e^x \le 1 + 2x$ for $x \le 1/2$, and here $x = 3\sum_{i=t_1}^{t-1}\alpha_i \le 3 \cdot (1/12) = 1/4$. In particular, since $\sum_{i=t_1}^{t_2-1}\alpha_i \le 1/12$,

$$\|\theta_t\| + r_{\max} + 2\|\theta^*\| \le 2(\|\theta_{t_1}\| + r_{\max} + 2\|\theta^*\|).$$

Therefore,

$$\|\theta_{t_2} - \theta_{t_1}\| \le \sum_{i=t_1}^{t_2-1}\|\theta_{i+1} - \theta_i\| \le \sum_{i=t_1}^{t_2-1}3\alpha_i(\|\theta_i\| + r_{\max} + 2\|\theta^*\|)$$

$$\le 6(\|\theta_{t_1}\| + r_{\max} + 2\|\theta^*\|)\sum_{i=t_1}^{t_2-1}\alpha_i,$$

which proves part (1). For part (2), using $\|\theta_{t_1}\| \le \|\theta_{t_2}\| + \|\theta_{t_2} - \theta_{t_1}\|$,

$$\|\theta_{t_2} - \theta_{t_1}\| \le 6(\|\theta_{t_1}\| + r_{\max} + 2\|\theta^*\|)\sum_{i=t_1}^{t_2-1}\alpha_i$$

$$\le 6(\|\theta_{t_2}\| + \|\theta_{t_2} - \theta_{t_1}\| + r_{\max} + 2\|\theta^*\|)\sum_{i=t_1}^{t_2-1}\alpha_i$$

$$\le \frac{1}{2}\|\theta_{t_2} - \theta_{t_1}\| + 6(\|\theta_{t_2}\| + r_{\max} + 2\|\theta^*\|)\sum_{i=t_1}^{t_2-1}\alpha_i,$$

where we used $\sum_{i=t_1}^{t_2-1}\alpha_i \le 1/12$. Rearranging gives

$$\|\theta_{t_2} - \theta_{t_1}\| \le 12(\|\theta_{t_2}\| + r_{\max} + 2\|\theta^*\|)\sum_{i=t_1}^{t_2-1}\alpha_i,$$

which proves part (2). □

## D.4. Proof of Lemma D.4

*Proof of Lemma D.4.* For simplicity, denote

$$g_t(\theta) = g(s_t, s'_t, \theta), \qquad f_t(\theta) = f(s_t, \hat{s}_t, \theta).$$

Throughout this proof we fix $t \in [\tau_{\mathrm{mix}}, T]$ and assume $\sum_{i=t-\tau_{\mathrm{mix}}}^{t-1} \alpha_i \le 1/12$.

By Lemma D.3 (part (2)),

$$\|\theta_t - \theta_{t-\tau_{\mathrm{mix}}}\| \le 12\big(\|\theta_t\| + r_{\max} + 2\|\theta^*\|\big) \sum_{i=t-\tau_{\mathrm{mix}}}^{t-1} \alpha_i$$

$$\le 12\big(\|\delta_t\| + r_{\max} + 3\|\theta^*\|\big) \sum_{i=t-\tau_{\mathrm{mix}}}^{t-1} \alpha_i.$$

**Step 1: bound the noise term for $g$.** Decompose

$$\mathbb{E}\Big[(\theta_t - \theta^*)^\top \big(g_t(\theta_t) - \bar{g}(\theta_t)\big)\Big]$$

$$= \underbrace{\mathbb{E}\Big[(\theta_t - \theta_{t-\tau_{\mathrm{mix}}})^\top \big(g_t(\theta_t) - \bar{g}(\theta_t)\big)\Big]}_{I_1} + \underbrace{\mathbb{E}\Big[(\theta_{t-\tau_{\mathrm{mix}}} - \theta^*)^\top \big(g_t(\theta_{t-\tau_{\mathrm{mix}}}) - \bar{g}(\theta_{t-\tau_{\mathrm{mix}}})\big)\Big]}_{I_2}$$

$$+ \underbrace{\mathbb{E}\Big[(\theta_{t-\tau_{\mathrm{mix}}} - \theta^*)^\top \big(g_t(\theta_t) - g_t(\theta_{t-\tau_{\mathrm{mix}}})\big)\Big]}_{I_3} + \underbrace{\mathbb{E}\Big[(\theta_{t-\tau_{\mathrm{mix}}} - \theta^*)^\top \big(\bar{g}(\theta_{t-\tau_{\mathrm{mix}}}) - \bar{g}(\theta_t)\big)\Big]}_{I_4}.$$

**Term $I_1$.** By Lemma C.1, $\|g_t(\theta_t)\| \le 2(\|\delta_t\| + r_{\max} + \|\theta^*\|)$, hence $\|g_t(\theta_t) - \bar{g}(\theta_t)\| \le 4(\|\delta_t\| + r_{\max} + \|\theta^*\|)$. Thus,

$$I_1 \le \mathbb{E}\Big[\|\theta_t - \theta_{t-\tau_{\mathrm{mix}}}\| \cdot \|g_t(\theta_t) - \bar{g}(\theta_t)\|\Big]$$

$$\le \mathbb{E}\Big[12 \sum_{i=t-\tau_{\mathrm{mix}}}^{t-1} \alpha_i(\|\delta_t\| + r_{\max} + 3\|\theta^*\|) \cdot 4(\|\delta_t\| + r_{\max} + \|\theta^*\|)\Big]$$

$$\le 48 \sum_{i=t-\tau_{\mathrm{mix}}}^{t-1} \alpha_i \, \mathbb{E}\Big[(\|\delta_t\| + r_{\max} + 3\|\theta^*\|)^2\Big]$$

$$\le 96 \sum_{i=t-\tau_{\mathrm{mix}}}^{t-1} \alpha_i \Big(\mathbb{E}\|\delta_t\|^2 + (r_{\max} + 3\|\theta^*\|)^2\Big).$$

**Term $I_2$.** First note that

$$\|\delta_{t-\tau_{\mathrm{mix}}}\| \le \|\delta_t\| + \|\theta_t - \theta_{t-\tau_{\mathrm{mix}}}\|$$

$$\le \|\delta_t\| + 12(\|\delta_t\| + r_{\max} + 3\|\theta^*\|) \sum_{i=t-\tau_{\mathrm{mix}}}^{t-1} \alpha_i$$

$$\le 2\|\delta_t\| + r_{\max} + 3\|\theta^*\|,$$

where we used $\sum_{i=t-\tau_{\mathrm{mix}}}^{t-1} \alpha_i \le 1/12$.

Next, by Eq.(4)

$$\sup_s \|P_{\tau_{\mathrm{mix}}}(\cdot \mid s) - \mu\|_1 \le C\beta^{\tau_{\mathrm{mix}}} \le \alpha_T \le \alpha_t.$$

Therefore,

$$\left\|\mathbb{E}\big[g_t(\theta_{t-\tau_{\mathrm{mix}}}) - \bar{g}(\theta_{t-\tau_{\mathrm{mix}}}) \mid \mathcal{F}_{t-\tau_{\mathrm{mix}}}\big]\right\| \le \sup_s \|P_{\tau_{\mathrm{mix}}}(\cdot \mid s) - \mu\|_1 \cdot \sup_{s,s'} \|g(s, s', \theta_{t-\tau_{\mathrm{mix}}})\|$$

$$\le \alpha_t \cdot 2(\|\delta_{t-\tau_{\mathrm{mix}}}\| + r_{\max} + \|\theta^*\|)$$

$$\le 4\alpha_t(\|\delta_t\| + r_{\max} + 2\|\theta^*\|).$$

Hence,

$$
\begin{aligned}
I_2 &\leq \mathbb{E}\Big[\|\delta_{t-\tau_{\text{mix}}}\| \cdot \Big\|\mathbb{E}\big[g_t(\theta_{t-\tau_{\text{mix}}}) - \bar{g}(\theta_{t-\tau_{\text{mix}}}) \mid \mathcal{F}_{t-\tau_{\text{mix}}}\big]\Big\|\Big] \\
&\leq \mathbb{E}\Big[(2\|\delta_t\| + r_{\max} + 3\|\theta^*\|) \cdot 4\alpha_t(\|\delta_t\| + r_{\max} + 2\|\theta^*\|)\Big] \\
&\leq 8\alpha_t \,\mathbb{E}\Big[(\|\delta_t\| + r_{\max} + 2\|\theta^*\|)^2\Big] \\
&\leq 16\alpha_t\Big(\mathbb{E}\|\delta_t\|^2 + (r_{\max} + 2\|\theta^*\|)^2\Big).
\end{aligned}
$$

**Terms $I_3$ and $I_4$.** By Lemma C.2, $g(\cdot)$ is 2-Lipschitz in $\theta$, and $\bar{g}(\cdot)$ is also 2-Lipschitz. Thus,

$$
\begin{aligned}
I_3 &\leq \mathbb{E}\Big[\|\delta_{t-\tau_{\text{mix}}}\| \cdot \|g_t(\theta_t) - g_t(\theta_{t-\tau_{\text{mix}}})\|\Big] \leq 2\mathbb{E}\Big[\|\delta_{t-\tau_{\text{mix}}}\| \cdot \|\theta_t - \theta_{t-\tau_{\text{mix}}}\|\Big], \\
I_4 &\leq \mathbb{E}\Big[\|\delta_{t-\tau_{\text{mix}}}\| \cdot \|\bar{g}(\theta_t) - \bar{g}(\theta_{t-\tau_{\text{mix}}})\|\Big] \leq 2\mathbb{E}\Big[\|\delta_{t-\tau_{\text{mix}}}\| \cdot \|\theta_t - \theta_{t-\tau_{\text{mix}}}\|\Big].
\end{aligned}
$$

Using the bounds on $\|\delta_{t-\tau_{\text{mix}}}\|$ and $\|\theta_t - \theta_{t-\tau_{\text{mix}}}\|$,

$$
\begin{aligned}
I_3 &\leq \mathbb{E}\Big[(2\|\delta_t\| + r_{\max} + 3\|\theta^*\|) \cdot 24(\|\delta_t\| + r_{\max} + 3\|\theta^*\|) \sum_{i=t-\tau_{\text{mix}}}^{t-1} \alpha_i\Big] \\
&\leq 48 \sum_{i=t-\tau_{\text{mix}}}^{t-1} \alpha_i \,\mathbb{E}\Big[(2\|\delta_t\| + r_{\max} + 3\|\theta^*\|)^2\Big] \\
&\leq 96 \sum_{i=t-\tau_{\text{mix}}}^{t-1} \alpha_i \Big(4\mathbb{E}\|\delta_t\|^2 + (r_{\max} + 3\|\theta^*\|)^2\Big),
\end{aligned}
$$

and the same bound holds for $I_4$.

Combining $I_1$–$I_4$, we get

$$
\begin{aligned}
\mathbb{E}\Big[(\theta_t - \theta^*)^\top (g_t(\theta_t) - \bar{g}(\theta_t))\Big] &\leq 16\alpha_t\Big(\mathbb{E}\|\delta_t\|^2 + (r_{\max} + 2\|\theta^*\|)^2\Big) \\
&\quad + \Big(864\mathbb{E}\|\delta_t\|^2 + 288(r_{\max} + 3\|\theta^*\|)^2\Big) \sum_{i=t-\tau_{\text{mix}}}^{t-1} \alpha_i.
\end{aligned}
$$

Since $\alpha_t \leq \sum_{i=t-\tau_{\text{mix}}}^{t-1} \alpha_i$, the above implies

$$
\mathbb{E}\Big[(\theta_t - \theta^*)^\top (g_t(\theta_t) - \bar{g}(\theta_t))\Big] \leq \Big(880\mathbb{E}\|\delta_t\|^2 + 304(r_{\max} + 3\|\theta^*\|)^2\Big) \sum_{i=t-\tau_{\text{mix}}}^{t-1} \alpha_i.
$$

**Step 2: bound the noise term for $f$.** Repeat the same decomposition for $f$:

$$
\begin{aligned}
&\mathbb{E}\Big[(\theta_t - \theta^*)^\top \big(f_t(\theta_t) - \bar{f}(\theta_t)\big)\Big] \\
&= \underbrace{\mathbb{E}\Big[(\theta_t - \theta_{t-\tau_{\text{mix}}})^\top \big(f_t(\theta_t) - \bar{f}(\theta_t)\big)\Big]}_{I_1'} + \underbrace{\mathbb{E}\Big[(\theta_{t-\tau_{\text{mix}}} - \theta^*)^\top \big(f_t(\theta_{t-\tau_{\text{mix}}}) - \bar{f}(\theta_{t-\tau_{\text{mix}}})\big)\Big]}_{I_2'} \\
&\quad + \underbrace{\mathbb{E}\Big[(\theta_{t-\tau_{\text{mix}}} - \theta^*)^\top \big(f_t(\theta_t) - f_t(\theta_{t-\tau_{\text{mix}}})\big)\Big]}_{I_3'} + \underbrace{\mathbb{E}\Big[(\theta_{t-\tau_{\text{mix}}} - \theta^*)^\top \big(\bar{f}(\theta_{t-\tau_{\text{mix}}}) - \bar{f}(\theta_t)\big)\Big]}_{I_4'}.
\end{aligned}
$$

**Term $I_1'$.** By Lemma C.1, $\|f_t(\theta_t)\| \le \|\delta_t\| + r_{\max} + \|\theta^*\|$, hence $\|f_t(\theta_t) - \bar{f}(\theta_t)\| \le 2(\|\delta_t\| + r_{\max} + \|\theta^*\|)$. Thus,

$$
\begin{aligned}
I_1' &\le \mathbb{E}\Big[\|\theta_t - \theta_{t-\tau_{\mathrm{mix}}}\| \cdot \|f_t(\theta_t) - \bar{f}(\theta_t)\|\Big] \\
&\le \mathbb{E}\Big[12 \sum_{i=t-\tau_{\mathrm{mix}}}^{t-1} \alpha_i(\|\delta_t\| + r_{\max} + 3\|\theta^*\|) \cdot 2(\|\delta_t\| + r_{\max} + \|\theta^*\|)\Big] \\
&\le 48 \sum_{i=t-\tau_{\mathrm{mix}}}^{t-1} \alpha_i \Big(\mathbb{E}\|\delta_t\|^2 + (r_{\max} + \|\theta^*\|)^2\Big).
\end{aligned}
$$

**Term $I_2'$.** By Eq.(4) (for both chains $s_t$ and $\hat{s}_t$),

$$
\sup_s \|P_{\tau_{\mathrm{mix}}}(\cdot \mid s) - \mu\|_1 \le \alpha_t, \qquad \sup_{\hat{s}} \|P_{\tau_{\mathrm{mix}}}(\cdot \mid \hat{s}) - \mu\|_1 \le \alpha_t.
$$

Hence,

$$
\begin{aligned}
\Big\|\mathbb{E}\big[f_t(\theta_{t-\tau_{\mathrm{mix}}}) - \bar{f}(\theta_{t-\tau_{\mathrm{mix}}}) \mid \mathcal{F}_{t-\tau_{\mathrm{mix}}}\big]\Big\| &\le \Big(\sup_s \|P_{\tau_{\mathrm{mix}}}(\cdot \mid s) - \mu\|_1 + \sup_{\hat{s}} \|P_{\tau_{\mathrm{mix}}}(\cdot \mid \hat{s}) - \mu\|_1\Big) \cdot \sup_{s,\hat{s}} \|f(s, \hat{s}, \theta_{t-\tau_{\mathrm{mix}}})\| \\
&\le 2\alpha_t \cdot (\|\delta_{t-\tau_{\mathrm{mix}}}\| + r_{\max} + \|\theta^*\|) \\
&\le 4\alpha_t(\|\delta_t\| + r_{\max} + 2\|\theta^*\|).
\end{aligned}
$$

Therefore,

$$
\begin{aligned}
I_2' &\le \mathbb{E}\Big[\|\delta_{t-\tau_{\mathrm{mix}}}\| \cdot \Big\|\mathbb{E}\big[f_t(\theta_{t-\tau_{\mathrm{mix}}}) - \bar{f}(\theta_{t-\tau_{\mathrm{mix}}}) \mid \mathcal{F}_{t-\tau_{\mathrm{mix}}}\big]\Big\|\Big] \\
&\le 16\alpha_t \Big(\mathbb{E}\|\delta_t\|^2 + (r_{\max} + 2\|\theta^*\|)^2\Big),
\end{aligned}
$$

where we used the same bound $\|\delta_{t-\tau_{\mathrm{mix}}}\| \le 2\|\delta_t\| + r_{\max} + 3\|\theta^*\|$.

**Terms $I_3'$ and $I_4'$.** By Lemma C.2, $f(\cdot)$ is 1-Lipschitz in $\theta$, and $\bar{f}(\cdot)$ is also 1-Lipschitz, hence

$$
I_3' \le \mathbb{E}\big[\|\delta_{t-\tau_{\mathrm{mix}}}\| \cdot \|\theta_t - \theta_{t-\tau_{\mathrm{mix}}}\|\big], \qquad I_4' \le \mathbb{E}\big[\|\delta_{t-\tau_{\mathrm{mix}}}\| \cdot \|\theta_t - \theta_{t-\tau_{\mathrm{mix}}}\|\big].
$$

Thus,

$$
\begin{aligned}
I_3' &\le \mathbb{E}\Big[(2\|\delta_t\| + r_{\max} + 3\|\theta^*\|) \cdot 12(\|\delta_t\| + r_{\max} + 3\|\theta^*\|) \sum_{i=t-\tau_{\mathrm{mix}}}^{t-1} \alpha_i\Big] \\
&\le 24 \sum_{i=t-\tau_{\mathrm{mix}}}^{t-1} \alpha_i \, \mathbb{E}\Big[(2\|\delta_t\| + r_{\max} + 3\|\theta^*\|)^2\Big] \\
&\le 48 \sum_{i=t-\tau_{\mathrm{mix}}}^{t-1} \alpha_i \Big(4\mathbb{E}\|\delta_t\|^2 + (r_{\max} + 3\|\theta^*\|)^2\Big),
\end{aligned}
$$

and the same bound holds for $I_4'$.

Combining $I_1'$–$I_4'$, we obtain

$$
\begin{aligned}
\mathbb{E}\Big[(\theta_t - \theta^*)^\top (f_t(\theta_t) - \bar{f}(\theta_t))\Big] &\le 16\alpha_t \Big(\mathbb{E}\|\delta_t\|^2 + (r_{\max} + 2\|\theta^*\|)^2\Big) \\
&\quad + \Big(432\mathbb{E}\|\delta_t\|^2 + 144(r_{\max} + 3\|\theta^*\|)^2\Big) \sum_{i=t-\tau_{\mathrm{mix}}}^{t-1} \alpha_i.
\end{aligned}
$$

Using $\alpha_t \le \sum_{i=t-\tau_{\mathrm{mix}}}^{t-1} \alpha_i$ gives the desired bound for $f$, and this completes the proof of Lemma D.4. $\qquad\square$

## D.5. Proof of Lemma D.5

*Proof of Lemma D.5.* We use induction. The base case is assumed in the statement. Now suppose $x_t \leq \frac{2c_2}{c_1} \frac{1}{(t+c_0)^\xi}$. Then

$$
\begin{aligned}
\frac{2c_2}{c_1} \frac{1}{(t+1+c_0)^\xi} - x_{t+1} &= \frac{2c_2}{c_1} \frac{1}{(t+1+c_0)^\xi} - \left(1 - \frac{c_1 c_2}{(t+c_0)^\xi}\right) x_t - \frac{c_2^2}{(t+c_0)^{2\xi}} \\
&\geq \frac{2c_2}{c_1} \frac{1}{(t+1+c_0)^\xi} - \left(1 - \frac{c_1 c_2}{(t+c_0)^\xi}\right) \frac{2c_2}{c_1} \frac{1}{(t+c_0)^\xi} - \frac{c_2^2}{(t+c_0)^{2\xi}} \\
&= \frac{2c_2}{c_1} \left[\frac{1}{(t+1+c_0)^\xi} - \frac{1}{(t+c_0)^\xi} + \frac{c_1 c_2}{2} \frac{1}{(t+c_0)^{2\xi}}\right] \\
&= \frac{2c_2}{c_1} \frac{1}{(t+c_0)^{2\xi}} \left[\frac{c_1 c_2}{2} - (t+c_0)^\xi \left(1 - \left(\frac{t+c_0}{t+1+c_0}\right)^\xi\right)\right].
\end{aligned}
$$

Observe that

$$
\left(\frac{t+c_0}{t+1+c_0}\right)^\xi = \left(1 + \frac{1}{t+c_0}\right)^{-\xi} \geq \exp\left(-\frac{\xi}{t+c_0}\right) \geq 1 - \frac{\xi}{t+c_0},
$$

where we used $e^x \geq 1 + x$. Hence,

$$
(t+c_0)^\xi \left(1 - \left(\frac{t+c_0}{t+1+c_0}\right)^\xi\right) \leq (t+c_0)^\xi \cdot \frac{\xi}{t+c_0} = \frac{\xi}{(t+c_0)^{1-\xi}}.
$$

Therefore,

$$
\frac{2c_2}{c_1} \frac{1}{(t+1+c_0)^\xi} - x_{t+1} \geq \frac{2c_2}{c_1} \frac{1}{(t+c_0)^{2\xi}} \left(\frac{c_1 c_2}{2} - \frac{\xi}{(t+c_0)^{1-\xi}}\right).
$$

The last term is nonnegative under either condition in the lemma statement, which yields $x_{t+1} \leq \frac{2c_2}{c_1} \frac{1}{(t+1+c_0)^\xi}$. This completes the induction. $\qquad\square$

# E. Analysis of Single Markov Chain Case with Constant Step-size

Recall the single-chain update components in Eq. (17):

$$f(s,w) := \phi(s) - w, \qquad g(s,s',w,\theta) := \left(r_s + \phi(s')^\top \theta - \phi(s)^\top \theta\right)\phi(s) - \left(r_s + \phi(s)^\top \theta\right)w.$$

Under stationarity $(s,s') \sim (\mu, P)$, define the mean fields

$$\bar{f}(w) := \mathbb{E}_{s\sim\mu}[f(s,w)] = \Phi^\top \mu - w = w^* - w, \tag{23}$$

$$\bar{g}(w,\theta) := \mathbb{E}_{s\sim\mu,\ s'\sim P(\cdot|s)}[g(s,s',w,\theta)] = \Phi^\top D\left(R + P\Phi\theta - \Phi\theta\right) - \mu^\top\left(R + \Phi\theta\right)w. \tag{24}$$

The fixed points are $w^* = \Phi^\top\mu$, and $\theta^*$ satisfies

$$\bar{g}(w^*,\theta^*) = 0. \tag{25}$$

First, we introduce some useful basic properties.

**Lemma E.1.** *For all $t \geq 0$, $\|w_t\| \leq 1$ and $\|\delta_t^w\| \leq 2$.*

*Proof of Lemma E.1.* The update $w_{t+1} = (1-\beta)w_t + \beta\phi(s_t)$ implies

$$\|w_{t+1}\| \leq (1-\beta)\|w_t\| + \beta\|\phi(s_t)\| \leq (1-\beta)\|w_t\| + \beta.$$

Since $\|w_0\| \leq 1$, induction gives $\|w_t\| \leq 1$ for all $t$. Also $\|w^*\| = \|\mathbb{E}_{s\sim\mu}[\phi(s)]\| \leq \mathbb{E}\|\phi(s)\| \leq 1$, hence $\|\delta_t^w\| \leq 2$. $\quad\square$

**Lemma E.2.** *For all $t \geq 0$,*

$$\|f(s_t,w_t)\| \leq 2, \qquad \|g(s_t,s'_t,w_t,\theta_t)\| \leq 2r_{\max} + 6R_\theta = 2(r_{\max} + 3R_\theta).$$

*Proof of Lemma E.2.* For $f(s,w) = \phi(s) - w$, $\|f(s_t,w_t)\| \leq \|\phi(s_t)\| + \|w_t\| \leq 2$. For $g$, using $\|\phi(\cdot)\| \leq 1$ and $\|w_t\| \leq 1$,

$$\begin{aligned}
\|g(s_t,s'_t,w_t,\theta_t)\| &\leq \left|r_{s_t} + (\phi(s'_t) - \phi(s_t))^\top\theta_t\right| \cdot \|\phi(s_t)\| + \left|r_{s_t} + \phi(s_t)^\top\theta_t\right| \cdot \|w_t\| \\
&\leq \left(r_{\max} + 2\|\theta_t\|\right) + \left(r_{\max} + \|\theta_t\|\right) \leq 2r_{\max} + 3\|\theta_t\| \leq 2r_{\max} + 6R_\theta.
\end{aligned}$$

$\quad\square$

**Lemma E.3.** *For all $s, s'$ and all $w, w_1, w_2, \theta, \theta_1, \theta_2$,*

$$\|f(s,w_1) - f(s,w_2)\| \leq \|w_1 - w_2\|,$$

$$\|g(s,s',w,\theta_1) - g(s,s',w,\theta_2)\| \leq 3\|\theta_1 - \theta_2\|,$$

*and*

$$\|g(s,s',w_1,\theta) - g(s,s',w_2,\theta)\| \leq (r_{\max} + 2R_\theta)\|w_1 - w_2\|.$$

*Proof of Lemma E.3.* The $f$ claim is immediate: $\|f(s,w_1) - f(s,w_2)\| = \|w_1 - w_2\|$.

For $\theta$,

$$g(s,s',w,\theta_1) - g(s,s',w,\theta_2) = \left((\phi(s') - \phi(s))^\top(\theta_1 - \theta_2)\right)\phi(s) - \left(\phi(s)^\top(\theta_1 - \theta_2)\right)w,$$

so

$$\|g(\cdot,\theta_1) - g(\cdot,\theta_2)\| \leq \|\phi(s') - \phi(s)\| \cdot \|\theta_1 - \theta_2\| \cdot \|\phi(s)\| + \|\phi(s)\| \cdot \|\theta_1 - \theta_2\| \cdot \|w\| \leq (2+1)\|\theta_1 - \theta_2\|.$$

For $w$,

$$g(s,s',w_1,\theta) - g(s,s',w_2,\theta) = -(r_s + \phi(s)^\top\theta)(w_1 - w_2),$$

hence $\|g(\cdot,w_1) - g(\cdot,w_2)\| \leq (r_{\max} + \|\theta\|)\|w_1 - w_2\| \leq (r_{\max} + 2R_\theta)\|w_1 - w_2\|.$ $\quad\square$

### E.1. Proof of Theorem 4.4

We first restate the theorem with all the constants.

**Theorem E.4** (Full statement of Theorem 4.4)**.** *Consider the single-chain algorithm in Eq. (17) with Markov sampling. Assume constant stepsizes $\alpha_t \equiv \alpha > 0$, $\beta_t \equiv \beta > 0$ and $\rho_0 := \beta/\alpha \leq 1$. Let $\lambda > 0$ satisfy $0 < \lambda^2 < \frac{2\eta}{r_{\max} + 2R_\theta}$ and define*

$$
\begin{aligned}
\zeta &:= \eta - \frac{\lambda^2(r_{\max} + 2R_\theta)}{2} > 0, \\
\kappa &:= 1 - 2\alpha\zeta \in (0,1), \\
G_1 &:= e^{-2\rho_0\alpha}.
\end{aligned}
$$

*Then for any $t \geq \tau_{\mathrm{mix}}$,*

$$
\begin{aligned}
\mathbb{E}\|\delta_t^\theta\|^2 \leq{}& \frac{4\alpha(r_{\max} + 2R_\theta)\,(t - \tau_{\mathrm{mix}})\,\max\{\kappa, G_1\}^{t - \tau_{\mathrm{mix}}}}{\lambda^2} \\
&+ 4R_\theta^2\,\kappa^{t - \tau_{\mathrm{mix}}} + \frac{\alpha\,G_{\mathrm{const}}}{2\zeta},
\end{aligned}
$$

*where*

$$
\begin{aligned}
G_{\mathrm{const}} :={}& \frac{(16\tau_{\mathrm{mix}} + 6)(r_{\max} + 2R_\theta)}{\lambda^2} \\
&+ (88\tau_{\mathrm{mix}} + 4)\,(r_{\max} + 3R_\theta)^2.
\end{aligned}
$$

*Proof of Theorem 4.4.* By non-expansiveness of projection,

$$
\|\delta_{t+1}^\theta\|^2 \leq \|\delta_t^\theta + \alpha\,g(s_t, s_t', w_t, \theta_t)\|^2. \tag{26}
$$

Taking expectation and expanding,

$$
\begin{aligned}
\mathbb{E}\|\delta_{t+1}^\theta\|^2 &\leq \mathbb{E}\|\delta_t^\theta\|^2 + 2\alpha\,\mathbb{E}\Big[\delta_t^{\theta\top} g(s_t, s_t', w_t, \theta_t)\Big] + \alpha^2\mathbb{E}\|g(s_t, s_t', w_t, \theta_t)\|^2 \\
&= \mathbb{E}\|\delta_t^\theta\|^2 + 2\alpha\,\mathbb{E}\Big[\delta_t^{\theta\top}\big(g(s_t, s_t', w_t, \theta_t) - \bar{g}(w_t, \theta_t)\big)\Big] + 2\alpha\,\mathbb{E}\Big[\delta_t^{\theta\top}\bar{g}(w_t, \theta_t)\Big] + \alpha^2\mathbb{E}\|g(s_t, s_t', w_t, \theta_t)\|^2.
\end{aligned} \tag{27}
$$

**Step 1: Markov-noise term.**

**Lemma E.5.** *For any $t \geq \tau_{\mathrm{mix}}$,*

$$
\mathbb{E}\Big[\delta_t^{\theta\top}\big(g(s_t, s_t', w_t, \theta_t) - \bar{g}(w_t, \theta_t)\big)\Big] \leq 44\,\alpha\,\tau_{\mathrm{mix}}\,(r_{\max} + 3R_\theta)^2.
$$

The proof of this lemma can be found in Section E.2.

By Lemma E.5, for $t \geq \tau_{\mathrm{mix}}$,

$$
2\alpha\,\mathbb{E}\Big[\delta_t^{\theta\top}\big(g(s_t, s_t', w_t, \theta_t) - \bar{g}(w_t, \theta_t)\big)\Big] \leq 88\,\alpha^2\,\tau_{\mathrm{mix}}\,(r_{\max} + 3R_\theta)^2. \tag{28}
$$

**Step 2: Mean-drift term.** Using $\bar{g}(w^*, \theta^*) = 0$,

$$
\delta_t^{\theta\top}\bar{g}(w_t, \theta_t) = -\|\Phi\delta_t^\theta\|_{\mathrm{Dir}}^2 - (\mu^\top\Phi\delta_t^\theta)^2 - \big(\mu^\top(R + \Phi\theta_t)\big)(\delta_t^{\theta\top}\delta_t^w).
$$

By Eq.(8) and Young's inequality, for any $\lambda > 0$,

$$
\big|\mu^\top(R + \Phi\theta_t)\big|\,\big|\delta_t^{\theta\top}\delta_t^w\big| \leq (r_{\max} + 2R_\theta)\cdot\frac{1}{2}\Big(\lambda^2\|\delta_t^\theta\|^2 + \|\delta_t^w\|^2/\lambda^2\Big).
$$

Therefore, with $\zeta = \eta - \lambda^2(r_{\max} + 2R_\theta)/2 > 0$,

$$2\alpha \, \mathbb{E}\big[\delta_t^{\theta\top} \bar{g}(w_t, \theta_t)\big] \leq -2\alpha\zeta \, \mathbb{E}\|\delta_t^\theta\|^2 + \alpha \, \frac{r_{\max} + 2R_\theta}{\lambda^2} \, \mathbb{E}\|\delta_t^w\|^2. \tag{29}$$

**Step 3: Second-moment term.** By Lemma E.2,

$$\alpha^2 \mathbb{E}\|g\|^2 \leq 4\alpha^2 \, (r_{\max} + 3R_\theta)^2. \tag{30}$$

Combining (27)–(30), for $t \geq \tau_{\mathrm{mix}}$,

$$\mathbb{E}\|\delta_{t+1}^\theta\|^2 \leq (1 - 2\alpha\zeta) \, \mathbb{E}\|\delta_t^\theta\|^2 + \alpha \, \frac{r_{\max} + 2R_\theta}{\lambda^2} \, \mathbb{E}\|\delta_t^w\|^2 + \alpha^2 (88\tau_{\mathrm{mix}} + 4)(r_{\max} + 3R_\theta)^2. \tag{31}$$

To deal with $\mathbb{E}\|\delta_t^w\|^2$, we introduce the following lemma:

**Lemma E.6.** *For any $t \geq \tau_{\mathrm{mix}}$,*

$$\mathbb{E}\|\delta_t^w\|^2 \;\leq\; 4e^{-2\beta(t - \tau_{\mathrm{mix}})} + (16\tau_{\mathrm{mix}} + 6)\beta = 4\,G_1^{t - \tau_{\mathrm{mix}}} + (16\tau_{\mathrm{mix}} + 6)\beta.$$

The proof of this lemma can be found in Section E.3.

Now apply Lemma E.6 and use $\beta = \rho_0\alpha \leq \alpha$:

$$\mathbb{E}\|\delta_{t+1}^\theta\|^2 \leq \kappa \, \mathbb{E}\|\delta_t^\theta\|^2 + \frac{4\alpha(r_{\max} + 2R_\theta)}{\lambda^2} \, G_1^{t - \tau_{\mathrm{mix}}} + \rho_0\alpha^2 \frac{(16\tau_{\mathrm{mix}} + 6)(r_{\max} + 2R_\theta)}{\lambda^2} + \alpha^2(88\tau_{\mathrm{mix}} + 4)(r_{\max} + 3R_\theta)^2. \tag{32}$$

Iterate (32) from $\tau_{\mathrm{mix}}$ to $t$ and use $\|\delta_{\tau_{\mathrm{mix}}}^\theta\| \leq 2R_\theta$:

$$\mathbb{E}\|\delta_t^\theta\|^2 \leq \kappa^{t - \tau_{\mathrm{mix}}} \mathbb{E}\|\delta_{\tau_{\mathrm{mix}}}^\theta\|^2 + \frac{4\alpha(r_{\max} + 2R_\theta)}{\lambda^2} \sum_{j=\tau_{\mathrm{mix}}}^{t-1} \kappa^{t-1-j} G_1^{j - \tau_{\mathrm{mix}}} + \Big(\alpha^2 G_{\mathrm{const}}\Big) \sum_{j=\tau_{\mathrm{mix}}}^{t-1} \kappa^{t-1-j},$$

where

$$G_{\mathrm{const}} := \frac{\rho_0(16\tau_{\mathrm{mix}} + 6)(r_{\max} + 2R_\theta)}{\lambda^2} + (88\tau_{\mathrm{mix}} + 4)(r_{\max} + 3R_\theta)^2.$$

Use

$$\sum_{j=\tau_{\mathrm{mix}}}^{t-1} \kappa^{t-1-j} G_1^{j - \tau_{\mathrm{mix}}} \leq (t - \tau_{\mathrm{mix}}) \, \max\{\kappa, G_1\}^{t - \tau_{\mathrm{mix}}}, \qquad \sum_{j=\tau_{\mathrm{mix}}}^{t-1} \kappa^{t-1-j} \leq \frac{1}{1 - \kappa} = \frac{1}{2\alpha\zeta}.$$

Then we complete the proof. $\qquad\square$

### E.2. Proof of Lemma E.5

*Proof of Lemma E.5.* For brevity write $g_t := g(s_t, s_t', w_t, \theta_t)$ and $\bar{g}_t := \bar{g}(w_t, \theta_t)$. Using $\delta_t^\theta = \delta_{t-\tau_{\mathrm{mix}}}^\theta + (\theta_t - \theta_{t-\tau_{\mathrm{mix}}})$,

$$\mathbb{E}\big[\delta_t^{\theta\top}(g_t - \bar{g}_t)\big] = \mathbb{E}\big[\delta_{t-\tau_{\mathrm{mix}}}^{\theta\top}(g_t - \bar{g}_t)\big] + \mathbb{E}\big[(\theta_t - \theta_{t-\tau_{\mathrm{mix}}})^\top(g_t - \bar{g}_t)\big].$$

We first note the crude drift bounds over the last $\tau_{\mathrm{mix}}$ steps:

$$\|\theta_t - \theta_{t-\tau_{\mathrm{mix}}}\| \leq \sum_{i=t-\tau_{\mathrm{mix}}}^{t-1} \|\theta_{i+1} - \theta_i\| \leq \alpha \sum_{i=t-\tau_{\mathrm{mix}}}^{t-1} \|g(s_i, s_i', w_i, \theta_i)\| \leq 2\alpha\tau_{\mathrm{mix}}(r_{\max} + 3R_\theta), \tag{33}$$

$$\|w_t - w_{t-\tau_{\mathrm{mix}}}\| \leq \sum_{i=t-\tau_{\mathrm{mix}}}^{t-1} \|w_{i+1} - w_i\| = \beta \sum_{i=t-\tau_{\mathrm{mix}}}^{t-1} \|f(s_i, w_i)\| \leq 2\beta\tau_{\mathrm{mix}}. \tag{34}$$

Now decompose

$$\delta^\theta_{t-\tau_{\mathrm{mix}}}{}^\top (g_t - \bar{g}_t) = \delta^\theta_{t-\tau_{\mathrm{mix}}}{}^\top \big(g_t - g(s_t, s'_t, w_{t-\tau_{\mathrm{mix}}}, \theta_{t-\tau_{\mathrm{mix}}})\big)$$
$$+ \delta^\theta_{t-\tau_{\mathrm{mix}}}{}^\top \big(\bar{g}(w_{t-\tau_{\mathrm{mix}}}, \theta_{t-\tau_{\mathrm{mix}}}) - \bar{g}_t\big)$$
$$+ \delta^\theta_{t-\tau_{\mathrm{mix}}}{}^\top \big(g(s_t, s'_t, w_{t-\tau_{\mathrm{mix}}}, \theta_{t-\tau_{\mathrm{mix}}}) - \bar{g}(w_{t-\tau_{\mathrm{mix}}}, \theta_{t-\tau_{\mathrm{mix}}})\big).$$

Using Lemma E.3, Lemma E.1, and (33)–(34),

$$\mathbb{E}\Big[\delta^\theta_{t-\tau_{\mathrm{mix}}}{}^\top \big(g_t - g(s_t, s'_t, w_{t-\tau_{\mathrm{mix}}}, \theta_{t-\tau_{\mathrm{mix}}})\big)\Big] \leq \mathbb{E}\Big[\|\delta^\theta_{t-\tau_{\mathrm{mix}}}\|\big(3\|\theta_t - \theta_{t-\tau_{\mathrm{mix}}}\| + (r_{\max} + 2R_\theta)\|w_t - w_{t-\tau_{\mathrm{mix}}}\|\big)\Big]$$
$$\leq 2R_\theta\Big(3 \cdot 2\alpha\tau_{\mathrm{mix}}(r_{\max} + 3R_\theta) + (r_{\max} + 2R_\theta) \cdot 2\beta\tau_{\mathrm{mix}}\Big)$$
$$\leq 16\alpha\tau_{\mathrm{mix}}(r_{\max} + 3R_\theta)^2,$$

and the same bound holds with $g$ replaced by $\bar{g}$.

For the remaining bias term, by the definition of $\tau_{\mathrm{mix}}(\beta)$ we have $\sup_s \|P_{\tau_{\mathrm{mix}}}(\cdot|s) - \mu\|_1 \leq \beta$, hence

$$\Big\|\mathbb{E}\big[g(s_t, s'_t, w_{t-\tau_{\mathrm{mix}}}, \theta_{t-\tau_{\mathrm{mix}}}) - \bar{g}(w_{t-\tau_{\mathrm{mix}}}, \theta_{t-\tau_{\mathrm{mix}}}) \mid \mathcal{F}_{t-\tau_{\mathrm{mix}}}\big]\Big\| \leq \beta \cdot \sup_{s,s'} \|g(s, s', w_{t-\tau_{\mathrm{mix}}}, \theta_{t-\tau_{\mathrm{mix}}})\| \leq 2\beta(r_{\max} + 3R_\theta),$$

so

$$\mathbb{E}\Big[\delta^\theta_{t-\tau_{\mathrm{mix}}}{}^\top \big(g(s_t, s'_t, w_{t-\tau_{\mathrm{mix}}}, \theta_{t-\tau_{\mathrm{mix}}}) - \bar{g}(w_{t-\tau_{\mathrm{mix}}}, \theta_{t-\tau_{\mathrm{mix}}})\big)\Big] \leq 2R_\theta \cdot 2\beta(r_{\max} + 3R_\theta) \leq 4\alpha(r_{\max} + 3R_\theta)^2.$$

Finally, for the second main term,

$$\mathbb{E}\big[(\theta_t - \theta_{t-\tau_{\mathrm{mix}}})^\top (g_t - \bar{g}_t)\big] \leq \|\theta_t - \theta_{t-\tau_{\mathrm{mix}}}\| \cdot \mathbb{E}\big[\|g_t\| + \|\bar{g}_t\|\big] \leq 2\alpha\tau_{\mathrm{mix}}(r_{\max} + 3R_\theta) \cdot 4(r_{\max} + 3R_\theta) = 8\alpha\tau_{\mathrm{mix}}(r_{\max} + 3R_\theta)^2.$$

Summing the pieces,

$$\mathbb{E}\big[\delta^\theta_t{}^\top (g_t - \bar{g}_t)\big] \leq (16 + 16 + 4 + 8)\alpha\tau_{\mathrm{mix}}(r_{\max} + 3R_\theta)^2 = 44\alpha\tau_{\mathrm{mix}}(r_{\max} + 3R_\theta)^2.$$

$\square$

### E.3. Proof of Lemma E.6

*Proof of Lemma E.6.* Using non-expansiveness of projection, for $t \geq 0$,

$$\mathbb{E}\|\delta^w_{t+1}\|^2 \leq \mathbb{E}\|\delta^w_t + \beta f(s_t, w_t)\|^2$$
$$= \mathbb{E}\|\delta^w_t\|^2 + 2\beta\,\mathbb{E}\big[(\delta^w_t)^\top f(s_t, w_t)\big] + \beta^2\mathbb{E}\|f(s_t, w_t)\|^2. \tag{35}$$

By Lemma E.2, $\beta^2\mathbb{E}\|f(s_t, w_t)\|^2 \leq 4\beta^2$.

For $t \geq \tau_{\mathrm{mix}}$, write $\bar{f}(w_t) = w^* - w_t = -\delta^w_t$ and decompose

$$\mathbb{E}\big[(\delta^w_t)^\top f(s_t, w_t)\big] = \mathbb{E}\big[(\delta^w_t)^\top (f(s_t, w_t) - \bar{f}(w_t))\big] + \mathbb{E}\big[(\delta^w_t)^\top \bar{f}(w_t)\big] = \mathbb{E}\big[(\delta^w_t)^\top (f(s_t, w_t) - \bar{f}(w_t))\big] - \mathbb{E}\|\delta^w_t\|^2.$$

A standard $\tau_{\mathrm{mix}}$-step decomposition together with $\|w_t - w_{t-\tau_{\mathrm{mix}}}\| \leq 2\beta\tau_{\mathrm{mix}}$, $\|f(s_t, w_t)\| \leq 2$, $\|\delta^w_t\| \leq 2$, and the fact $\sup_s \|P_{\tau_{\mathrm{mix}}}(\cdot|s) - \mu\|_1 \leq \beta$ yields

$$\mathbb{E}\big[(\delta^w_t)^\top (f(s_t, w_t) - \bar{f}(w_t))\big] \leq (16\tau_{\mathrm{mix}} + 4)\beta.$$

Therefore, for $t \geq \tau_{\mathrm{mix}}$,

$$\mathbb{E}\big[(\delta^w_t)^\top f(s_t, w_t)\big] \leq -\mathbb{E}\|\delta^w_t\|^2 + (16\tau_{\mathrm{mix}} + 4)\beta.$$

Plugging into (35) gives

$$\mathbb{E}\|\delta^w_{t+1}\|^2 \leq (1 - 2\beta)\mathbb{E}\|\delta^w_t\|^2 + (32\tau_{\mathrm{mix}} + 12)\beta^2.$$

Iterating from $t = \tau_{\text{mix}}$ and using $\mathbb{E}\|\delta_{\tau_{\text{mix}}}^{w}\|^2 \leq 4$,

$$\mathbb{E}\|\delta_t^w\|^2 \leq (1-2\beta)^{t-\tau_{\text{mix}}} \cdot 4 + (32\tau_{\text{mix}} + 12)\beta^2 \sum_{i=0}^{t-\tau_{\text{mix}}-1} (1-2\beta)^i \leq 4e^{-2\beta(t-\tau_{\text{mix}})} + (16\tau_{\text{mix}} + 6)\beta.$$

$\square$

# F. Analysis of Single Markov Chain Algorithm with decaying step-size

In this section, we provide the proof to Theorem F.1.

We now restate and prove the theorem for the single-chain algorithm with decaying step-size.

**Theorem F.1.** *Suppose Assumption 2.1 holds. Let $a, c_0 > 0$, $\rho_0 \in (0, 1]$, and choose step-sizes*

$$\alpha_t = \frac{a}{(t + c_0)^\xi}, \qquad \beta_t = \rho_0 \alpha_t = \frac{\rho_0 a}{(t + c_0)^\xi}.$$

*Let*

$$c_{1,\lambda} := \eta - \frac{\lambda^2 (r_{\max} + 2R_\theta)}{2},$$

$$c_{2,\lambda} := \frac{r_{\max} + 2R_\theta}{\lambda^2},$$

$$\lambda^2 \le \frac{2\eta}{r_{\max} + 2R_\theta}.$$

*Define $\Delta_T := (T + c_0)^{1-\xi} - (\tau_{\mathrm{mix}} + c_0)^{1-\xi}$, and $G_1(T)$ as in (37), $G_2(T)$ as in (42), $\Gamma_0$ as in (43), $\Gamma_1$ as in (44). Then:*

*1. If $\xi \in (0, 1)$ and*

$$c_0 \ge \max \left\{ \left( \frac{\xi}{ac_{1,\lambda}} \right)^{\frac{1}{1-\xi}}, \left( \frac{\xi}{\rho_0 a} \right)^{\frac{1}{1-\xi}} \right\},$$

*then*

$$\mathbb{E}\big[\|\delta_T^\theta\|^2\big]$$

$$\le 4R_\theta^2 \cdot \exp\left(-\frac{ac_{1,\lambda}}{1-\xi}\Delta_T\right) + \frac{a\big(G_1(T) + \rho_0 c_{2,\lambda} G_2(T)\big)}{c_{1,\lambda}(T + c_0)^\xi}$$

$$+ \frac{4ac_{2,\lambda}}{1-\xi} \exp\left(\frac{ac_{1,\lambda}}{c_0^\xi}\right) \Delta_T \exp\left(-\frac{a}{1-\xi}\rho_{\min}\Delta_T\right).$$

*where $\rho_{\min} = \min\{c_{1,\lambda}, 2\rho_0\}$*

*2. If $\xi = 1$, $a\eta < 1$, and $\lambda^2 \ge \frac{2\eta - 4\rho_0}{r_{\max} + 2R_\theta}$, then*

$$\mathbb{E}\big[\|\delta_T^\theta\|^2\big] \le 4R_\theta^2 \cdot \left(\frac{\tau_{\mathrm{mix}} + c_0}{T + c_0}\right)^{ac_{1,\lambda}} + \frac{\Gamma_1 + \Gamma_0 G_1(T)}{(T + c_0)^{ac_{1,\lambda}}},$$

*where $\Gamma_0$ and $\Gamma_1$ are defined in Appendix F.*

*Proof of Theorem F.1.* As before, the dynamic of $\theta_t$ satisfies

$$\mathbb{E}\|\delta_{t+1}^\theta\|^2 \le \mathbb{E}\|\delta_t^\theta\|^2 + 2\alpha_t \mathbb{E}\Big[\delta_t^{\theta\top} g(s_t, s_t', w_t, \theta_t)\Big] + \alpha_t^2 \mathbb{E}\big[\|g(s_t, s_t', w_t, \theta_t)\|^2\big]$$

$$= \mathbb{E}\|\delta_t^\theta\|^2 + \underbrace{2\alpha_t \mathbb{E}\Big[\delta_t^{\theta\top}\big(g(s_t, s_t', w_t, \theta_t) - \bar{g}(w_t, \theta_t)\big)\Big]}_{I_1}$$

$$+ \underbrace{2\alpha_t \mathbb{E}\Big[\delta_t^{\theta\top}\bar{g}(w_t, \theta_t)\Big]}_{I_2} + \underbrace{\alpha_t^2 \mathbb{E}\big[\|g(s_t, s_t', w_t, \theta_t)\|^2\big]}_{I_3},$$

where

$$\bar{g}(w_t, \theta_t) := \mathbb{E}_{s \sim \mu} g(s, s', w_t, \theta_t) = \Phi^\top D\big(R + P\Phi\theta_t - \Phi\theta_t\big) - \mu^\top(R + \Phi\theta_t)w_t.$$

**Term $I_1$:** We use the following Markov-noise bound.

**Lemma F.2.** *Suppose $t \geq \tau_{\mathrm{mix}}$. Then*

$$\mathbb{E}\left[\delta_t^{\theta\top}\big(g(s_t, s_t', w_t, \theta_t) - \bar{g}(w_t, \theta_t)\big)\right] \leq \rho_1 (r_{\max} + 3R_\theta)^2 \sum_{i=t-\tau_{\mathrm{mix}}}^{t-1} \alpha_i,$$

*where $\rho_1 := 28 + 8\rho_0$.*

The proof is deferred to Section F.1. Using Lemma F.2,

$$I_1 \leq 2\rho_1 \, \alpha_t (r_{\max} + 3R_\theta)^2 \sum_{i=t-\tau_{\mathrm{mix}}}^{t-1} \alpha_i.$$

**Term $I_2$:** By the same argument as Eq. (29) in Appendix E, we have

$$I_2 \leq -2\alpha_t c_{1,\lambda} \, \mathbb{E}\|\delta_t^\theta\|^2 + \alpha_t c_{2,\lambda} \, \mathbb{E}\|\delta_t^w\|^2,$$

where $c_{1,\lambda} = \eta - \lambda^2 (r_{\max} + 2R_\theta)/2$ and $c_{2,\lambda} = (r_{\max} + 2R_\theta)/\lambda^2$.

**Term $I_3$:** By Lemma E.2,

$$I_3 \leq 4\alpha_t^2 (r_{\max} + 3R_\theta)^2.$$

Combining the three terms yields, for $t \geq \tau_{\mathrm{mix}}$,

$$\mathbb{E}\|\delta_{t+1}^\theta\|^2 \leq (1 - 2\alpha_t c_{1,\lambda}) \, \mathbb{E}\|\delta_t^\theta\|^2 + \alpha_t c_{2,\lambda} \, \mathbb{E}\|\delta_t^w\|^2 + 4\alpha_t^2 (r_{\max} + 3R_\theta)^2$$
$$+ 2\rho_1 \, \alpha_t (r_{\max} + 3R_\theta)^2 \sum_{i=t-\tau_{\mathrm{mix}}}^{t-1} \alpha_i.$$

Using Lemma D.2, $\sum_{i=t-\tau_{\mathrm{mix}}}^{t-1} \alpha_i \leq 2L_1 \alpha_t (\log(1/\alpha_t) + 1)$, we further obtain

$$\mathbb{E}\|\delta_{t+1}^\theta\|^2 \leq (1 - 2\alpha_t c_{1,\lambda}) \, \mathbb{E}\|\delta_t^\theta\|^2 + \alpha_t c_{2,\lambda} \, \mathbb{E}\|\delta_t^w\|^2 + G_1(T)\alpha_t^2, \tag{36}$$

where we used the monotonicity of $\alpha_t$ to bound $\log(1/\alpha_t) \leq \log(1/\alpha_T)$ for all $t \leq T$, and defined

$$G_1(T) := \Big(4\rho_1 L_1 \big(\log(T + c_0) - \log a + 1\big) + 4\Big) (r_{\max} + 3R_\theta)^2. \tag{37}$$

For convenience in the iteration, we relax the contraction factor as

$$(1 - 2\alpha_t c_{1,\lambda}) \leq (1 - \alpha_t c_{1,\lambda}) \qquad (\text{since } \alpha_t c_{1,\lambda} \geq 0),$$

so from (36),

$$\mathbb{E}\|\delta_{t+1}^\theta\|^2 \leq (1 - \alpha_t c_{1,\lambda}) \, \mathbb{E}\|\delta_t^\theta\|^2 + \alpha_t c_{2,\lambda} \, \mathbb{E}\|\delta_t^w\|^2 + G_1(T)\alpha_t^2. \tag{38}$$

Iterating (38) from $\tau_{\mathrm{mix}}$ to $T$ gives

$$\mathbb{E}\|\delta_T^\theta\|^2 \leq \underbrace{\prod_{i=\tau_{\mathrm{mix}}}^{T-1} (1 - c_{1,\lambda}\alpha_i)}_{I_1'} \mathbb{E}\|\delta_{\tau_{\mathrm{mix}}}^\theta\|^2 + \sum_{j=\tau_{\mathrm{mix}}}^{T-1} \left(\prod_{i=j+1}^{T-1} (1 - c_{1,\lambda}\alpha_i)\right) \alpha_j c_{2,\lambda} \, \mathbb{E}\|\delta_j^w\|^2$$
$$+ G_1(T) \sum_{j=\tau_{\mathrm{mix}}}^{T-1} \alpha_j^2 \left(\prod_{i=j+1}^{T-1} (1 - c_{1,\lambda}\alpha_i)\right). \tag{39}$$

We first bound $I_1'$. Using $1 - x \le e^{-x}$,

$$I_1' \le \exp\left(-c_{1,\lambda} \sum_{i=\tau_{\text{mix}}}^{T-1} \alpha_i\right) = \exp\left(-ac_{1,\lambda} \sum_{i=\tau_{\text{mix}}}^{T-1} \frac{1}{(i+c_0)^\xi}\right) \le \exp\left(-ac_{1,\lambda} \int_{\tau_{\text{mix}}}^{T} \frac{dx}{(x+c_0)^\xi}\right), \tag{40}$$

hence

$$I_1' \le \begin{cases} \left(\dfrac{\tau_{\text{mix}}+c_0}{T+c_0}\right)^{ac_{1,\lambda}}, & \xi = 1, \\ \exp\left(-\dfrac{ac_{1,\lambda}}{1-\xi}\left((T+c_0)^{1-\xi} - (\tau_{\text{mix}}+c_0)^{1-\xi}\right)\right), & \xi \in (0,1). \end{cases} \tag{41}$$

Next we control $\mathbb{E}\|\delta_j^w\|^2$.

**Lemma F.3.** *Let*

$$G_2(T) := 64L_1\big(\log(T+c_0) - \log(\rho_0 a) + 1\big) + 12. \tag{42}$$

*Then for all $t \in [\tau_{\text{mix}}, T]$:*

1. *If $\xi \in (0,1)$ and $c_0 \ge \left(\frac{\xi}{\rho_0 a}\right)^{\frac{1}{1-\xi}}$, then*

$$\mathbb{E}\|\delta_t^w\|^2 \le 4\exp\left(-\frac{2\rho_0 a}{1-\xi}\left((t+c_0)^{1-\xi} - (\tau_{\text{mix}}+c_0)^{1-\xi}\right)\right) + G_2(T) \cdot \frac{\rho_0 a}{(t+c_0)^\xi}.$$

2. *If $\xi = 1$, then:*

    (a) *If $\rho_0 a \in (0, 1/2)$, then*

    $$\mathbb{E}\|\delta_t^w\|^2 \le 4\left(\frac{\tau_{\text{mix}}+c_0}{t+c_0}\right)^{2\rho_0 a} + G_2(T) \cdot \frac{4\rho_0^2 a^2}{(1-2\rho_0 a)(t+c_0)^{2\rho_0 a}}.$$

    (b) *If $\rho_0 a = 1/2$, then*

    $$\mathbb{E}\|\delta_t^w\|^2 \le 4\left(\frac{\tau_{\text{mix}}+c_0}{t+c_0}\right)^{2\rho_0 a} + G_2(T) \cdot \frac{4\rho_0^2 a^2 \log(t+c_0)}{t+c_0}.$$

    (c) *If $\rho_0 a \in (1/2, \infty)$, then*

    $$\mathbb{E}\|\delta_t^w\|^2 \le 4\left(\frac{\tau_{\text{mix}}+c_0}{t+c_0}\right)^{2\rho_0 a} + G_2(T) \cdot \frac{4e\rho_0^2 a^2}{(2\rho_0 a - 1)(t+c_0)}.$$

The proof is deferred to Section F.2.

We now split into two cases.

**Case 1: $\xi = 1$.** Using (39) and (41) (with $\xi = 1$),

$$\mathbb{E}\|\delta_T^\theta\|^2 \le \left(\frac{\tau_{\text{mix}}+c_0}{T+c_0}\right)^{ac_{1,\lambda}} \mathbb{E}\|\delta_{\tau_{\text{mix}}}^\theta\|^2 + \underbrace{\sum_{j=\tau_{\text{mix}}}^{T-1} \left(\frac{j+1+c_0}{T+c_0}\right)^{ac_{1,\lambda}} \frac{ac_{2,\lambda}}{j+c_0} \mathbb{E}\|\delta_j^w\|^2}_{I_2'}$$

$$+ G_1(T) \underbrace{\sum_{j=\tau_{\text{mix}}}^{T-1} \frac{a^2}{(j+c_0)^2}\left(\frac{j+1+c_0}{T+c_0}\right)^{ac_{1,\lambda}}}_{I_3'}.$$

*Bound on $I_3'$.* Since $(j + 1 + c_0)/(j + c_0) \leq 1 + 1/c_0 \leq 2$ when $c_0 \geq 1$,

$$I_3' = \frac{a^2}{(T + c_0)^{ac_{1,\lambda}}} \sum_{j=\tau_{\mathrm{mix}}}^{T-1} \frac{(j + 1 + c_0)^{ac_{1,\lambda}}}{(j + c_0)^2} \leq \frac{4a^2}{(T + c_0)^{ac_{1,\lambda}}} \sum_{j=\tau_{\mathrm{mix}}}^{T-1} (j + 1 + c_0)^{ac_{1,\lambda}-2}$$

$$\leq \frac{4a^2}{(1 - ac_{1,\lambda})(T + c_0)^{ac_{1,\lambda}}} \qquad (ac_{1,\lambda} \leq a\eta < 1).$$

Define

$$\Gamma_0 := \frac{4a^2}{1 - ac_{1,\lambda}}. \tag{43}$$

Then $I_3' \leq \Gamma_0/(T + c_0)^{ac_{1,\lambda}}$.

*Bound on $I_2'$.* Apply Lemma F.3 (case $\xi = 1$). If $\rho_0 a \in (0, 1/2)$, then

$$I_2' \leq \sum_{j=\tau_{\mathrm{mix}}}^{T-1} \left( \frac{j + 1 + c_0}{T + c_0} \right)^{ac_{1,\lambda}} \frac{ac_{2,\lambda}}{j + c_0} \left[ 4 \left( \frac{\tau_{\mathrm{mix}} + c_0}{j + c_0} \right)^{2\rho_0 a} + G_2(T) \cdot \frac{4\rho_0^2 a^2}{(1 - 2\rho_0 a)(j + c_0)^{2\rho_0 a}} \right]$$

$$= \frac{ac_{2,\lambda}}{(T + c_0)^{ac_{1,\lambda}}} \left( 4(\tau_{\mathrm{mix}} + c_0)^{2\rho_0 a} + \frac{4\rho_0^2 a^2}{1 - 2\rho_0 a} G_2(T) \right) \sum_{j=\tau_{\mathrm{mix}}}^{T-1} \frac{(j + 1 + c_0)^{ac_{1,\lambda}}}{(j + c_0)^{1+2\rho_0 a}}$$

$$\leq \frac{2^{ac_{1,\lambda}} ac_{2,\lambda}}{(T + c_0)^{ac_{1,\lambda}}} \left( 4(\tau_{\mathrm{mix}} + c_0)^{2\rho_0 a} + \frac{4\rho_0^2 a^2}{1 - 2\rho_0 a} G_2(T) \right) \sum_{j=\tau_{\mathrm{mix}}}^{T-1} (j + c_0)^{ac_{1,\lambda}-1-2\rho_0 a}$$

$$\leq \frac{2^{ac_{1,\lambda}} ac_{2,\lambda}}{(T + c_0)^{ac_{1,\lambda}}} \left( 4(\tau_{\mathrm{mix}} + c_0)^{2\rho_0 a} + \frac{4\rho_0^2 a^2}{1 - 2\rho_0 a} G_2(T) \right) \cdot \frac{1}{2\rho_0 a - ac_{1,\lambda}},$$

where the last step uses $ac_{1,\lambda} < 2\rho_0 a$, which follows from $\lambda^2 \geq \frac{2\eta - 4\rho_0}{r_{\max} + 2R_\theta}$ (equivalently, $c_{1,\lambda} \leq 2\rho_0$). The cases $\rho_0 a = 1/2$ and $\rho_0 a > 1/2$ are handled similarly, yielding the same $(T + c_0)^{-ac_{1,\lambda}}$ scaling. Define

$$\Gamma_1 := \begin{cases} \dfrac{2^{ac_{1,\lambda}} ac_{2,\lambda}}{2\rho_0 a - ac_{1,\lambda}} \left( 4(\tau_{\mathrm{mix}} + c_0)^{2\rho_0 a} + \dfrac{4\rho_0^2 a^2}{1 - 2\rho_0 a} G_2(T) \right), & \rho_0 a \in (0, 1/2), \\[3mm] \dfrac{4ac_{2,\lambda}}{1 - ac_{1,\lambda}} \left( 4(\tau_{\mathrm{mix}} + c_0)^{2\rho_0 a} + 4\rho_0^2 a^2 \log(T + c_0) \, G_2(T) \right), & \rho_0 a = 1/2, \\[3mm] \dfrac{4ac_{2,\lambda}}{1 - ac_{1,\lambda}} \left( 4(\tau_{\mathrm{mix}} + c_0)^{2\rho_0 a} + \dfrac{4e\rho_0^2 a^2}{2\rho_0 a - 1} G_2(T) \right), & \rho_0 a \in (1/2, \infty). \end{cases} \tag{44}$$

Then $I_2' \leq \Gamma_1/(T + c_0)^{ac_{1,\lambda}}$.

Putting the bounds together and using $\mathbb{E}\|\delta_{\tau_{\mathrm{mix}}}^\theta\|^2 \leq 4R_\theta^2$, we obtain

$$\mathbb{E}\|\delta_T^\theta\|^2 \leq 4R_\theta^2 \left( \frac{\tau_{\mathrm{mix}} + c_0}{T + c_0} \right)^{ac_{1,\lambda}} + \frac{\Gamma_1 + \Gamma_0 G_1(T)}{(T + c_0)^{ac_{1,\lambda}}}.$$

**Case 2: $\xi \in (0, 1)$.** Recall from Lemma F.3 (case $\xi \in (0, 1)$) that for all $t \in [\tau_{\mathrm{mix}}, T]$,

$$\mathbb{E}\|\delta_t^w\|^2 \leq 4 \exp\left( -A\left( (t + c_0)^{1-\xi} - (\tau_{\mathrm{mix}} + c_0)^{1-\xi} \right) \right) + \rho_0 G_2(T)\alpha_t, \qquad A := \frac{2\rho_0 a}{1 - \xi}. \tag{45}$$

Define

$$\Delta_T := (T + c_0)^{1-\xi} - (\tau_{\mathrm{mix}} + c_0)^{1-\xi}, \qquad S_T := \sum_{j=\tau_{\mathrm{mix}}}^{T-1} \alpha_j^2 \left( \prod_{i=j+1}^{T-1} (1 - c_{1,\lambda}\alpha_i) \right).$$

We also define the exponential-part sum

$$S_T^{\mathrm{exp}} := \sum_{j=\tau_{\mathrm{mix}}}^{T-1} \alpha_j \left( \prod_{i=j+1}^{T-1} (1 - c_{1,\lambda}\alpha_i) \right) \exp\left( -A\left( (j + c_0)^{1-\xi} - (\tau_{\mathrm{mix}} + c_0)^{1-\xi} \right) \right). \tag{46}$$

**Lemma F.4.** *Let $\xi \in (0, 1)$. Then*

$$S_T^{\text{exp}} \le \exp\left(\frac{ac_{1,\lambda}}{c_0^\xi}\right) \cdot \frac{a}{1-\xi} \Delta_T \exp\left(-\frac{a}{1-\xi} \min\{c_{1,\lambda}, 2\rho_0\} \Delta_T\right). \tag{47}$$

The proof is deferred to Section F.3.

Now plug (45) into the second term of (39). Using $c_{2,\lambda}\alpha_j \rho_0 G_2(T)\alpha_j = \rho_0 c_{2,\lambda} G_2(T)\alpha_j^2$, we obtain

$$\sum_{j=\tau_{\text{mix}}}^{T-1} \left(\prod_{i=j+1}^{T-1}(1 - c_{1,\lambda}\alpha_i)\right)\alpha_j c_{2,\lambda} \mathbb{E}\|\delta_j^w\|^2$$
$$\le 4c_{2,\lambda} S_T^{\text{exp}} + \rho_0 c_{2,\lambda} G_2(T) S_T.$$

Therefore, by (39) and (41) (with $\xi \in (0, 1)$),

$$\mathbb{E}\|\delta_T^\theta\|^2 \le \exp\left(-\frac{ac_{1,\lambda}}{1-\xi}\Delta_T\right)\mathbb{E}\|\delta_{\tau_{\text{mix}}}^\theta\|^2 + \left(G_1(T) + \rho_0 c_{2,\lambda} G_2(T)\right) S_T + 4c_{2,\lambda} S_T^{\text{exp}}.$$

By Lemma D.5 and the condition $c_0 \ge \left(\frac{\xi}{ac_{1,\lambda}}\right)^{\frac{1}{1-\xi}}$, we have

$$S_T \le \frac{a}{c_{1,\lambda}(T + c_0)^\xi}.$$

Applying Lemma F.4 (i.e., (47)) and using $\mathbb{E}\|\delta_{\tau_{\text{mix}}}^\theta\|^2 \le 4R_\theta^2$ yields the following bound for Case 2:

$$\mathbb{E}\|\delta_T^\theta\|^2 \le 4R_\theta^2 \exp\left(-\frac{ac_{1,\lambda}}{1-\xi}\Delta_T\right) + \frac{a\left(G_1(T) + \rho_0 c_{2,\lambda} G_2(T)\right)}{c_{1,\lambda}(T + c_0)^\xi}$$
$$+ 4c_{2,\lambda} \exp\left(\frac{ac_{1,\lambda}}{c_0^\xi}\right) \cdot \frac{a}{1-\xi} \Delta_T \exp\left(-\frac{a}{1-\xi}\min\{c_{1,\lambda}, 2\rho_0\}\Delta_T\right).$$

$\square$

## F.1. Proof of Lemma F.2

*Proof of Lemma F.2.* The iteration of $w$ suggests that

$$\|w_t - w_{t-\tau_{\text{mix}}}\| \le \sum_{i=t-\tau_{\text{mix}}}^{t-1} \|w_{i+1} - w_i\| \le \sum_{i=t-\tau_{\text{mix}}}^{t-1} \beta_i\|f(s_i, w_i)\|$$
$$\le 2 \sum_{i=t-\tau_{\text{mix}}}^{t-1} \beta_i. \tag{48}$$

For simplicity, denote $g_t(w_t, \theta_t) := g(s_t, s_t', w_t, \theta_t)$. First, by telescoping and Lemma E.2,

$$\|\theta_t - \theta_{t-\tau_{\text{mix}}}\| \le \sum_{i=t-\tau_{\text{mix}}}^{t-1} \|\theta_{i+1} - \theta_i\| \le \sum_{i=t-\tau_{\text{mix}}}^{t-1} \alpha_i\|g(s_i, s_i', w_i, \theta_i)\|$$
$$\le (2r_{\max} + 6R_\theta) \sum_{i=t-\tau_{\text{mix}}}^{t-1} \alpha_i = 2(r_{\max} + 3R_\theta) \sum_{i=t-\tau_{\text{mix}}}^{t-1} \alpha_i. \tag{49}$$

Decompose

$$\mathbb{E}\left[\delta_t^{\theta\top}\left(g_t(w_t, \theta_t) - \bar{g}(w_t, \theta_t)\right)\right]$$
$$= \mathbb{E}\left[\delta_{t-\tau_{\text{mix}}}^{\theta\top}\left(g_t(w_t, \theta_t) - g_t(w_{t-\tau_{\text{mix}}}, \theta_{t-\tau_{\text{mix}}})\right)\right] + \mathbb{E}\left[\delta_{t-\tau_{\text{mix}}}^{\theta\top}\left(\bar{g}(w_{t-\tau_{\text{mix}}}, \theta_{t-\tau_{\text{mix}}}) - \bar{g}(w_t, \theta_t)\right)\right]$$
$$+ \mathbb{E}\left[\delta_{t-\tau_{\text{mix}}}^{\theta\top}\left(g_t(w_{t-\tau_{\text{mix}}}, \theta_{t-\tau_{\text{mix}}}) - \bar{g}(w_{t-\tau_{\text{mix}}}, \theta_{t-\tau_{\text{mix}}})\right)\right] + \mathbb{E}\left[(\theta_t - \theta_{t-\tau_{\text{mix}}})^\top\left(g_t(w_t, \theta_t) - \bar{g}(w_t, \theta_t)\right)\right]$$
$$=: I_1 + I_2 + I_3 + I_4.$$

**Terms $I_1$ and $I_2$.** By Lemma E.3, $\|g(\cdot, \cdot, w, \theta_1) - g(\cdot, \cdot, w, \theta_2)\| \leq 2\|\theta_1 - \theta_2\|$ and $\|g(\cdot, \cdot, w_1, \theta) - g(\cdot, \cdot, w_2, \theta)\| \leq (r_{\max} + 2R_\theta)\|w_1 - w_2\|$. Also, by (48) and $\beta_i = \rho_0 \alpha_i$,

$$\|w_t - w_{t-\tau_{\mathrm{mix}}}\| \leq 2 \sum_{i=t-\tau_{\mathrm{mix}}}^{t-1} \beta_i = 2\rho_0 \sum_{i=t-\tau_{\mathrm{mix}}}^{t-1} \alpha_i.$$

Thus, using $\|\delta_{t-\tau_{\mathrm{mix}}}^\theta\| \leq 2R_\theta$,

$$
\begin{aligned}
I_1 &\leq \mathbb{E}\big[\|\delta_{t-\tau_{\mathrm{mix}}}^\theta\| \cdot 2\|\theta_t - \theta_{t-\tau_{\mathrm{mix}}}\|\big] + \mathbb{E}\big[\|\delta_{t-\tau_{\mathrm{mix}}}^\theta\| \cdot (r_{\max} + 2R_\theta)\|w_t - w_{t-\tau_{\mathrm{mix}}}\|\big] \\
&\leq 2R_\theta \cdot 4(r_{\max} + 3R_\theta) \sum_{i=t-\tau_{\mathrm{mix}}}^{t-1} \alpha_i + 2R_\theta \cdot (r_{\max} + 2R_\theta) \cdot 2\rho_0 \sum_{i=t-\tau_{\mathrm{mix}}}^{t-1} \alpha_i \\
&\leq (8 + 4\rho_0)R_\theta(r_{\max} + 3R_\theta) \sum_{i=t-\tau_{\mathrm{mix}}}^{t-1} \alpha_i.
\end{aligned}
$$

The same bound applies to $I_2$ since $\bar{g}$ is the expectation of $g$.

**Term $I_3$.** Conditioning on $\mathcal{F}_{t-\tau_{\mathrm{mix}}}$ and using the mixing bound $\|P_{\tau_{\mathrm{mix}}}(\cdot \mid s) - \mu\|_1 \leq C\beta^{\tau_{\mathrm{mix}}} \leq \beta_T \leq \alpha_t$ (for $t \leq T$),

$$
\begin{aligned}
I_3 &\leq \mathbb{E}\left[\|\delta_{t-\tau_{\mathrm{mix}}}^\theta\| \cdot \|P_{\tau_{\mathrm{mix}}}(\cdot \mid s_{t-\tau_{\mathrm{mix}}}) - \mu\|_1 \cdot \sup_{s,s'} \|g(s, s', w_{t-\tau_{\mathrm{mix}}}, \theta_{t-\tau_{\mathrm{mix}}})\|\right] \\
&\leq 2R_\theta \cdot \alpha_t \cdot (2r_{\max} + 6R_\theta) = 4\alpha_t R_\theta(r_{\max} + 3R_\theta).
\end{aligned}
$$

**Term $I_4$.** Using (49) and Lemma E.2,

$$
\begin{aligned}
I_4 &\leq \mathbb{E}\big[\|\theta_t - \theta_{t-\tau_{\mathrm{mix}}}\| \cdot \big(\|g_t(w_t, \theta_t)\| + \|\bar{g}(w_t, \theta_t)\|\big)\big] \\
&\leq 2(r_{\max} + 3R_\theta) \sum_{i=t-\tau_{\mathrm{mix}}}^{t-1} \alpha_i \cdot 2 \cdot (2r_{\max} + 6R_\theta) \\
&= 8(r_{\max} + 3R_\theta)^2 \sum_{i=t-\tau_{\mathrm{mix}}}^{t-1} \alpha_i.
\end{aligned}
$$

Summing the four terms and using $R_\theta \leq r_{\max} + 3R_\theta$ and $\alpha_t \leq \sum_{i=t-\tau_{\mathrm{mix}}}^{t-1} \alpha_i$,

$$
\begin{aligned}
\mathbb{E}\left[\delta_t^{\theta^\top} \big(g_t(w_t, \theta_t) - \bar{g}(w_t, \theta_t)\big)\right] &\leq (16 + 8\rho_0)R_\theta(r_{\max} + 3R_\theta) \sum_{i=t-\tau_{\mathrm{mix}}}^{t-1} \alpha_i + 4\alpha_t R_\theta(r_{\max} + 3R_\theta) \\
&\quad + 8(r_{\max} + 3R_\theta)^2 \sum_{i=t-\tau_{\mathrm{mix}}}^{t-1} \alpha_i \\
&\leq (28 + 8\rho_0)(r_{\max} + 3R_\theta)^2 \sum_{i=t-\tau_{\mathrm{mix}}}^{t-1} \alpha_i,
\end{aligned}
$$

which is the desired result with $\rho_1 = 28 + 8\rho_0$. $\qquad\square$

## F.2. Proof of Lemma F.3

*Proof of Lemma F.3.* The iterates of $w$ suggests that

$$\mathbb{E}\|\delta_{t+1}^w\|^2 \leq \mathbb{E}\|\delta_t^w\|^2 + 2\beta_t \mathbb{E}\big[(\delta_t^w)^\top f(s_t, w_t)\big] + \beta_t^2 \mathbb{E}\|f(s_t, w_t)\|^2.$$

By Lemma E.2, $\mathbb{E}\|f(s_t, w_t)\|^2 \leq 4$, hence the third term is $\leq 4\beta_t^2$.

For the second term, decompose as in the constant-step proof (Appendix E):

$$\mathbb{E}\big[(\delta_t^w)^\top f(s_t, w_t)\big] = I_1 + I_2 + I_3 + I_4 + I_5,$$

where the five terms are exactly those in Section E.3 with $\beta$ replaced by $\beta_t$ (and sums over the last $\tau_{\mathrm{mix}}$ indices). Using the same arguments but keeping the time-varying step-size, we obtain:

$$I_1 \leq 4 \sum_{i=t-\tau_{\mathrm{mix}}}^{t-1} \beta_i,$$

$$I_2 \leq 4\beta_t,$$

$$I_3 \leq 4 \sum_{i=t-\tau_{\mathrm{mix}}}^{t-1} \beta_i,$$

$$I_4 \leq 8 \sum_{i=t-\tau_{\mathrm{mix}}}^{t-1} \beta_i,$$

$$I_5 = -\mathbb{E}\|\delta_t^w\|^2,$$

and therefore

$$\mathbb{E}\big[(\delta_t^w)^\top f(s_t, w_t)\big] \leq -\mathbb{E}\|\delta_t^w\|^2 + 16 \sum_{i=t-\tau_{\mathrm{mix}}}^{t-1} \beta_i + 4\beta_t.$$

Plugging back yields

$$\mathbb{E}\|\delta_{t+1}^w\|^2 \leq (1 - 2\beta_t)\mathbb{E}\|\delta_t^w\|^2 + 32\beta_t\hat{\beta}_t + 12\beta_t^2, \qquad \hat{\beta}_t := \sum_{i=t-\tau_{\mathrm{mix}}}^{t-1} \beta_i. \tag{50}$$

As in Lemma D.2, one can show that

$$\hat{\beta}_t \leq 2L_1 \beta_t \big(\log(1/\beta_t) + 1\big), \qquad t \leq T,$$

where $L_1 = \max\left\{1, \frac{\log C+1}{\log(1/\beta)}\right\}$. Substituting this bound into (50) and using $\log(1/\beta_t) \leq \log(1/\beta_T) = \log(T + c_0) - \log(\rho_0 a)$ for $t \leq T$ gives

$$\mathbb{E}\|\delta_{t+1}^w\|^2 \leq (1 - 2\beta_t)\mathbb{E}\|\delta_t^w\|^2 + G_2(T)\,\beta_t^2,$$

with $G_2(T)$ as defined in Lemma F.3. Unrolling the recursion yields

$$\mathbb{E}\|\delta_t^w\|^2 \leq \underbrace{\prod_{i=\tau_{\mathrm{mix}}}^{t-1} (1 - 2\beta_i)\, \mathbb{E}\|\delta_{\tau_{\mathrm{mix}}}^w\|^2}_{J_1} + G_2(T) \underbrace{\sum_{i=\tau_{\mathrm{mix}}}^{t-1} \beta_i^2 \prod_{j=i+1}^{t-1} (1 - 2\beta_j)}_{J_2}.$$

The term $J_1$ is bounded by $J_1 \leq \exp(-2\sum_{i=\tau_{\mathrm{mix}}}^{t-1} \beta_i)$, which gives the stated exponential bounds for $\xi \in (0, 1)$ and the stated power bounds for $\xi = 1$. The term $J_2$ is bounded by the standard summation estimates, yielding the three sub-cases for $\xi = 1$ and, for $\xi \in (0, 1)$ under $c_0 \geq \left(\frac{\xi}{\rho_0 a}\right)^{\frac{1}{1-\xi}}$, the bound $J_2 \leq \frac{\rho_0 a}{(t+c_0)^\xi}$ via Lemma D.5. Finally, Lemma E.1 gives $\mathbb{E}\|\delta_{\tau_{\mathrm{mix}}}^w\|^2 \leq 4$, completing the proof. $\qquad \square$

### F.3. Proof of Lemma F.4

*Proof of Lemma F.4.* Let $u(t) := (t + c_0)^{1-\xi}$ and $\Delta_T = u(T) - u(\tau_{\mathrm{mix}})$. For any $j \in [\tau_{\mathrm{mix}}, T - 1]$, using $1 - x \leq e^{-x}$ and $\alpha_i \geq 0$,

$$\prod_{i=j+1}^{T-1} (1 - c_{1,\lambda}\alpha_i) \leq \exp\left(-c_{1,\lambda} \sum_{i=j+1}^{T-1} \alpha_i\right).$$

Since $x \mapsto (x + c_0)^{-\xi}$ is decreasing,

$$\sum_{i=j+1}^{T-1} \alpha_i = a \sum_{i=j+1}^{T-1} \frac{1}{(i + c_0)^\xi} \geq a \int_{j+1}^{T} \frac{dx}{(x + c_0)^\xi} = \frac{a}{1 - \xi}\big(u(T) - u(j + 1)\big).$$

Moreover,

$$u(j + 1) = u(j) + \big(u(j + 1) - u(j)\big) \leq u(j) + (1 - \xi)(j + c_0)^{-\xi} = u(j) + \frac{1 - \xi}{a}\alpha_j,$$

hence

$$u(T) - u(j + 1) \geq u(T) - u(j) - \frac{1 - \xi}{a}\alpha_j.$$

Combining the above gives

$$\prod_{i=j+1}^{T-1}(1 - c_{1,\lambda}\alpha_i) \leq \exp\Big(-\frac{ac_{1,\lambda}}{1 - \xi}\big(u(T) - u(j)\big) + c_{1,\lambda}\alpha_j\Big) \leq \exp\Big(\frac{ac_{1,\lambda}}{c_0^\xi}\Big) \cdot \exp\Big(-\frac{ac_{1,\lambda}}{1 - \xi}\big(u(T) - u(j)\big)\Big),$$

where we used $\alpha_j \leq \alpha_0 = a/c_0^\xi$.

Therefore, each summand in (46) satisfies

$$\alpha_j\Big(\prod_{i=j+1}^{T-1}(1 - c_{1,\lambda}\alpha_i)\Big)\exp\big(-A\big(u(j) - u(\tau_{\text{mix}})\big)\big)$$

$$\leq \exp\Big(\frac{ac_{1,\lambda}}{c_0^\xi}\Big) \cdot \alpha_j \exp\Big(-\frac{ac_{1,\lambda}}{1 - \xi}\big(u(T) - u(j)\big) - \frac{2\rho_0 a}{1 - \xi}\big(u(j) - u(\tau_{\text{mix}})\big)\Big).$$

Since $u(j) \in [u(\tau_{\text{mix}}), u(T)]$, we have for all such $j$,

$$c_{1,\lambda}\big(u(T) - u(j)\big) + 2\rho_0\big(u(j) - u(\tau_{\text{mix}})\big) \geq \min\{c_{1,\lambda}, 2\rho_0\} \cdot \big(u(T) - u(\tau_{\text{mix}})\big) = \min\{c_{1,\lambda}, 2\rho_0\}\Delta_T,$$

and hence the exponential factor is at most $\exp\Big(-\frac{a}{1-\xi}\min\{c_{1,\lambda}, 2\rho_0\}\Delta_T\Big)$. Summing and using

$$\sum_{j=\tau_{\text{mix}}}^{T-1} \alpha_j \leq a \int_{\tau_{\text{mix}}}^{T} \frac{dx}{(x + c_0)^\xi} = \frac{a}{1 - \xi}\Delta_T$$

yields (47). $\qquad\square$

# G. Numerical Results

In this section, we provide numerical results on our proposed algorithms. We consider tasks from OpenAI Gym (Towers et al., 2024), MO-Gymnasium (Felten et al., 2023) and Gridworlds. All of the tasks share the following settings:

- Policy: the policy $\pi$ is learned by using the $Q$-learning algorithm.

- Continuous task: some tasks are episodic. To make it a continuous task, the agent will proceed to the starting point after reaching any terminal state with a reward of $0$.

- Ergodic MDP: to ensure the induced Markov chain under this policy is ergodic, we modify the transition matrix as follows: for each state $s$, we examine the $s$-th row of the transition matrix $P$. If the original transition probabilities under the learned policy contains entries that are 0 (e.g., $P(\cdot|s) = [0, 0, 1, 0, 0]$), we redistribute a small portion of the probability mass to all previously unreachable states. Specifically, we assign a small probability $\epsilon$ equally among the zero-probability entries, and reduce the original non-zero entries accordingly to ensure the row still sums to 1. For instance, when $\epsilon = 0.1$, the row above becomes $[0.025, 0.025, 0.9, 0.025, 0.025]$. This adjustment is applied to all rows of $P$, ensuring that every state has a non-zero probability of transitioning to every other state, thus enforcing ergodicity. The choice of $\epsilon$ for different tasks can be found in Table 2.

- Reward function: the reward function $R$ is a vector whose $s$-th row $R(s)$ is defined by the deterministic one-step reward of performing the policy $\pi$ in state $s$.

- Stationary distribution: the stationary distribution $\mu$ is obtained by solving $\mu^T = \mu^T P$.

- Averaged reward: the average reward function $g$ is defined as $g = \mu^T R$.

- Feature matrix: the feature matrix $\Phi$ is defined to be a $|S| \times d$ matrix. We first generate a matrix $\tilde{\Phi} \in \mathbb{R}^{|S| \times (d-2)}$, where each element is drawn from the Bernoulli distribution with success probability $p = 0.5$. Then, we construct $\Phi$ by stacking the all-ones vector $e$ and the true value function $W^*$ as columns into the matrix $\tilde{\Phi}$, i.e., $\Phi = [\tilde{\Phi}, e, W^*]$. The process is repeated until the feature matrix has full column rank. We further normalize the features to ensure $\|\phi(s)\| \leq 1$ for all $s \in S$.

We plot the value function error $\|W_t - W^*\|$ for four algorithms. Two of them are proposed in this paper, namely the Double-Chain and Single-Chain algorithms. As baselines, we include representative prior methods from Table 1. We note that (Tsitsiklis & Van Roy, 1999), (Zhang et al., 2021b), (Haque & Maguluri, 2024), and (Chen et al., 2025) use essentially the same update rule, while (Li et al., 2024) incorporates variance reduction and therefore converges much more slowly.

Here, for our algorithms, we compute $W^\star$ as the unique solution to

$$W^\star + ge = PW^\star + R, \qquad \mu^\top W^\star = 0.$$

For the other algorithms, we measure the error in value-function space modulo additive constants, since in the average-reward setting the relative value function is only defined up to a constant shift. Specifically, rather than comparing $W_t$ with a single representative $W^\star$, we measure the Euclidean distance from $W_t$ to the affine space

$$\mathcal{W}^\star := \{W^\star + ce : c \in \mathbb{R}\},$$

where $e$ denotes the all-one vector. Equivalently, we remove from $W_t - W^\star$ its component in the constant direction $e$, and retain only the orthogonal component. The resulting error metric is

$$\text{dist}(W_t, \mathcal{W}^\star) = \left\| \left( I - \frac{ee^\top}{e^\top e} \right) (W_t - W^\star) \right\|_2.$$

Each curve is averaged over three independent runs. The step-size schedule $\alpha_t$ and the total number of iterations $T$ are reported in Table 2.

*Table 2.* Parameters in different tasks

| Task | $d$ | $\alpha_t$ | $T$ | $\epsilon$ |
|------|-----|-----------|-----|-----------|
| Random Walk (50) | 5 | $150/(t+1000)$ | 150000 | N/A |
| Random Walk (100) | 20 | $150/(t+1000)$ | 150000 | N/A |
| Random Walk (1000) | 100 | $150/(t+1000)$ | 150000 | N/A |
| Frozen Lake | 10 | $150/(t+1000)$ | 150000 | 0.1 |
| Cliff Walking | 20 | $150/(t+1000)$ | 150000 | 0.1 |
| Taxi | 100 | $150/(t+1000)$ | 150000 | 0.5 |
| Grid World (5x5) | 10 | $200/(t+1000)$ | 200000 | 0.2 |
| Grid World (10x10) | 40 | $300/(t+1000)$ | 300000 | 0.2 |
| Grid World (2x11) | 10 | $300/(t+1000)$ | 300000 | 0.2 |
| Deep sea | 20 | $200/(t+1000)$ | 200000 | 0.2 |
| Deep sea (concave) | 20 | $200/(t+1000)$ | 200000 | 0.2 |
| Resource Gathering | 50 | $500/(t+1000)$ | 500000 | 0.5 |
| Fruit Tree (depth = 5) | 50 | $150/(t+1000)$ | 150000 | 0.2 |
| Fruit Tree (depth = 6) | 50 | $150/(t+1000)$ | 150000 | 0.2 |
| Fruit Tree (depth = 7) | 50 | $300/(t+1000)$ | 300000 | 0.2 |

*Table 3.* Distance in value space in different tasks (mean $\pm$ std)

| TASK | DOUBLECHAIN | SINGLECHAIN | (HAQUE & MAGULURI, 2024) & (ZHANG ET AL., 2021B) & (CHEN ET AL., 2025) | (KIM ET AL., 2025) |
|------|-------------|-------------|------------------------------------------------------------------------|---------------------|
| RANDOM WALK (50) | $0.12 \pm 0.02$ | $\mathbf{0.04 \pm 0.00}$ | $\mathbf{0.04 \pm 0.00}$ | $\mathbf{0.04 \pm 0.00}$ |
| RANDOM WALK (100) | $0.19 \pm 0.02$ | $0.07 \pm 0.01$ | $\mathbf{0.06 \pm 0.01}$ | $\mathbf{0.06 \pm 0.01}$ |
| RANDOM WALK (1000) | $2.94 \pm 0.13$ | $\mathbf{2.82 \pm 0.03}$ | $2.84 \pm 0.03$ | $2.92 \pm 0.03$ |
| FROZEN LAKE | $\mathbf{0.44 \pm 0.00}$ | $0.44 \pm 0.01$ | $0.44 \pm 0.01$ | $0.44 \pm 0.01$ |
| CLIFF WALKING | $\mathbf{0.62 \pm 0.12}$ | $0.85 \pm 0.02$ | $0.88 \pm 0.01$ | $0.88 \pm 0.01$ |
| TAXI | $15.88 \pm 2.96$ | $\mathbf{5.90 \pm 0.38}$ | $9.05 \pm 0.45$ | $8.96 \pm 0.42$ |
| GRID WORLD (5x5) | $0.04 \pm 0.01$ | $\mathbf{0.01 \pm 0.00}$ | $0.12 \pm 0.00$ | $0.12 \pm 0.00$ |
| GRID WORLD (10x10) | $\mathbf{0.44 \pm 0.03}$ | $0.49 \pm 0.01$ | $0.45 \pm 0.01$ | $0.48 \pm 0.01$ |
| GRID WORLD (2x11) | $0.04 \pm 0.00$ | $\mathbf{0.01 \pm 0.00}$ | $0.07 \pm 0.00$ | $0.08 \pm 0.00$ |
| DEEP SEA | $0.22 \pm 0.06$ | $\mathbf{0.09 \pm 0.02}$ | $0.28 \pm 0.01$ | $0.29 \pm 0.01$ |
| DEEP SEA (CONCAVE) | $0.15 \pm 0.03$ | $\mathbf{0.05 \pm 0.01}$ | $0.15 \pm 0.01$ | $0.15 \pm 0.01$ |
| RESOURCE GATHERING | $\mathbf{0.00 \pm 0.00}$ | $\mathbf{0.00 \pm 0.00}$ | $\mathbf{0.00 \pm 0.00}$ | $\mathbf{0.00 \pm 0.00}$ |
| FRUIT TREE (DEPTH = 5) | $0.87 \pm 0.14$ | $\mathbf{0.49 \pm 0.04}$ | $0.51 \pm 0.02$ | $0.52 \pm 0.02$ |
| FRUIT TREE (DEPTH = 6) | $1.22 \pm 0.10$ | $\mathbf{0.51 \pm 0.03}$ | $0.85 \pm 0.01$ | $0.85 \pm 0.01$ |
| FRUIT TREE (DEPTH = 7) | $0.86 \pm 0.17$ | $\mathbf{0.38 \pm 0.05}$ | $0.46 \pm 0.06$ | $0.45 \pm 0.05$ |

*Table 4.* Condition numbers in different tasks

| Task | $\eta_1$ | $\eta_2$ | $\eta_3$ |
|---|---|---|---|
| Random Walk (50) | $\mathbf{3.64 \times 10^{-3}}$ | $2.77 \times 10^{-3}$ | $3.04 \times 10^{-3}$ |
| Random Walk (100) | $\mathbf{3.68 \times 10^{-3}}$ | $2.63 \times 10^{-3}$ | $3.19 \times 10^{-3}$ |
| Random Walk (1000) | $\mathbf{1.91 \times 10^{-4}}$ | $1.09 \times 10^{-4}$ | $1.82 \times 10^{-4}$ |
| Frozen Lake | $\mathbf{4.81 \times 10^{-4}}$ | $4.26 \times 10^{-4}$ | $5.20 \times 10^{-5}$ |
| Cliff Walking | $\mathbf{5.04 \times 10^{-4}}$ | $3.27 \times 10^{-4}$ | $7.27 \times 10^{-5}$ |
| Taxi | $\mathbf{2.29 \times 10^{-4}}$ | $1.75 \times 10^{-4}$ | $1.69 \times 10^{-5}$ |
| Grid World (5x5) | $\mathbf{4.21 \times 10^{-3}}$ | $3.94 \times 10^{-3}$ | $1.60 \times 10^{-3}$ |
| Grid World (10x10) | $\mathbf{4.09 \times 10^{-4}}$ | $2.44 \times 10^{-4}$ | $7.83 \times 10^{-5}$ |
| Grid World (2x11) | $3.77 \times 10^{-3}$ | $\mathbf{3.88 \times 10^{-3}}$ | $1.23 \times 10^{-3}$ |
| Deep sea | $\mathbf{1.73 \times 10^{-3}}$ | $1.19 \times 10^{-3}$ | $4.16 \times 10^{-4}$ |
| Deep sea (concave) | $\mathbf{1.92 \times 10^{-3}}$ | $1.28 \times 10^{-3}$ | $3.72 \times 10^{-4}$ |
| Resource Gathering | $4.35 \times 10^{-3}$ | $\mathbf{4.36 \times 10^{-3}}$ | $2.07 \times 10^{-3}$ |
| Fruit Tree (depth = 5) | $\mathbf{1.28 \times 10^{-5}}$ | $7.94 \times 10^{-6}$ | $5.71 \times 10^{-6}$ |
| Fruit Tree (depth = 6) | $\mathbf{1.07 \times 10^{-4}}$ | $6.36 \times 10^{-5}$ | $3.75 \times 10^{-5}$ |
| Fruit Tree (depth = 7) | $\mathbf{1.92 \times 10^{-4}}$ | $1.11 \times 10^{-4}$ | $5.75 \times 10^{-5}$ |

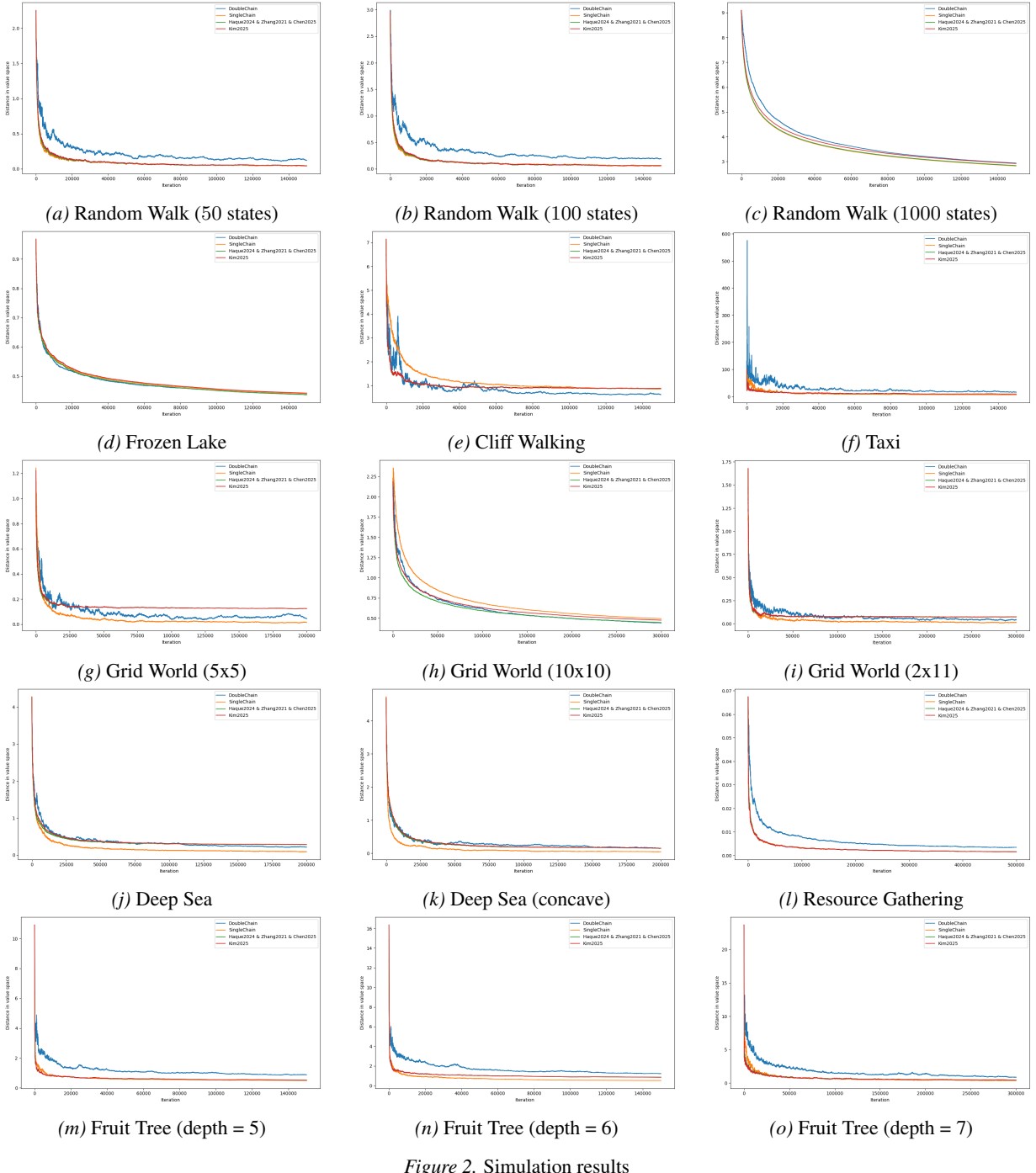

*(a)* Random Walk (50 states)     *(b)* Random Walk (100 states)     *(c)* Random Walk (1000 states)

*(d)* Frozen Lake     *(e)* Cliff Walking     *(f)* Taxi

*(g)* Grid World (5x5)     *(h)* Grid World (10x10)     *(i)* Grid World (2x11)

*(j)* Deep Sea     *(k)* Deep Sea (concave)     *(l)* Resource Gathering

*(m)* Fruit Tree (depth = 5)     *(n)* Fruit Tree (depth = 6)     *(o)* Fruit Tree (depth = 7)

*Figure 2.* Simulation results

# H. Numerical Experiment: Battery Storage Dispatch

We evaluate the proposed double-chain and single-chain average-reward TD algorithms on a policy-evaluation task arising from grid-connected battery storage arbitrage. This domain provides a realistic, continuous-state testbed with interpretable economics: the battery earns revenue by buying electricity when prices are low and selling when prices are high, subject to capacity constraints and round-trip efficiency losses. We focus on a uniformly random policy $\pi_{\text{rand}}$, which produces a rich, exploratory trajectory that exercises the full state space.

## H.1. Environment

The state at time $t$ is $(e_t, p_t, h_t)$, where $e_t \in [0, \bar{E}]$ is the stored energy (MWh), $p_t \geq 0$ is the electricity spot price (\$/MWh), and $h_t \in \{0, 1, \ldots, 23\}$ is the hour of day. The action $a_t \in [-\bar{P}_D, \bar{P}_C]$ specifies the charge (positive) or discharge (negative) rate in MW. The per-step cost is $c_t = p_t \cdot a_t \cdot \Delta t$, so selling energy (negative action) generates revenue (negative cost).

The stored energy evolves as

$$e_{t+1} = \text{clip}\Big(e_t + \phi(a_t),\ 0,\ \bar{E}\Big), \qquad \phi(a) = \begin{cases} \eta_C\, a & \text{if } a \geq 0 \text{ (charging)}, \\ a/\eta_D & \text{if } a < 0 \text{ (discharging)}, \end{cases}$$

where $\eta_C = \eta_D = 0.95$ are the charge and discharge efficiencies. The round-trip efficiency is $\eta_C \eta_D \approx 0.90$, meaning roughly 10% of energy is lost per charge–discharge cycle.

The spot price follows an AR(1) process with diurnal mean:

$$p_{t+1} = \mu(h_{t+1}) + \rho_P\big(p_t - \mu(h_t)\big) + \sigma_P\, \varepsilon_t, \qquad \varepsilon_t \sim \mathcal{N}(0, 1),$$

where $\rho_P = 0.85$, $\sigma_P = 8$, and the diurnal mean is $\mu(h) = \bar{\mu} + \alpha_1 \sin(2\pi h/24 + \varphi_1) + \alpha_2 \sin(4\pi h/24 + \varphi_2)$ with $\bar{\mu} = 40$, $\alpha_1 = 15$, $\varphi_1 = -\pi/3$, $\alpha_2 = 5$, $\varphi_2 = \pi/4$. Prices are clipped below at zero. The hour advances deterministically: $h_{t+1} = (h_t + 1) \bmod 24$, with time step $\Delta t = 1$ hour.

Other parameters are battery capacity $\bar{E} = 4$ MWh; maximum charge/discharge power $\bar{P}_C = \bar{P}_D = 1$ MW.

## H.2. Algorithms

All algorithms use a shared 12-dimensional linear feature vector $\psi(s) \in \mathbb{R}^{12}$:

$$\psi(e, p, h) = \left[\ \frac{e}{\bar{E}},\ \left(\frac{e}{\bar{E}}\right)^2,\ \frac{p}{\bar{\mu}},\ \left(\frac{p}{\bar{\mu}}\right)^2,\ \sin\frac{2\pi h}{24},\ \cos\frac{2\pi h}{24},\ \sin\frac{4\pi h}{24},\ \cos\frac{4\pi h}{24},\ \frac{e}{\bar{E}} \cdot \frac{p}{\bar{\mu}},\ \frac{e}{\bar{E}} \cdot \sin\frac{2\pi h}{24},\ \frac{p}{\bar{\mu}} \cdot \sin\frac{2\pi h}{24},\ 1\ \right]^\top.$$

This basis captures the main structure of the differential value function: quadratic dependence on energy and price, Fourier components for diurnal periodicity, cross terms for the arbitrage interaction, and a constant offset.

The evaluated policy $\pi_{\text{rand}}$ selects an action uniformly at random from the feasible discrete action set $\mathcal{A}(e_t) \subset \{-1.0, -0.8, \ldots, 0.8, 1.0\}$ (11 evenly spaced values), where feasibility is determined by the current energy level and battery constraints. Under this policy, the Monte Carlo estimate of the true average cost is $\rho^\star = +1.077 \pm 0.008$ \$/hr (estimated from $10^6$ steps with block-averaged standard error), indicating a net cost: random cycling against the 10% round-trip loss destroys value.

We compare our single and double chain methods against a baseline of differential TD and discounted TD with high discount factor.

## H.3. Experimental Protocol

The true average cost $\rho^\star$ is estimated from a single Monte Carlo rollout of $T_{\text{MC}} = 10^6$ steps (approximately 114 simulated years), using seed 42. Standard errors are computed via block averaging (100 blocks). From the second half of this rollout, $M = 50{,}000$ held-out transitions are extracted for computing the mean squared TD error (MSTDE).

Each algorithm is trained for a million steps. Step sizes are swept over $\alpha \in \{0.001, 0.003, 0.01, 0.03\}$; for each algorithm family, the step size yielding the lowest final MSTDE (median over seeds) is selected. All experiments use $N = 5$ independent seeds with common random numbers: for a given seed, all algorithms observe the same primary trajectory. The double-chain algorithm additionally uses an independent secondary trajectory.

We evaluate every $K = 5,000$ steps the mean squared TD error on held-out data, $\text{MSTDE} = \frac{1}{M} \sum_{i=1}^{M} \left( c_i - \hat{\rho} + \psi(s_i')^\top \theta - \psi(s_i)^\top \theta \right)^2$. Statistics are reported as medians with ranges across the 5 seeds.

## H.4. Results

Table 5 shows results at double the training horizon, and Figure 3 shows the corresponding learning curves.

*Table 5.* Final MSTDE and $|\hat{\rho} - \rho^\star|$ for $\pi_{\text{rand}}$ at $T_{\text{train}} = 1,000,000$ (median over 5 seeds, best step size per algorithm). The best MSTDE is in bold.

| Method | $\alpha^\star$ | MSTDE |
|---|---|---|
| Double-Chain (ours) | 0.001 | 84.6 |
| Single-Chain (ours) | 0.003 | **71.4** |
| Differential TD | 0.01 | 73.5 |
| Disc. TD ($\gamma{=}0.99$) | 0.003 | 72.6 |
| Disc. TD ($\gamma{=}0.999$) | 0.01 | 73.6 |
| Disc. TD ($\gamma{=}0.9999$) | 0.03 | 86.6 |

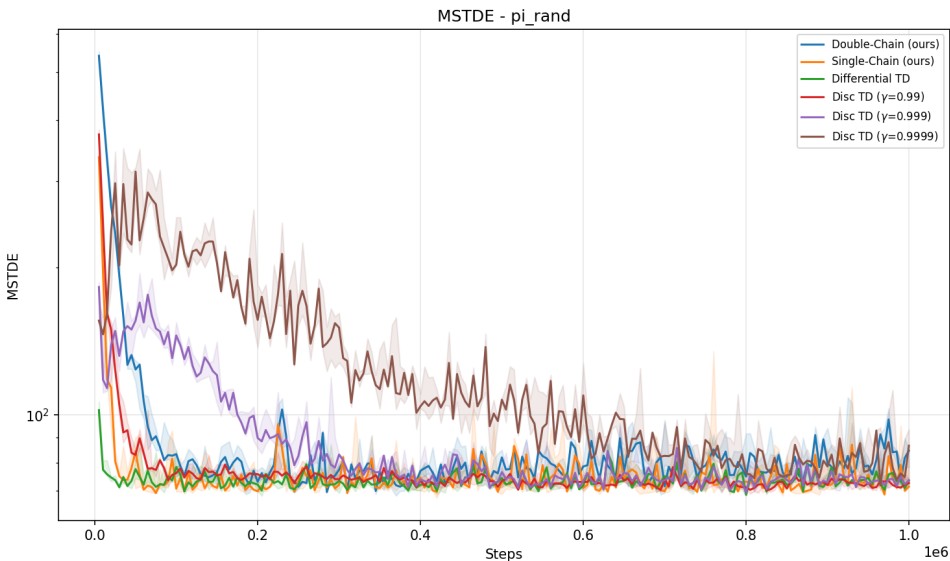

*Figure 3.* MSTDE vs. training steps for $\pi_{\text{rand}}$ at $T_{\text{train}} = 1,000,000$. The single-chain algorithm achieves the lowest final MSTDE (71.4) among all methods.

With additional training, all methods improve on $\pi_{\text{rand}}$. Notably, the single-chain algorithm achieves the *lowest* MSTDE among all methods (71.4), ahead of discounted TD at $\gamma = 0.99$ (72.6) and differential TD (73.5). The preferred step sizes shift toward larger values at $T_{\text{train}} = 10^6$ (e.g., single-chain selects $\alpha = 0.003$ instead of 0.001), reflecting the longer horizon available for averaging out noise.

