# OpenReview forum: "Bridging the Gap Between Average and Discounted TD Learning"
_ICML.cc/2026/Conference — ICML 2026 regular_

### Official Review · Reviewer_Sfs4 · 2026-03-07

**Soundness:** 3
**Presentation:** 2
**Significance:** 3
**Originality:** 2
**Overall Recommendation:** 4
**Confidence:** 4

**Summary:**

This paper studies average reward RL with linear function approximation. The paper first developed a fixed-point typed characterization of the solution, and developed two approaches to address the double sampling issue. Convergence and sample complexity are then analyzed under two approaches.

**Compliance With Llm Reviewing Policy:**

Affirmed.

**Final Justification:**

My concerns have been mostly addressed.

**Key Questions For Authors:**

(1). Could you explain how the convergence is guaranteed, when the Bellman operator is not a contraction (as you mentioned in the conclusion)?

(2). On the other hand, in Lemma A.1, the projected operator is then shown to be a contraction under D-norm. Is the convergence based on this contraction? If so, what is the difference compared to discounted studies, where the contraction-based approaches are developed?

(3). In early works on linear function approximation for average reward, it is generally assumed that $e=(1,..,1)$ is not in the span space of the linear features, but not in this paper. Could you explain the reason of bypass this assumption?

(4). To address the double sampling issue, you proposed a double markov chain method and a single chain method. In the single chain method, are you using the standard two time-scale approach? If so, what are the differences compared to previous two time-scale algorithm analysis like TDC/GTD?

**Limitations:**

I believe the major limitation is that the work is mainly focusing on policy evaluation problem.

**Strengths And Weaknesses:**

S:

(1). The paper studies an important and interesting problem of average reward RL. The paper writing is (mostly) clear, especially the method development.
(2). The sample complexity derived is SOTA.

W:
(1). If I am not wrong, this paper only studies policy evaluation problem (for a fixed policy). But a more important problem is on optimal control.
(2). The discussion of the methods could be a bit more clear (see my questions part).

---

> ### Author Rebuttal · Authors · 2026-03-30
>
> We appreciate the reviewer's close reading and address their concerns below.
>
> **Q1: How is convergence guaranteed when the Bellman operator is not a contraction?**
>
> This is the central technical question of the paper, and the answer has two layers. First, while $T_\pi$ is not a contraction under any norm, the *projected* operator $\Pi T_\pi$ is a contraction under the $D$-norm (Lemma 2.1 in the paper). This is what guarantees that the fixed point $W^*$ is unique and that a deterministic iteration based on $\Pi T_\pi$ would converge. However, our convergence proof does *not* directly exploit this contraction to analyze the stochastic algorithm. Instead, the core tool is *gradient splitting*: we show that the expected update direction of our algorithm,
> is a gradient splitting of an appropriately chosen function. The gradient splitting perspective converts the stochastic iteration into something analyzable by standard SGD arguments.
>
> **Q2: Is convergence based on the contraction of $\Pi T_\pi$, and what is different from the discounted case?**
>
> The contraction of $\Pi T_\pi$ (Lemma 2.1) plays an important conceptual role, in that it confirms $W^*$ is well-defined, but it is not used directly in the finite-sample proof.
>
> The key structural difference from the discounted case is the following. In discounted TD, $T_\pi^\gamma W = R + \gamma P W$ is itself a contraction under the $D$-norm with factor $\gamma < 1$, and this contraction factor directly controls the convergence rate. In the average reward case, $\gamma = 1$, so $T_\pi$ is only non-expansive, and the contraction factor of $\Pi T_\pi$ is determined by mixing properties of the chain, specifically $\omega < 1$ from Lemma 2.1, rather than being a freely chosen discount factor. The gradient splitting approach sidesteps the need to work with $\omega$ directly, replacing it with the condition number $\eta$.
>
> **Q3: Why is the assumption $\Phi\theta \neq ce$ not needed here?**
>
> A lot of prior work required this assumption specifically to ensure uniqueness of the solution to the projected Bellman equation $W = \Pi T_\pi W$. Without it, the solution set is an affine subspace (solutions defined up to an additive constant times $e$), and convergence to a specific point cannot be guaranteed. The tabular case immediately violates this assumption.
>
> Our approach bypasses this assumption by directly incorporating the steady-state constraint. As noted in Section 2.2, the relative value function $W^*$ satisfies both $W^* = \Pi(T_\pi W^*)$ *and* $\mu^T W^* = 0$.  Our algorithm is designed around this equation, so uniqueness is guaranteed by the structure of the problem.
>
> **Q4: Is the single-chain algorithm two-timescale, and what distinguishes it from GTD/TDC?**
>
> It is not quite two-timescale because the ratio of step-sizes $\beta_t/\alpha_t$ is constant rather than going to infinity. It differs from GTD/TDC in purpose and structure. In GTD and TDC, the auxiliary variable $w_t$ tracks the product $\Phi^T D \delta_t$ where $\delta_t$ is a TD error. This is specific to the discounted off-policy setting and the gradient of the MSPBE objective, and is introduced to overcome the deadly triad challenge. In our algorithm, the auxiliary variable $w_t$ tracks $\Phi^T\mu$, the feature matrix weighted by the stationary distribution. This is a different quantity requiring a different analysis.
>
> **Significance of policy evaluation**
>
> The history of RL theory is, to a surprising degree, a history of policy evaluation advances unlocking policy optimization advances.
>
> TD with linear FA (Tsitsiklis & Van Roy, *IEEE TAC* 1997) is the foundation on which every finite-time two-timescale actor-critic proof rests, e.g., Konda & Tsitsiklis (2003), Wu et al. (NeurIPS 2020), and Olshevsky & Gharesifard (SIAM 2023) all explicitly invoke TD convergence for the critic.
>
> LSTD (Bradtke & Barto, *Machine Learning* 1996) directly enabled LSPI (Lagoudakis & Parr, *JMLR* 2003), which built on incremental TD.
>
> GTD/TDC (Sutton et al., *ICML* 2009; Maei et al., *NeurIPS* 2009), the first convergent off-policy TD with function approximation, was the a precondition for provably convergent off-policy actor-critic (e.g., COF-PAC, Zhang et al. *ICML* 2020).
>
> GAE (Schulman et al., ICLR 2016) provided a practically important bias–variance tradeoff for advantage estimation and became a standard component of TRPO-style policy optimization, later carrying over naturally into PPO.
>
> More recent examples are V-trace (Espeholt et al., *ICML* 2018) for RL at scale (IMPALA, SEED RL) and Retrace($\lambda$) (Munos et al., *NeurIPS* 2016)  for off-policy actor-critic with experience replay (ACER, Wang et al. *ICLR* 2017).
>
> The pattern is not a coincidence: policy evaluation is the inner loop of actor-critic. Along similar lines, please see also our response to reviewer ARxj explaining how convergence to a sample-independent fixed point (established in this work) is required for current actor-critic analyses to work.

---

> > ### Author Rebuttal · Reviewer_Sfs4 · 2026-04-01
> >
> > I sincerely appreciate the authors' response. Most of my concerns are addressed and I will adjust my rating accordingly.

---

### Official Review · Reviewer_n79Y · 2026-03-09

**Soundness:** 3
**Presentation:** 3
**Significance:** 3
**Originality:** 3
**Overall Recommendation:** 5
**Confidence:** 4

**Summary:**

The paper addresses the problem of on-policy evaluation in the average reward setting.
The paper introduces an analogous to the projected bellman error for the average reward setting. Using the introduced projection, it derives a recursive update rule (the double-chain algorithm) for the relative value function update for tabular and linear function approximation. The double-chain algorithm uses two independent samples to estimate the updates in order to address the double-sampling problem. The paper then introduces a single-chain algorithm that bypasses the double sampling by introducing a second update rule that estimates the expected value of the features rather than using the features from another independent sample.

The paper establishes the gradient splitting view point for their method and proves the convergence of the double-chain algorithm when using a fixed step size and when using a decaying step-size under both iid sampling assumption and markov sampling, and also proves the convergence of the single-chain algorithm under constant step-size and markov sampling. Finally, the paper evaluated the proposed algorithms across some tabular MDPs.

**Compliance With Llm Reviewing Policy:**

Affirmed.

**Final Justification:**

The rebuttal had addressed all my concerns and I still positively support this paper.

**Key Questions For Authors:**

- In the equation defining the condition number η (Eq.8), the variable x is used but it was never defined. Did you mean \theta?
- The double chain algorithm seems to be notably worse on several tasks in Figure 2, is there a hypothesis on why this is the case? I was assuming that the single-chain will actually be worse since it has two updates and has worse scaling with the condition number.
- Do you think that using variance reduction methods won’t improve the sample complexity of the proposed algorithms?

**Limitations:**

The authors didn't discuss the limitations of their work. I see the following as limitations:

- Limited experiments on the proposed algorithms. Only tabular experiments were performed and no linear function approximation experiments. I don’t think this is major though since the paper’s main contribution is theoretical.
- There are a few overclaims regarding the contributions that need to be addressed such as the dimensionality dependance and the variance reduction.

**Strengths And Weaknesses:**

## Strengths
- Theoretical analysis of average reward setting is limited and the paper provides some useful theoretical results in this setting. Particularly, around the quadratic scaling with the condition number.
- The proposed convergence analysis is applicable to both linear and tabular setting.
- The paper is easy to read and follow.

## Weaknesses
- The double chain algorithm can be only used when a simulator is available. Otherwise, it’s impossible to sample two independent markov chains from a single chain of interactions. I think the authors need to clarify this problem since it makes this algorithm impractical. This limitation is inherent to the double sampling problem in general and is not specific to this paper in particular but I think it’s worth emphasizing here.

- I find the claim on no variance reduction needed to be a bit of an overclaim. While the paper didn’t need to use variance reduction techniques in their proof, they haven’t shown any variance analysis for their algorithms. Hence, it’s hard to conclude that “no variance reduction is needed”.

- I think the claim on page 2 that said
> “Dependence on Dimensionality: Our convergence
bound does not have any explicit factors of d, the dimension of θ. While all algorithms have terms like ∥θ∗∥that might implicitly scale with dimension, our algorithm has no terms scaling with din addition to those.”

contradicts the bounds in Theorem 4.1 ,and  4.2  as they depend on the norm of θ∗ which scales with d.  The authors did make an assumption in Table 1 that said “ We consider a convergence time to be independent of the dimension d if its dependence on d appears only through norm of theta_0 or theta^*” but this assumptions just hides the dependency and not address it. I think that assumption and the claim based on it should be re-evaluated.

I think this is a good paper, my acceptance is conditioned on the authors appropriately addressing the points above.

---

> ### Author Rebuttal · Authors · 2026-03-30
>
> We thank the reviewer for the careful reading and the positive overall assessment. We address each point in turn.
>
> **Does the double-chain algorithm require a simulator?**
>
> The reviewer is correct that an update of the double-chain algorithm requires two independent trajectories. However, this does not necessarily require a generative model: running two rollouts sequentially and pairing them by time index satisfies the independence requirement. The practical limitation is the ability to **reset the environment** and run a second episode, which is standard in simulation but unavailable in true single-interaction online settings.
>
> So this comes pretty close to requiring a simulator, but nevertheless is not exactly the same. For example, the double chain algorithm could be implemented in a real-world setting where rollouts can be restarted.
>
> **Variance reduction claim**
>
> We only meant that we did not use variance reduction in SVRG/SAGA sense to achieve our results. We did not mean to say that variance is/isn't reduced (as the reviewer correctly points out, no such analysis was done). We'll rephrase our claim to avoid confusion on this point.
>
> **Dimensionality dependence**
>
> We appreciate the reviewer pressing on this point, and we want to be precise about what we are and are not claiming. The claim is not that the bounds are dimension-free in an absolute sense; clearly $ || \theta^* || $ can grow with $d$ in the worst case, and we acknowledge this. The claim is specifically a comparison to [Haque & Maguluri, 2024], whose bounds contain terms that scale explicitly with $d$ *beyond* what appears through $ || \theta^* || $ and $ || \theta_0 || $. Concretely, their bounds contain a factor which depends on $c_\alpha^2 = O(d)$ n addition to scaling with $|| \theta^* || $.  Our bounds do not have such *additional* $d$-dependent prefactors. We agree  that the footnote in Table 1 could be made clearer, and we will revise it.
>
> **Question 1: Definition of $x$ in Eq. (8).**
>
> We thank the reviewer for catching this. The variable $x \in \mathbb{R}^d$ is the parameter direction being minimized over. We will add "where $x \in \mathbb{R}^d$" explicitly after Eq. (8).
>
> **Question 2: Why does the double-chain algorithm perform worse on several tasks in Figure 2?**
>
> This is a great observation and counterintuitive given that the double-chain algorithm has better theoretical guarantees under Markov sampling. One possibility is that it may be possible to give an improved analysis of the single-chain method which is better at least some classes of MDPs. This is a question we leave for future work.
>
> **Question 3: Could variance reduction improve sample complexity?**
>
> Possibly yes, and we consider this an interesting open question! Adapting SVRG to the average reward setting using our gradient splitting framework seems feasible in principle, since the gradient splitting interpretation explicitly identifies the quadratic objective being minimized, and SVRG-style methods can be applied to any such objective. Whether this yields improved finite-sample rates in our specific setting is a question we leave for future work.

---

> > ### Author Rebuttal · Reviewer_n79Y · 2026-04-01
> >
> > Thank you for your answers, that cleared up my questions. I will maintain my positive score.

---

### Official Review · Reviewer_ARxj · 2026-03-13

**Soundness:** 2
**Presentation:** 2
**Significance:** 1
**Originality:** 2
**Overall Recommendation:** 2
**Confidence:** 4

**Summary:**

Focus is on showing that for average reward TD learning, the convergence occurs to a single point (which does not depend upon sample path). The problem studied is not well-motivated, and there are benefits of showing this property. There are some controversial issuues regarding the convergence rates too. I am willing to change my score if the authors address my concern.

**Compliance With Llm Reviewing Policy:**

Affirmed.

**Final Justification:**

The rebuttal has not addressed my concerns satisfactorily. Though the authors made an attempt to address my concerns. The paper does seem to be technically sound. Only the reply to point 2. and 4 (partially) looks convincing.

**Key Questions For Authors:**

1. Please motivate the problem that is being studied by discussing why this uniqueness of solution matters. There have been works which show convergence to a set or to sample-dependent fixed points, and this convergence is sufficient for evaluating the policy.

2. There have also been works which show convergence to a point (Haque & Maguluri, 2024) and (Chen et al., 2025). The improvement over these results is the quadratic (instead of quartic) dependence on the "condition number". Explicitly describe what these condition numbers represent in practice.

3. The literature survey seems to be missing work on average reward control algorithms such as RVI Q-learning and SSP Q-learning.  Moreover, the tools used for analysis of RVI Q-learning are very similar in some aspects. Can you highlight this explicitly?

4. The numerical simulations seem to be doing an unfair comparison. The authors state "we evaluate all methods using their approximation error relative to the true solution rather than properties of their iterates." The definition of the true solution here is \theta^* which is the solution that the algorithm in this paper converges to. Is there a reason why it is considered the "true solution". A fair evaluation should be based on the property of the iterates which we actually care about.

I am willing to increase the score if my concerns are addressed properly.

**Limitations:**

yes

**Strengths And Weaknesses:**

Strength:
The mathematical analysis seems ok, but the techniques and proofs lack novelty.

Weaknesses

 1. The authors seem to motivate this paper by highlighting the fact that their algorithm converges to a unique solution. But it is unclear why this uniqueness of solution matters. There have been works which show convergence to a set or to sample-dependent fixed points, and this convergence is sufficient for evaluating the policy. The authors have not explained why this uniqueness matters.

2. There have also been works which show convergence to a point (Haque & Maguluri, 2024) and (Chen et al., 2025). The improvement over these results is the quadratic (instead of quartic) dependence on the "condition number". While mathematically interesting, the importance of this dependence is unclear as no comments are made about what these condition numbers represent in practice.

3. The literature survey seems to be missing work on average reward control algorithms such as RVI Q-learning and SSP Q-learning. This is surprising since the authors mention the tabular case of TD learning several times. Moreover, the tools used for analysis of RVI Q-learning are very similar in some aspects.

4. If I understood the simulation correctly, the numerical simulations seem to be doing an unfair comparison. The authors state "we evaluate all methods using their approximation error relative to the true solution rather than properties of their iterates." The definition of the true solution here is \theta^* which is the solution that the algorithm in this paper converges to. Is there a reason why it is considered the "true solution". A fair evaluation should be based on the property of the iterates which we actually care about.

---

> ### Author Rebuttal · Authors · 2026-03-30
>
> **1. Convergence to a point.** When the iterates converge only to a linear subspace, the parameter vector $\theta_t$ itself need not remain bounded.  Such numerical instability will become a practical challenge in implementations of TD learning.
>
> One challenge is **reproducibility**: one would like to run the same code twice and get the same results. The most prominent issue arises when policy evaluation is used as a subroutine in actor-critic algorithms. The vast majority of two-timescale stochastic approximation require the fast-timescale iterate (the critic) to converge to a *unique* equilibrium that varies continuously as a function of the slow-timescale iterate (the actor). Concretely, this dependency appears explicitly in the actor-critic literature as explicit requirements in the papers **Wu, Zhang, Xu, and Gu (NeurIPS 2020)**,  **Olshevsky and Gharesifard (SIAM J. Control Optim., 2023)**,  and **Hong, Wai, Wang, and Yang (SIAM J. Optim., 2023)**, among others.
>
> **Absence of a natural stopping criterion.** With convergence to a unique point, the standard stopping criterion $ || \theta_{t+1} - \theta_t || < \varepsilon$ is both valid and easy to implement, since it detects when the iterates have stabilized. With convergence to a subspace, this criterion fails.
>
> **2. Condition numbers in practice.**
>
> Great question; in general, the condition numbers are small. In the tabular setting, it is easy to see that all three condition numbers ($\eta_1$, $\eta_2$, $\eta_3$) are bounded above by $2\mu_{\min}$, where $\mu_{\min}$ is the smallest entry of the stationary distribution. For a chain with $n$ states, we have $\mu_{\min} \leq 1/n$.
>
> With $\eta$ that small, a quartic dependence $O(1/\eta^4)$ gives sample complexity scaling as $\Omega(n^4)$, whereas our quadratic dependence $O(1/\eta^2)$ gives $O(n^2)$.
>
> We also include a table of condition numbers on the problems we simulated to show how small they are.
>
>
> | Task                     | η₁              | η₂              | η₃              |
> |--------------------------|----------------|----------------|----------------|
> | Random Walk (50)         | 3.64e-3    | 2.77e-3        | 3.04e-3        |
> | Random Walk (100)        | 3.68e-3    | 2.63e-3        | 3.19e-3        |
> | Random Walk (1000)       | 1.91e-4    | 1.09e-4        | 1.82e-4        |
> | Frozen Lake              | 4.81e-4    | 4.26e-4        | 5.20e-5        |
> | Cliff Walking            | 5.04e-4    | 3.27e-4        | 7.27e-5        |
> | Taxi                     | 2.29e-4    | 1.75e-4        | 1.69e-5        |
> | Grid World (5x5)         | 4.21e-3    | 3.94e-3        | 1.60e-3        |
> | Grid World (10x10)       | 4.09e-4    | 2.44e-4        | 7.83e-5        |
> | Grid World (2x11)        | 3.77e-3        | 3.88e-3    | 1.23e-3        |
> | Deep sea                 | 1.73e-3    | 1.19e-3        | 4.16e-4        |
> | Deep sea (concave)       | 1.92e-3    | 1.28e-3        | 3.72e-4        |
> | Resource Gathering       | 4.35e-3        | 4.36e-3    | 2.07e-3        |
> | Fruit Tree (depth = 6)   | 1.07e-4    | 6.36e-5        | 3.75e-5        |
> | Fruit Tree (depth = 7)   | 1.92e-4    | 1.11e-4        | 5.75e-5        |
>
> **3. RVI + SSP Q-learning**
>
> We thank the reviewer for pointing us to these and we agree that a discussion of this line of work belongs in the paper.
>
> **4. Simulations.** This was an oversight on our part: we made several different comparisons but had neglected to include them all.  In the table below, we record distance between the final iterate and the subspace
> $$ \mathcal W^\star := \{W^\star + c e : c \in \mathbb R\}.$$
>
>
> | Task | DoubleChain | SingleChain | (Haque & Maguluri, 2024) \& (Zhang et al., 2021b) \& (Chen et al., 2025) | (Kim et al., 2025) |
> |---|---:|---:|---:|---:|
> | Random Walk (50) | $0.12 ± 0.02$ | **0.04 ± 0.00** | **0.04 ± 0.00** | **0.04 ± 0.00** |
> | Random Walk (100) | $0.19 ± 0.02$ | $0.07 ± 0.01$ | **0.06 ± 0.01** | **0.06 ± 0.01** |
> | Random Walk (1000) | $2.94 ± 0.13$ | **2.82 ± 0.03** | $2.84 ± 0.03$ | $2.92 ± 0.03$ |
> | Frozen Lake | **0.44 ± 0.00** | **0.44 ± 0.01** | **0.44 ± 0.01** | **0.44 ± 0.01** |
> | Cliff Walking | **0.62 ± 0.12** | $0.85 ± 0.02$ | $0.88 ± 0.01$ | $0.88 ± 0.01$ |
> | Taxi | $15.88 ± 2.96$ | **5.90 ± 0.38** | $9.05 ± 0.45$ | $8.96 ± 0.42$ |
> | Grid World (5x5) | $0.04 ± 0.01$ | **0.01 ± 0.00** | $0.12 ± 0.00$ | $0.12 ± 0.00$ |
> | Grid World (10x10) | **0.44 ± 0.03** | $0.49 ± 0.01$ | $0.45 ± 0.01$ | $0.48 ± 0.01$ |
> | Grid World (2x11) | $0.04 ± 0.00$ | **$0.01 ± 0.00$** | $0.07 ± 0.00$ | $0.08 ± 0.00$ |
> | Deep Sea | $0.22 ± 0.06$ | **0.09 ± 0.02** | $0.28 ± 0.01$ | $0.29 ± 0.01$ |
> | Deep Sea (concave) | $0.15 ± 0.03$ | **0.05 ± 0.01** | $0.15 ± 0.01$ | $0.15 ± 0.01$ |
> | Resource Gathering | **0.00 ± 0.00** | **0.00 ± 0.00** | **0.00 ± 0.00** | **0.00 ± 0.00** |
> | Fruit Tree (depth = 6) | $1.22 ± 0.10$ | **0.51 ± 0.03** | $0.85 ± 0.01$ | $0.85 ± 0.01$ |
> | Fruit Tree (depth = 7) | $0.86 ± 0.17$ | **0.38 ± 0.05** | $0.46 ± 0.06$ | $0.45 ± 0.05$ |

---

> > ### Author Rebuttal · Reviewer_ARxj · 2026-04-03
> >
> > With regards to points raised in the weaknesses section, only the reply to point 2. and 4 (partially) looks convincing. Remaining are not convincing. I am willing to change the score if these concerns are addressed too.

---

> > > ### Author Response · Authors · 2026-04-05
> > >
> > > We thank the reviewer for the continued interaction. In the following, we provide further responses to points 1 and 3.
> > >
> > > **On point 1**: It is indeed true that, theoretically, to implement policy gradient, it is sufficient to have set convergence for TD learning. However, since the set is a linear subspace, the TD learning iterates may still diverge within this subspace, which poses a crucial practical issue, as maintaining the stability and boundedness of the iterates is of vital practical importance.
> > >
> > > Let's illustrate the pitfalls of convergence to a subspace with a very simple **two-state** example. We will compare our double chain method to the differential TD method of (Zhang et al., Finite Sample Analysis of Average-Reward TD Learning and Q -Learning, NeurIPS 2021), which only converges to a subspace. The method itself is Algorithm 1 (Section 3.2) of the above paper.
> > >
> > > Let us take an example where the method of (Zhang et al., 2021) is  better than ours and investigate the effect of round off errors from iterates getting large.  For this, we run both algorithms with the same constant step-size.
> > >
> > > We consider a simple two state MDP with
> > > $P =  \left( \begin{array}{cc} 0.5 & 0.5 \\\\ 0.3 & 0.7 \end{array} \right)$ and rewards $r = \begin{pmatrix} 1 \\\\0 \end{pmatrix}$  and $\Phi = \left( \begin{array}{cc} 1 & 1 \\\\ 1 & -1 \end{array} \right)$.  The results are at:
> > >
> > >
> > >
> > >                            https://imgur.com/a/GSZHMe4
> > >
> > >
> > >
> > > We immediately see two things:
> > >
> > > (a) Our iterates remain bounded, whereas the iterates (Zhang et al. 2021) grow unbounded.
> > >
> > > (b) Iterate unboundedness can cause a problem because large values cause round off errors. Note that there is no problem in float64, but decreasing to float8 or float16 completely ruins the performance. By contrast, our method has the same performance (up to 0.01%) in float64/float16/float8.
> > >
> > > We emphasize two additional points. First, we see this phenomenon **even a simple example with two-states.** Iterates can larger even more quickly on larger examples. Second, the issue is endemic to any algorithm that converges to a subspace and is not particular to (Zhang et al., 2021).
> > >
> > > We remark that this discussion is an addition to the early points we already made on convergence to a point (i.e., convergence to a point allows one to use the simple $||\theta_{t} - \theta_{t+1} || < \epsilon$ stopping criterion, ensures results are reproducible, and is necessary for allmost all actor-critic analyses).
> > >
> > >
> > >
> > >
> > > **On point 3**:
> > > We thank the reviewer for pointing us to the RVI Q-learning and SSP Q-learning literature, and we agree that a discussion of this line of work belongs in the paper. We will add citations to Abounadi, Bertsekas, and Borkar (SIAM J. Control Optim., 2001) and more recent works, e.g., Wan, Naik, Sutton, and others.  Nevertheless, we would like to respectfully push back on the suggestion that the analytical tools are similar to ours.
> > >
> > > To clarify, the majority of results on average-reward Q-learning focus on asymptotic convergence in the tabular setting. To the best of our knowledge, no existing work provides a finite-time analysis of Q-learning (RVI or SSP) with function approximation. Even for tabular Q-learning, finite-time analysis has appeared only very recently (Chandak et al., 2025; Chen, 2025).
> > >
> > > That being said, from a technical perspective, the key to analyzing stochastic iterative algorithms is the construction of a valid Lyapunov function. From this standpoint, SSP or RVI Q-learning and TD-learning are fundamentally different. In particular, the defining feature of Q-learning is the Bellman optimality operator, which is nonlinear due to the $\max$. For example, existing results analyze this operator using the span seminorm $|\cdot|_s$, exploiting the crucial property that $T^*$ is either contractive or nonexpansive under $|\cdot|_s$, depending on the structural assumptions on the MDP.
> > >
> > > Our analysis, by contrast, works with the D-norm and Dirichlet seminorm, and the central technical device is gradient splitting, which recasts the TD update as a noisy gradient step on a certain quadratic objective. This enables a direct SGD-style finite-sample analysis that **has no counterpart** in the Q-learning literature.
> > >
> > > Relatedly, both methods take different approaches to resolving the constant-shift degeneracy in the solution: RVI-based approaches designate a reference state whose value anchors the solution, while SSP-based approaches impose a boundary condition at a recurrent state. By contrast, our $\mu^T W = 0$  normalization is the natural choice in the TD setting, since $\mu$
> > > is already central to the average-reward analysis, and it is this choice that enables the gradient splitting approach.

---

### Official Review · Reviewer_u4h6 · 2026-03-18

**Soundness:** 3
**Presentation:** 3
**Significance:** 3
**Originality:** 3
**Overall Recommendation:** 5
**Confidence:** 2

**Summary:**

The paper gives an algorithm for evaluating the average reward of running a policy that takes $\tilde{\mathcal{O}}(\epsilon^{-1}\eta^{-2})$ samples to get an $\epsilon$ close estimate when the Markov chain induced by the policy on the MDP has condition number $\eta$. This matches the number of samples needed to do policy evaluation in discounted MDPs.

To accomplish this, the authors prove that the average-reward value function $W^*$ satisfies a projected Bellman equation for the correct choice of projection. They then perform a fixed point iteration from samples. The problem with doing this naively is that it involves estimating a term of the form $\mu\mu^\top$, where $\mu$ is the stationary distribution of the Markov chain, which results in a double sampling error if the empirical estimate of the stationary distribution is squared. To fix this, the authors introduce a method of approximating this quantity from samples by running two independent instances of the Markov chain multiplying the estimates from each instead. They then introduce a single chain method that can do this without having to use two independent instances, but has slower convergence.

**Compliance With Llm Reviewing Policy:**

Affirmed.

**Key Questions For Authors:**

1) What should I expect $\eta_1$, $\eta_2$, and $\eta_3$ to equal in tabular settings?
2) What would it take to extend these results beyond policy evaluation to instead solve for the optimal average case policy?
3) The step size seems to require being set to the inverse of the condition number. In the experiments, the step size seems to be a finetuned constant. Is it possible to compute the condition number $\eta$ efficiently? Or must this be guessed in practice?
4) Can you elaborate on the connection between $\eta_1$ and $\eta_{\textrm{discounted}}$? One of the main claimed contributions of the paper is that the algorithm achieves the same dependence on the condition number as discounted policy evaluation, but the two quantities seem to be defined differently.
5) If the one-chain algorithm was added to Table 1, what would its row look like?

**Limitations:**

yes

**Strengths And Weaknesses:**

Strengths:
* The paper is technically very interesting, introducing a very natural method of doing average reward policy evaluation that improves on past work and seems to be simpler too.
* The presentation is strong. The authors do a thorough job comparing this result to prior work, and the main idea is explained clearly. The main body flows well, the ideas are motivated well, and the paper is easy to read, despite the proofs in the appendix being very technically involved.
* I didn't check over the proofs in the appendix in detail, but all the math in the main body and the high level ideas of the proofs seem correct to me.
* The algorithms are empirically tested and they consistently across a wide range of environments do better than the baselines, confirming the theory.

Weaknesses:
* The algorithms only solve policy evaluation and don't give you a way to find the optimal average reward policy.
* The improvement over prior work requires running two simulations of the policy at once. The single chain algorithm introduced that doesn't need two simulations does not outperform prior work in terms of the dependence on the condition number.

Minor Errors:
* There is a $E_\mu$ missing on line 426 from $E_\mu[f(s_t, \hat{s}_t)] - g(s_t, s_t')$.
* A few citations in the introduction should be a \citet instead of a \citep, e.g. on lines 73, 74, and 81.

---

> ### Author Rebuttal · Authors · 2026-03-30
>
> We are grateful for the thorough and positive assessment, and especially for the careful reading of the main body.
>
> **Weaknesses**
>
> Regarding policy evaluation vs. policy: please see our response to Reviewer GJn9, where we make the point that better policy evaluation algorithms have historically lead to better policy optimization algorithms.
>
> Regarding the single-chain algorithm's condition number dependence: this is accurate and we appreciate the precise characterization. We will make this tradeoff more explicit in Table 1 and the surrounding discussion (see Q5 below).
>
> **Minor Errors**
>
> We thank the reviewer for catching these. We will add the missing $\Phi$ on line 426 and correct the `\citep` → `\citet` on lines 73, 74, and 81.
>
> **Key Questions**
>
> **Q1: What should $\eta$, $\eta_1$, $\eta_2$ equal in the tabular case?**
>
> All three of these condition numbers are upper bounded by $2 \mu_{min}$, where $\mu_{\min}$ is th smallest entry of the stationary distribution. Since $\mu_{\min} \leq O(1/n)$, our quartic-to-quadratic improvement is at least an $n^4$ to $n^2$ improvement.
>
> This bound can be established via explicit test vectors. For $\eta_1$: take $x = e_{s*}$, the standard basis vector at the state $s^* = \arg\min_s \mu(s)$. For $\eta_2$: take $x(s*) = \sqrt{(n-1)/n}$ and $x(s) = -1/\sqrt{n(n-1)}$ for $s \neq s*$. For $\eta_3$: since $\min_{\|x\|=1} x^\top D x = \mu_{\min}$ and the minimum of $ y^T D(I-P)y$ over the set $y$ of unit D-norm that is D-orthogonal to the all ones vector is upper bounded by $2$ (already established in Appendix B of our paper), we have $\eta_3 \leq 2\mu_{\min}$.
>
> Please also see our response to Reviewer ARxj for a table of these condition numbers on the examples we simulated.
>
> **Q2: What would it take to extend these results to optimal average-reward control?**
>
> The natural path is to use our policy evaluation algorithm as the critic inside an average-reward actor-critic loop. We do not see any difficulties in using the double-chain algorithm and coupling it with the AC analysis we are most familiar with (**Olshevsky, Gharesifard, SIAM Jour. on Control and Optim., 2023**). As far as we can tell, it should be possible to redo the analysis of that work almost verbatim. The single-chain algorithm is more challenging to extend and would require a three-timescale approximation.
>
>
> **Q3: Can the condition number be computed efficiently, or must the step size be guessed?**
>
> Guessed. It's possible to compute the condition number (its a quadratic optimization problem which could be solved with SGD in the unknown-MDP case), but it's not obvious that this is actually any faster than solving the entire policy evaluation problem. We note that all step-sizes for TD (discounted or average reward) which achieve $1/t$ rates have step-sizes which are bounded as functions of condition number (not just the present paper).
>
> One approach to make our method parameter-free is to combine it with Polyak–Ruppert averaging, which robustly achieves an $1/t$ rate with a more flexible stepsize schedule. However, this introduces an additional iterate (i.e., the running average) and, while an interesting future direction, is beyond the scope of this work.
>
> **Q4: What is the relationship between $\eta$ and the discounted condition number?**
>
> We use the closest analogue, which is condition number in Liu & Olshevsky (ICML 2021) for discounted TD;
>
> $$\eta_{\rm discount} = \min_{\|x\|=1} \;\gamma\|\Phi x\|_{\rm Dir}^2 + (1-\gamma)\|\Phi x\|_D^2,$$
>
>
>
> Our condition number for average reward TD is $\eta$ as defined in Eq. (4):
>
> $$\eta = \min_{\|x\|=1} \;\|\Phi x\|_{\rm Dir}^2 + (\mu^T \Phi x)^2,$$
>
>  The structural analogy is exact: in both cases a Dirichlet seminorm term is combined with a second term that pins down the solution along the direction that would otherwise be degenerate. In the discounted case the pinning term is $(1-\gamma)\|\Phi x\|_D^2$, which reflects the contractive effect of discounting. In the average reward case it is $(\mu^T \Phi x)^2$, which enforces the steady-state constraint $\mu^T W^* = 0$.
>
>
>
> **Q5: What would the single-chain algorithm's row in Table 1 look like?**
>
> If the single-chain algorithm were added to Table 1, what would its row look like? A concise row would be: setting: tabular and linear; converges to a sample-independent point: yes; sample complexity: $\tilde{O}(\epsilon^{-1} \eta_1^{-4})$ in the common regime discussed in the paper; scaling with dimension: no explicit \(d\)-dependence under the paper's feature normalization convention; iterate: last; method: coupled stochastic approximation. This matches the theorem-level message: the single-chain method preserves pointwise convergence and the overall $\tilde{O}(1/T)$ rate, but its condition-number dependence is worse than quadratic.

---

> > ### Author Rebuttal · Reviewer_u4h6 · 2026-04-04
> >
> > I appreciate the author's response, my concerns have been adequately addressed.

---

### Decision · Program_Chairs · 2026-04-30

**Decision:**

Accept (regular)

**Comment:**

This paper proposes new algorithms for the policy evaluation problem in average reward MDP. The advantage of the algorithm is that it is guaranteed to converge to a single point (which can address the stopping criterion problem) for both tabular and linear setup (here linear setup does not necessarily supersede tabular setup). The convergence rate also has a better dependency on condition number than some prior works. Overall, the reviewers agree that this paper makes solid technical contribution. I, therefore, recommend Accept.

ARxj is eventually concerned about motivation and related works. I think the authors did a good job in motivating why converging to a unique point is important in rebuttal. For related works, I do think there are some missing pieces, including which will further strengthen the work.
The first is [x]. Particularly, their Theorem 5 confirms that a centered variant of their tabular differential TD converges to a single point almost surely (though no rate). They also make use of the fact that $u^\top W_* = 0$. The second is [y]. Particularly, they prove that the standard linear average reward TD converges to a set, making no assumptions on the feature.

[x] Wan, Y., Naik, A., & Sutton, R. S. Learning and planning in average-reward markov decision processes. ICML 2021.
[y] Xie, Z., Liu, X., Chandra, R., & Zhang, S. Finite sample analysis of linear temporal difference learning with arbitrary features. NeurIPS 2025.